# SHIELD: Multi-task Multi-distribution Vehicle Routing Solver with Sparsity & Hierarchy in Efficiently Layered Decoder

## Abstract

Recent advances toward foundation models for routing problems have shown great potential of a unified deep model for various VRP variants. However, they overlook the complex real-world customer distributions. In this work, we advance the Multi-Task VRP (MTVRP) setting to the more realistic yet challenging Multi-Task Multi-Distribution VRP (MTMDVRP) setting, and introduce SHIELD, a novel model that leverages both *sparsity* and *hierarchy* principles. Building on a deeper decoder architecture, we first incorporate the Mixture-of-Depths (MoD) technique to enforce sparsity. This improves both efficiency and generalization by allowing the model to dynamically choose whether to use or skip each decoder layer, providing the needed capacity to adaptively allocate computation for learning the task/distribution specific and shared representations. We also develop a context-based clustering layer that exploits the presence of hierarchical structures in the problems to produce better local representations. These two designs inductively bias the network to identify key features that are common across tasks and distributions, leading to significantly improved generalization on unseen ones. Our empirical results demonstrate the superiority of our approach over existing methods on 9 real-world maps with 16 VRP variants each.

## 1 Introduction

Combinatorial optimization problems (COPs) appear in many real-world applications, such as logistics (Cattaruzza et al., 2017) and DNA sequencing (Caserta & Voß, 2014), and have historically attracted significant attention (Bengio et al., 2021). A key example of COPs is the Vehicle Routing Problem (VRP), which asks: *Given a set of customers, what is the optimal set of routes for a fleet of vehicles to minimize overall costs while satisfying all constraints?* Traditionally, they are solved with exact or approximate solvers. However, these solvers are either inefficient for large instances or rely heavily on expert-designed heuristic rules. Recently, the emerging Neural Combinatorial Optimization (NCO) community has been increasingly focused on developing novel neural solvers for VRPs based on deep (reinforcement) learning (Kool et al., 2018; Kwon et al., 2020; Bogyrbayeva et al., 2024). These solvers learn to construct solutions autoregressively, improving efficiency and reducing the need for domain knowledge, showing significant promise over traditional solvers.

Motivated by the recent breakthroughs in foundation models (Floridi & Chiriatti, 2020; Touvron et al., 2023; Achiam et al., 2023), a notable trend in the NCO community is the push towards developing a unified neural solver for handling multiple VRP variants, known as the Multi-Task VRP (MTVRP) setting (Liu et al., 2024; Zhou et al., 2024; Berto et al., 2024). These solvers are trained on multiple VRP variants and show impressive zero-shot generalization to new tasks. Compared to single-task solvers, unified solvers offer a key advantage: there is no longer a need to construct different solvers or heuristics for each specific problem variant. However, despite the importance of the MTVRP setup, it does not fully capture real-world industrial applications, as the underlying distributions are assumed to be uniform, lacking the structural properties of real-world data.

In this work, we extend the MTVRP framework to real-world scenarios by incorporating realistic distributions (Goh et al., 2024). Consider, for example, a logistics company operating across multiple cities/countries, with each region having a fixed set of $M$ locations, governed by its geographical

layout. When a subset of $V$ orders arises, the problem is reduced to serving only those customers. To model this, we generate realistic distributions by selecting smaller subsets of $V$ from the fixed set of $M$ locations, ensuring that $V$ retains the geographical distribution characteristics of $M$. A unified model with strong performance across tasks and distributions allows for flexible, efficient deployment. This transforms MTVRP into the Multi-Task Multi-Distribution VRP (MTMDVRP), a novel and challenging setting that, to our knowledge, has not been explored in the literature.

Nevertheless, MTMDVRP poses unique challenges for learning unified neural VRP models. First, beyond managing the diverse constraints of MTVRP, the model must further learn to handle arbitrary, distribution-specific layouts. Unfortunately, task-related contexts often interdepend with distribution-related contexts during decision-making (e.g., selecting the next node), adding further complexity. Moreover, balancing shared and task/distribution-specific representations becomes more difficult, as the model needs to generalize across a broader representation space to serve as a more foundational NCO model. Consequently, this calls for learning unified deep models that balances the expressiveness required for complex decision-making with the simplicity needed for efficient generalization – an issue we explore in depth in this paper.

To this end, we introduce **S**parsity & **H**ierarchy **i**n **E**fficiently **L**ayered **D**ecoder (SHIELD) to address the above challenges with two key innovations. First, SHIELD leverages *sparsity* by incorporating a customized Mixture-of-Depths (MoD) approach (Raposo et al., 2024) to the NCO decoders. While adding more decoder layers can improve predictive power, the autoregressive nature of neural VRP solver significantly hampers efficiency. In contrast, our MoD is designed to dynamically adjust the proper computational depth (number of decoder layers) based on the decision context. This allows adaptively allocated computation for learning the task/distribution specific and shared representations, while acting as a regularization mechanism to prevent overfitting by possibly reducing redundant computations. Secondly, we employ a clustering mechanism that considers *hierarchy* during node selection by forcing the learning of a small set of key representations of unvisited nodes, enabling compact modeling of the complex decision-making information. Together, these two designs encourage the model to learn some compact, simple, generalizable representations with limited computational budgets, enhancing generalization across tasks and distributions, which is also in line with the Information Bottleneck perspective. This paper highlights the following contributions:

- We propose Multi-Task Multi-Distribution VRP (MTMDVRP), a novel, more realistic yet challenging setting that better represents real-world industry scenarios.

- We present SHIELD, a neural solver that leverages *sparsity* through a customized NCO decoder with MoD layers and *hierarchy* through context-based cluster representation, advancing towards a more generalizable foundation model for neural VRP solvers.

- We demonstrate the impressive in-distribution and generalization benefits of SHIELD via extensive experiments across 9 real-world maps and 16 VRP variants, achieving state-of-the-art performance compared to existing unified neural VRP solvers.

## 2 RELATED WORK

**Multi-task VRP Solver.** Recent work in (Liu et al., 2024) explored training of a Multi-Task VRP solver across a range of VRP variants which share a set of common features indicating the presence or absence of specific constraints. Zhou et al. (2024) enhanced the model architecture by introducing a Mixture-of-Experts within the transformer layers, allowing the model to effectively capture representations tailored to different tasks. These studies focus on zero-shot generalization, where models are trained on a subset of tasks and evaluated on unseen tasks that are combinations of common features. Additionally, other studies (Wang & Yu, 2023; Drakulic et al., 2024) investigate this promising direction, but with different problem settings. Alternatively, Berto et al. (2024) improved convergence robustness by training on all possible tasks within a batch using a mixed environment. In this work, we mainly build on the setting presented by Liu et al. (2024); Zhou et al. (2024).

**Generalization Study.** Joshi et al. (2021) highlighted the generalization challenge faced by neural combinatorial solvers, where their performance drops significantly on out-of-distribution (OOD) instances. Numerous studies have sought to improve generalization performance in cross-size (Bdeir et al., 2022; Son et al., 2023), cross-distribution (Wang et al., 2021; Jiang et al., 2022; Bi et al., 2022; Zhang et al., 2022; Zhou et al., 2023), and cross-task (Lin et al., 2024; Liu et al., 2024; Zhou

et al., 2024; Berto et al., 2024) settings. However, their methods are tailored to specific settings and cannot handle our MTMDVRP setup, which considers crossing both tasks and realistic customer distributions. While a recent work Goh et al. (2024) explores more realistic TSPs, their approach still struggles with complex cross-problem scenarios. In this paper, we take a step further by exploring generalization across both different problems and real-world distributions in VRPs. We refer the reader to Appendix A.1 for details regarding single-task VRP solvers.

## 3  PRELIMINARIES

**CVRP and its Variants.** The CVRP is defined as an instance of $N$ nodes in a graph $\mathcal{G} = \{\mathcal{V}, \mathcal{E}\}$, where the depot node is denoted as $v_0$, customer nodes are denoted as $\{v_i\}_{i=1}^N \in \mathcal{V}$, and edges are defined as $e(v_i, v_j) \in \mathcal{E}$ between nodes $v_i$ and $v_j$ such that $i \neq j$. Every customer node has a demand $\delta_i$, and every vehicle has a maximum capacity limit $Q$. For a given problem, the final solution (tour) can be presented as a sequence of nodes with multiple sub-tours. Each sub-tour represents a vehicle's path, starting and ending at the depot. As a vehicle visits a customer node, the demand is fulfilled and subtracted from the vehicle's capacity. A solution is considered feasible if each customer node is visited exactly once, and the total demand in a sub-tour does not exceed the capacity limit of the vehicle. In this paper, we consider the nodes defined in Euclidean space within a unit square $[0, 1]$, and the overall cost of a solution, $c(\cdot)$, is calculated via the total Euclidean distance of all sub-tours. The objective is to find the optimal tour $\tau^*$ such that the cost is minimized, given by $\tau^* = \mathrm{argmin}_{\tau \in \Phi} c(\tau | \mathcal{G})$ where $\Phi$ defines the set of all possible solutions.

We define the following practical constraints that are integrated with CVRP: (1) *Open route (O)*: The vehicle is no longer required to return to the depot after visiting the customers; (2) *Backhaul (B)*: Demand $\delta_i$ is a positive value, indicating that goods are unloaded at a customer node. Instead, demand on some nodes can be negative, meaning that these nodes will load goods into the vehicle. Practically, this mimics the pick-up and drop-off scenarios in logistics. We label nodes with positive demand $\delta_i > 0$ as linehauls, and nodes with negative demand $\delta_i < 0$ as backhauls. Note that routes can have a mixed sequence of linehauls and backhauls without strict precedence; (3) *Duration Limit (L)*: Each sub-tour is upper bounded by a threshold limit on the total length; (4) *Time Window (TW)*: Each node $v_i$ is defined with a time window $[w_i^o, w_i^c]$, signifying the opening and close times of the window, and $s_i$ the service time at a node. Essentially, a customer can only be served if the vehicle arrives within the time window, and the total time taken at the node is the service time. If a vehicle arrives earlier, it has to wait until $w_i^o$. All vehicles have to return to the depot before $w_0^c$.

**Neural Constructive Solvers.** Neural constructive solvers are typically parameterized by a neural network, where a policy, $\pi_\theta$, is trained by reinforcement learning to construct a solution sequentially (Kool et al., 2018; Kwon et al., 2020). The attention-based mechanism (Vaswani, 2017) is popularly used, with attention scores guiding the decision-making process in an autoregressive fashion. The feasibility of a solution can be managed through masking, where invalid moves are excluded during the construction process. Generally, neural constructive solvers employ an encoder-decoder architecture and are trained as sequence-to-sequence models (Sutskever, 2014). The probability of a sequence can be factorized using the chain-rule of probability, $p_\theta(\tau | \mathcal{G}) = \prod_{t=1}^T p_\theta(\tau_t | \mathcal{G}, \tau_{1:t-1})$. The encoder typically stacks multiple transformer layers to extract node embeddings, while the decoder generates solutions autoregressively using a contextual embedding $\mathbf{h}_{(c)}$. We leave additional details about the architecture to Appendix A.3. The contextual embedding can be represented as $\mathbf{h}_{(c)} = \mathbf{h}_{\mathrm{LAST}}^L + \mathbf{h}_{\mathrm{START}}^L$. Then, the attention mechanism is used to produce the attention scores. Concretely, the context vectors $\mathbf{h}_{(c)}$ serves as query vectors, while the keys and values are the set of $N$ node embeddings. This is mathematically represented as

$$a_j = \begin{cases} U \cdot \mathrm{TANH}(\frac{\mathbf{Q}\mathbf{K}^\top}{\sqrt{\mathrm{DIM}}}) & j \neq \tau_{t'}, \forall t' < t \\ -\infty & \text{otherwise} \end{cases}, \ p_i = p_\theta(\tau_t = i | s, \tau_{1:t-1}) = \frac{e^{a_j}}{\sum_j e^{a_j}} \quad (1)$$

where $U$ is a clipping function and DIM the dimension of the latent vector. These attention scores are then normalized using a softmax function to generate the probability distribution. Finally, given a baseline function $b(\cdot)$, the policy is trained with the REINFORCE algorithm (Williams, 1992) and gradient ascent, with the expected return $J$ and the reward of each solution $R$ (i.e., the negative length of the solution tour): $\nabla_\theta J(\theta) \approx \mathbb{E}\Big[(R(\tau^i) - b^i(s))\nabla_\theta \log p_\theta(\tau^i | s)\Big]$.

**Mixture-of-Experts and Mixture-of-Depths.** Previous work (Liu et al., 2024) demonstrated the ability of state-of-the-art transformers such as POMO (Kwon et al., 2020) to generalize across MTVRP instances. More recently, (Zhou et al., 2024) improved upon the transformer architecture with the introduction of the Mixture-of-Experts. Formally, a MoE layer consists of $m$ experts $\{E_1, E_2, ..., E_m\}$, whereby each expert is a feed-forward MLP. A gating network $G$ produces a scalar score based on an input $x$ which is then responsible for deciding how the inputs are distributed to the experts. A MoE layer's output can be defined as $\text{MOE}(x) = \sum_{j=1}^{m} G(x)_j E_j(x)$. The gating network operates such that only the top-k experts are activated, so as to prevent computation from exploding. For MVMoE, Zhou et al. (2024) introduces MoE layers at each transformer block at the token-level, meaning that every token uses at most $k$ experts. Additionally, a hierarchical gate is introduced in the decoder at the problem level, whereby depending on the problem instance, the network learns to decide whether or not to use experts at each decoding step.

Apart from MoE, MoD is introduced in an effort to improve computational efficiency in large language models (LLMs) (Raposo et al., 2024). Effectively, the authors replace alternate transformer layers in the LLM's encoder, making learning embeddings more computationally efficient. Now, instead of gating network $G(x)$ routing to various experts, it routes tokens through the transformer layer or bypasses it. The capacity of $G(x)$ defines the total number of tokens allowed for a layer. Empirical evidence showed improvement in training loss by intertwining these sparser layers.

## 4 METHODOLOGY

### 4.1 MTVRP AND MTMDVRP SETUP

Formally, the optimization objective of a MTVRP instance is given by

$$\min(C(X)) = \mathbb{E}_{k \sim \mathcal{K}} \left[ \sum_{s \in \mathcal{S}} \sum_{p_i \in s} d(p_i, p_{i+1}) \right] \tag{2}$$

where $\mathcal{K}$ the set of all tasks, $\mathcal{S}$ the set of all sub-tours in an instance, $p_i$ the $i$-th node in the sequence of $s$, and $d(\cdot, \cdot)$ the Euclidean distance function. For the MTMDVRP in this paper, we expand on the MTVRP scenarios in (Liu et al., 2024; Zhou et al., 2024). The $x_i$ and $y_i$ coordinates for the instances are now sampled from a known underlying distribution of points, as opposed from the uniform distribution. This enables the sample problems to mimic most of the structural distributions and patterns available in the problem. The optimization objective can be summarized as follows

$$\min(C(X)) = \mathbb{E}_{q \sim \mathcal{Q}} \left[ \mathbb{E}_{k \sim \mathcal{K}} \left[ \sum_{s \in \mathcal{S}} \sum_{p_i \in s} d(p_i, p_{i+1}) \right] \right] \tag{3}$$

where $\mathcal{Q}$ is the set of all distributions. The following practical scenario can visualize our MTMDVRP: assume a logistics company X deploys a deep learning model to solve multiple known variants for its current business. In an ideal world, it would have access to all forms of logistics problems generated across all possible structured distributions in the world, whereby a country map $q \in \mathcal{Q}$. Realistically, company X only has historical data in some tasks and presence in a handful of countries, such that $q' \in \mathcal{Q}'$, whereby $\mathcal{Q}' \subset \mathcal{Q}$, meaning that it only has data drawn from a subset of distributions in $\mathcal{Q}$. Likewise, it has only faced a subset of tasks such that $k' \in \mathcal{K}', \mathcal{K}' \subset \mathcal{K}$. Based on this historical data, company X can train a single model using $\mathcal{Q}'$ and $\mathcal{K}'$. Now, if company X wishes to expand its presence to other parts of the world, it would see new data samples from new distributions and meet new tasks that were not present in the training set. Thus, it would be highly beneficial for company X to be able to apply its model readily. To do so, the model has to be robust to the task and distribution deviation simultaneously, suggesting strong generalization properties across these two aspects.

**Challenges of MTMDVRP.** While adding distributions may seem straightforward, it introduces significant complexity. First, the model must learn representations that capture both constraint and distribution context when selecting the next node to visit. However, in MTMDVRP, task and distribution contexts often interdepend, complicating decision-making. For example, in a skewed map such as Egypt (EG7146) in Figure 5 in Appendix A.13, the task complexity is closely tied to the geographic layout. The depot's position significantly impacts the solution; a depot near clustered

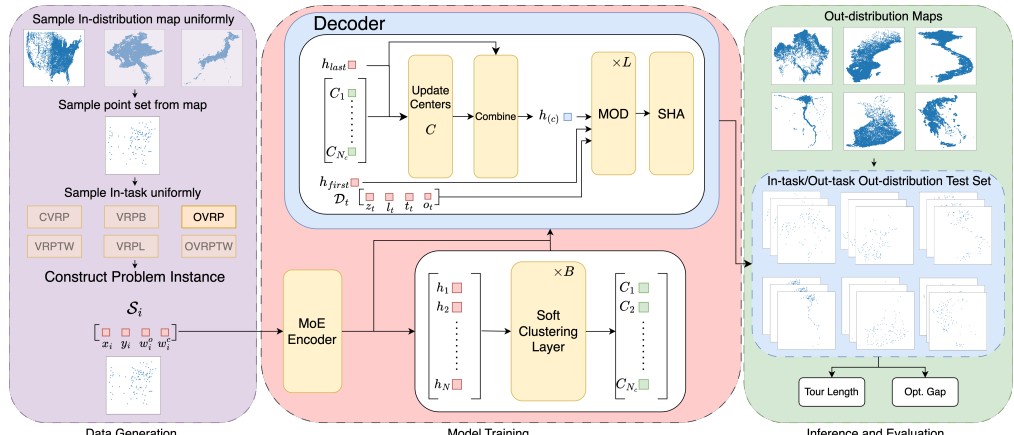

Figure 1: Overall proposed approach for MTMDVRP. First, in-distribution maps are sampled uniformly and a set of points is sampled. After which, the in-task is sampled uniformly. Based on these, a batch of problem instances is formed and passed through SHIELD. SHIELD encompasses an MoE encoder, followed by a context-based clustering layer, and finally the MoD decoder. The decoder is applied autoregressively to in-task/out-task out-distribution instances and the optimality gap is calculated using known solvers.

customer nodes is less complex to solve than one located in a sparse region with distant customer nodes. Additionally, balancing shared and task/distribution-specific representations is more difficult, as the model must generalize across a broader space to serve as a foundational NCO model. Thus, strong generalization across both tasks and distributions is essential for a robust foundation model.

For our setup, we adopt the following feature set. At each epoch, we are faced with a problem instance $i$ such that $\mathcal{S}_i = \{x_i, y_i, \delta_i, w_i^o, w_i^c\}$, where $x_i$ and $y_i$ are the respective coordinates, $\delta_i$ the demand, $w_i^o$ and $w_i^c$ the respective opening and closing times of the time window. This is passed through the encoder resulting in a set $\mathbf{H}$ of $d$-dimensional embeddings. At the $t$-th decoding step, the decoder receives this set of embeddings $\mathbf{H}$, the clustering embeddings $\mathbf{C}$, and a set of dynamic features $\mathcal{D}_t = \{z_t, l_t, t_t, o_t\}$, where $z_t$ denotes the remaining capacity of the vehicle, $l_t$ the length of the current partial route, $t_t$ the current time step, and $o_t$ indicates if the route is an open route or not.

## 4.2 INFORMATION BOTTLENECK AND GENERALIZATION

In the context of MTVRP, the MoE model was proposed as an effective learning framework for multi-task settings (Zhou et al., 2024). However, it is not immediately clear why simply improving predictive power with a mixture model would be particularly beneficial in this context. We examine this from the perspective of the Information Bottleneck principle (Tishby et al., 2000; Tishby & Zaslavsky, 2015; Saxe et al., 2019), which suggests that representations that are highly predictive but have minimal complexity are better suited for generalization. In Multi-Task and Multi-Distribution VRP, there is invariably shared information across tasks or distributions that can be leveraged, while representations must also retain task or distribution specific information to improve predictive performance. Federici et al. (2020) studied the multi-view case wherein different views share common label and showed that maximizing joint information between views with the shared labels is helpful. Contrapositively, this implies that in scenarios where labels or distributions differ, such as in the MTMDVRP setting, balancing shared and task-specific information is essential for generalization. However, MoE lacks an inductive bias to enforce this balance.

We propose that an adaptive learning approach, which regulates the balance between learning shared and task-specific representations, is more appropriate. The customized MoD approach addresses this by enforcing *sparsity* through possibly reduced network depths and lighter computation, forcing the model to learn generalizable representations across tasks/distributions. The clustering mechanism forces the network to condense information into a handful of representations. In a multi-task scenario, we posit that these encourage the network to efficiently generalize by balancing the computational budget for task-specific information while leaving common information to be learned across other tasks or distributions, encouraging efficient generalization across tasks and distributions.

### 4.3 GOING DEEPER BUT SPARSER

Our proposed architecture is shown in Figure 1. In order to increase the predictive power of the MV-MoE, one can easily hypothesize that increasing the number of parameters would necessitate that. However, due to the nature of the autoregressive decoding, we find that this quickly becomes extremely complex. Instead, we propose the integration of the Mixture-of-Depths (MoD) (Raposo et al., 2024) approach into the decoder. Given a dense transformer layer and $N$ tokens, MoD selects the top $\beta$-th percentile of tokens to pass through the transformer layer. In contrast, the remaining unselected tokens are routed around the layer with a residual connection around the layers, avoiding the need to compute all $N$ attentional scores. Formally, the layer can be represented as follows

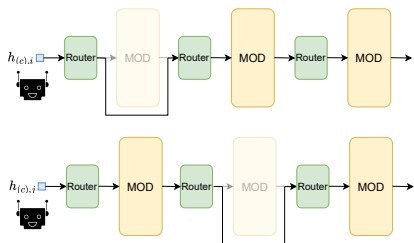

Figure 2: Token is routed differently for each agent depending on the router.

$$\mathbf{h}_i^{l+1} = \begin{cases} r_i^l f_i(\tilde{\mathbf{H}}^l) + \mathbf{h}_i^l & \text{if } r_i^l > P_\beta(\mathbf{r}^l) \\ \mathbf{h}_i^l & \text{if } r_i^l < P_\beta(\mathbf{r}^l) \end{cases} \qquad (4)$$

where $r_i = \mathbf{W}_\theta^\top \mathbf{h}_i^l$ is router score given for token $i$ at layer $l$, $W_\theta$ is learnable parameters in the router that converts a $d$-dimensional embedding into a scalar score, $\mathbf{r}^l$ the set of all router scores at layer $l$, $P_\beta(\mathbf{r}^l)$ the $\beta$-th percentile of router scores, and $\tilde{\mathbf{H}}$ the subset of tokens in the $\beta$-th percentile. In this work, we utilize token-level routing, whereby each token is passed through the router, and the top $\beta$ percentile tokens are selected. By controlling $\beta$, we control the sparsity of the architecture by determining how many tokens are passed into the layer for processing. For each layer, we apply this routing mechanism to $\mathbf{h}_{(c)}$, the contextual vectors. Each transformer layer still receives all $N$ node embeddings together with a mask that determines whether a previous node has been visited. Effectively, we limit the total number of query tokens to the transformer layer in the decoder. As each query token is the contextual vector $\mathbf{h}_{(c)}$, this means that the network learns to identify which *current locations* are more important to be processed. This effect naturally introduces sparsity in the architecture: not all tokens are processed multiple times equally as it is passed through the decoder.

### 4.4 CONTEXTUAL CLUSTERING

Apart from sparsity in compute, we introduce hierarchy in the form of representation. Goh et al. (2024) first showed that for structured TSPs, one can apply a form of soft-clustering to summarize the set of unvisited cities into a handful of representations. This is then used to guide agents, providing crucial information about the groups of nodes left in the problem, which is highly useful for structured distributions.

In addition to structured distributions, the MTMDVRP has underlying commonalities among its tasks. As such, we hypothesize that nodes and it's associated task features can be grouped together. While spatial structure can typically be measured in Euclidean space, it is not so straightforward for tasks and its features. Thus, an EM-inspired soft clustering algorithm in latent space provides a sensible approach to this problem. We first define a set of $\mathbf{C} \in \mathrm{R}^{N_c \times d}$ representations, such that $N_c$ of these denote the number of cluster centers. The soft clustering algorithm poses the forward pass of the attention layer as an estimation of the E-step, and the re-estimation of $\mathbf{C}$ using the weighted sum of the learnt attention weights as the M-step. Repeated passes through this layer simulate a roll-out of a pseudo-EM algorithm. Effectively, the network learns the initial cluster centers and the parameters required to transform these centers to the final centroids based on the input embeddings.

In this work, we modify the soft clustering algorithm and introduce context prompts to capture the task dependencies. For the same spatial graph, if the task at hand is different, the clustering mechanism should be sufficiently flexible to accommodate the various intricacies of the task. To handle this, we model this contextual prompt as a latent representation $\alpha_k = \mathbf{W}_\theta^\top \gamma_k$ where and $\mathbf{W}_\theta$ is a set of learnable parameters that transforms the constraints to latent representations, and $\gamma_k$ is a one-hot encoded vector of constraints for task $k$, such that each feature corresponds to a constraint. In this work, we have $\gamma_k = [\gamma_k^1, \gamma_k^2, \gamma_k^3, \gamma_k^4]$, where $\gamma_k^1$ denotes *open*, $\gamma_k^2$ denotes *time-window*, $\gamma_k^3$

denotes *route length*, and $\gamma_k^4$ denotes *backhaul* constraints. Since the model learns to convert these to latent vectors, we hypothesize that it learns to effectively stitch the various constraints together to form unique representations for all 16 variants. We then pass this vector onto the clustering layer:

$$\hat{\mathbf{h}}_i = \mathbf{W}_H \mathbf{h}_i, \hat{\mathbf{c}}_j = \mathbf{W}_C[\mathbf{c}_j, \alpha_d], \psi_{i,j} = \text{SOFTMAX}(\frac{\hat{\mathbf{h}}_i \hat{\mathbf{c}}_j^\top}{\sqrt{\text{DIM}}}), \mathbf{c}_j = \sum_i \psi_{i,j} \mathbf{h}_i \quad (5)$$

whereby $\mathbf{W}_H$ and $\mathbf{W}_C$ are weight matrices, $[\cdot]$ denotes the concatenation operation, $\Psi$ the set of all mixing coefficients $\psi_{i,j}$, $\hat{\mathbf{c}}_j$ the learnable initial cluster center representation, $\hat{\mathbf{h}}_i$ the input node embeddings, and $\mathbf{c}_j$ the final cluster representation as a weighted sum of input embeddings after multiple passes. Essentially, Equation 5 is repeated $B$-times. The overall process can be viewed in Algorithm 1 in Appendix A.4. The output of these cluster centroids is fed to the decoder and serves as additional information for the decoding process. At each step, we update clusters by taking a weighted subtraction of visited nodes, given by

$$\mathbf{h}_{(c)} = W_{\text{COMBINE}}[\mathbf{h}_{\text{LAST}}^L, \mathbf{c}_1, \mathbf{c}_2, ..., \mathbf{c}_{N_c}] + \mathbf{h}_{\text{FIRST}}^L, \mathbf{c}_j' = \mathbf{c}_j - (\psi_{i,j} * \mathbf{h}_i), \forall j \in N_c \quad (6)$$

## 5 EXPERIMENTS

We mainly conform to a similar problem setup in (Liu et al., 2024; Zhou et al., 2024), using a total of 16 VRP variants with five constraints, as described in section 3. All experiments are run on a NVIDIA DGX Workstation with A100-80Gb GPUs.

**Datasets.** We utilize the following 9 country maps[1]: (1) USA13509: USA containing 13,509 cities; (2) JA9847: Japan containing 9,847 cities; (3) BM33708: Burma containing 33,708 cities; (4) KZ9976: Kazakhstan containing 9,976; (5) SW24978: Sweden containing 24,978 cities; (6) VM22775: Vietnam containing 22,775 cities; (7) EG7146: Egypt containing 7,146 cities; (8) FI10639: Finland containing 10,639 cities; (9) GR9882: Greece containing 9,882 cities.

**Task Setups.** For the MTMDVRP, we define the following: (1) *in-task* refers to tasks that the models are trained on; (2) *out-task* refers to tasks that the models are not trained on; (3) *in-distribution* refers to distributions that the models observe during training; (4) *out-distribution* refers to distributions that the models do not observe during training. For the 16 VRP variants, we denote the following 6 as in-task: CVRP, OVRP, VRPB, VRPL, VRPTW, OVRPTW, and the remaining 10 as out-task: OVRPB, OVRPL, VRPBL, VRPBTW, VRPLTW, OVRPBL, OVRPBTW, OVRPLTW, VRPBLTW, OVRPBLTW. For the distributions, the following 3 countries are defined as in-dist: USA13509, JA9847, BM33708, and the remaining 6 countries are denoted as out-dist: KZ9976, SW24978, VM22775, EG7146, FI10639, GR9882. We present all 9 full country maps to show their unique shapes in Appendix A.13. We also detail the constraint generation and feature set in Appendix A.2.

**Traditional Solvers.** We use HGS (Vidal, 2022) for CVRP and VRPTW instances, and Google's OR-tools routing solver (Furnon & Perron). For HGS, we use the default hyperparameters, while for OR-tools, we apply parallel cheapest insertion as the initial solution strategy and guided local search as the local search strategy. The timelimit is set to 20s and 40s for soving a single instance of size $N = 50, 100$, respectively. We utilize 256 CPU cores in parallel for these traditional solvers.

**Neural Constructive Solvers.** We compare the following unified solvers: (1) POMO-MTVRP which applies POMO to the MTVRP setting Liu et al. (2024); (2) MVMoE that extends POMO to include MoE layers Zhou et al. (2024); (3) MVMoE-Light, a variant of MVMoE whereby an additional hierarchical gate in the decoder makes inference and training faster Zhou et al. (2024); (4) MVMoE-Deeper whereby we increase the depth of MVMoE to have the same number of layers in the decoder as SHIELD so that both models have similar capacity; (5) SHIELD-MoD where we train our model only with MoD layers and without the clustering; (6) SHIELD, our proposed model.

**Hyperparameters.** We use the ADAM optimizer to train the neural solvers with a learning rate of $1e^{-4}$ and batch size of 128. All models are trained from scratch on $20,000$ instances per epoch for $1,000$ epochs. All models plateau at this epoch, and the relative rankings do not change with further training. At each training epoch, we uniformly sample a country from the in-distribution set, followed by a subset of points from the distribution and a problem from the in-task set. For

---

[1] https://www.math.uwaterloo.ca/tsp/world/countries.html

Table 1: Overall performance of models trained on 50 node and 100 node problems. Bold scores indicate best performing models in their respective groups. The scores and optimality gaps presented are averaged across their respective groups.

| | Model | MTMDVRP50 | | | | | | MTMDVRP100 | | | | | |
| | | In-dist | | | Out-dist | | | In-dist | | | Out-dist | | |
| | | Obj | Gap | Time | Obj | Gap | Time | Obj | Gap | Time | Obj | Gap | Time |
|---|---|---|---|---|---|---|---|---|---|---|---|---|---|
| In-task | POMO-MTVRP | 6.0778 | 3.5079% | 2.65s | 6.4261 | 3.9911% | 2.76s | 9.4123 | 4.0824% | 8.13s | 10.1147 | 5.0253% | 8.20s |
| | MVMoE | 6.0557 | 3.1479% | 3.65s | 6.3924 | 3.5071% | 3.67s | 9.3722 | 3.5969% | 10.97s | 10.0827 | 4.6855% | 11.30s |
| | MVMoE-Light | 6.0666 | 3.3595% | 3.41s | 6.4045 | 3.6860% | 3.43s | 9.3987 | 3.9088% | 10.04s | 10.1027 | 4.8979% | 10.46s |
| | MVMoE-Deeper | 6.0337 | 2.7343% | 9.03s | 6.3677 | 3.1333% | 9.03s | OOM | OOM | OOM | OOM | OOM | OOM |
| | SHIELD-MoD | 6.0220 | 2.5041% | 5.40s | 6.2933 | 2.9517% | 5.38s | 9.3453 | 2.5443% | 17.59s | 9.9800 | 3.5255% | 17.66s |
| | SHIELD | **6.0136** | **2.3747%** | 6.13s | **6.2784** | **2.7376%** | 6.11s | **9.2743** | **2.4397%** | 19.93s | **9.9501** | **3.1638%** | 20.25s |
| Out-task | POMO-MTVRP | 5.8611 | 7.6284% | 2.83s | 6.2556 | 8.0311% | 2.70s | 9.4304 | 8.1068% | 8.39s | 10.2056 | 8.8907% | 8.46s |
| | MVMoE | 5.8328 | 7.1553% | 3.81s | 6.2196 | 7.5174% | 3.73s | 9.3811 | 7.4092% | 11.13s | 10.1665 | 8.5140% | 11.44s |
| | MVMoE-Light | 5.8466 | 7.4996% | 3.46s | 6.2346 | 7.8236% | 3.50s | 9.4173 | 7.9110% | 10.27s | 10.1945 | 8.8620% | 10.75s |
| | MVMoE-Deeper | 5.8207 | 6.7924% | 9.40s | 6.2136 | 7.2962% | 9.45s | OOM | OOM | OOM | OOM | OOM | OOM |
| | SHIELD-MoD | 5.7902 | 6.2672% | 5.47s | 5.2238 | 6.6155% | 5.48s | 9.2740 | 6.0296% | 17.75s | 10.0349 | 6.9029% | 17.79s |
| | SHIELD | **5.7779** | **6.0810%** | 6.20s | **6.1570** | **6.3520%** | 6.20s | **9.2400** | **5.6104%** | 19.92s | **9.9867** | **6.2727%** | 20.18s |

Table 2: Performance of SHIELD with varying levels of sparsity on MTMDVRP50.

| | | In-dist | | Out-dist | |
| | Model | Obj | Gap | Obj | Gap |
|---|---|---|---|---|---|
| In-task | SHIELD (10%) | 6.0136 | 2.3747% | 6.2784 | 2.7376% |
| | SHIELD (20%) | 6.0055 | 2.2268% | 6.3578 | 2.8442% |
| | SHIELD (30%) | 6.0033 | 2.1948% | 6.3656 | 2.9608% |
| | SHIELD (40%) | 6.0131 | 2.3450% | 6.3718 | 3.0507% |
| | MVMoE-Deeper (100%) | 6.0337 | 2.7343% | 6.3677 | 3.1333% |
| Out-task | SHIELD (10%) | 5.7779 | 6.0810% | 6.1570 | 6.3520% |
| | SHIELD (20%) | 5.7772 | 6.0327% | 6.1671 | 6.4654% |
| | SHIELD (30%) | 5.7991 | 6.4241% | 6.1732 | 6.5603% |
| | SHIELD (40%) | 5.8068 | 6.5770% | 6.1862 | 6.7831% |
| | MVMoE-Deeper (100%) | 5.8206 | 6.7924% | 6.2136 | 7.2962% |

SHIELD, we use 3 MoD layers in the decoder and only allow 10% of tokens per layer. The number of clusters is set to $N_c = 5$, with $B = 5$ iterations of soft clustering. The encoder consists of 6 MoE layers. We provide full details of the hyperparameters in Appendix A.8.

**Performance Metrics.** We sample 1,000 test examples per problem for each country map and solve them using traditional solvers. Each sample is augmented 8 times following Kwon et al. (2020), and we report the tour length and optimality gap of the best solution found across these augmentations. The optimality gap is calculated as the percentage difference of tour length between the neural solver and the traditional solver, with smaller values indicating better performance. We provide the mathematical details of augmentation and optimality gap calculation in Appendix A.7.

## 5.1 EMPIRICAL RESULTS

Table 1 presents the average tour length (Obj) and optimality gap (Gap) across the respective tasks (in-task/out-task) and distributions (in-dist/out-dist). In summary, SHIELD clearly demonstrates significantly stronger predictive capabilities compared to other neural solvers in all scenarios. Notably, SHIELD outperforms all other neural solvers across all tasks and distributions, as evidenced by Tables 13 through 21. Essentially, we can view MVMoE-Deeper as a model that processes each token heavily with multiple layers, and MVMoE as a model that processes each token only once. SHIELD is thus a middle point between these two models that learns how to adapt the processing according to the token and problem state. Consequently, this suggests that overprocessing (MVMoE-Deeper) and underprocessing (MVMoE) nodes can serve as a problem in building an efficient foundation model. As shown, increasing the depth of the decoder to MVMoE-Deeper improves its overall performance, especially in the in-task in-distribution case. However, the autoregressive nature quickly renders the model untrainable on MTMDVRP100. Instead, if we replace these dense layers with sparse ones (as in SHIELD), we see significant improves in both task and distribution generalization.

Table 1 further highlights the positive effect of contextual clustering, especially in larger problems with 100 nodes. The benefits of clustering are most evident in the model's generalization across both tasks and distributions. It is clear that being able to summarize the larger set of points into a concise one helps the model identify keypoints in route construction.

Table 3: Ablation study for the number of clusters in SHIELD on MTMDVRP50. Keeping the number of clusters low, and thus having a sparser approach, is beneficial to the model.

|  | Model | In-dist | | Out-dist | |
|---|---|---|---|---|---|
|  |  | Obj | Gap | Obj | Gap |
| In-task | SHIELD | 6.0136 | 2.3747% | 6.2784 | 2.7376% |
|  | SHIELD ($N_c = 10$) | 6.0100 | 2.3166% | 6.3400 | 3.7522% |
|  | SHIELD ($N_c = 20$) | 6.0124 | 2.3272% | 6.3437 | 3.8127% |
| Out-task | SHIELD | 5.7779 | 6.0810% | 6.1570 | 6.3520% |
|  | SHIELD ($N_c = 10$) | 5.8019 | 6.9521% | 6.1740 | 7.0129% |
|  | SHIELD ($N_c = 20$) | 5.9824 | 11.3453% | 6.3369 | 10.8044% |

Table 4: Experimental study for the impacts of using MoD layers in the encoder on MTMDVRP50. Even by increasing the number of layers, the model's performance is unsatisfactory.

|  | Model | In-dist | | Out-dist | |
|---|---|---|---|---|---|
|  |  | Obj | Gap | Obj | Gap |
| In-task | SHIELD | 6.0136 | 2.3747% | 6.2784 | 2.7376% |
|  | SHIELD (MoDEnc-6) | 6.2271 | 6.2578% | 6.6213 | 7.6650% |
|  | SHIELD (MoDEnc-12) | 6.1838 | 5.4944% | 6.5817 | 7.1229% |
| Out-task | SHIELD | 5.7779 | 6.0810% | 6.1570 | 6.3520% |
|  | SHIELD (MoDEnc-6) | 6.0432 | 11.5021% | 6.4894 | 12.9905% |
|  | SHIELD (MoDEnc-12) | 5.9846 | 10.3009% | 6.4322 | 12.0432% |

## 5.2 ABLATION AND ANALYSES

We discuss key ablation studies here and provide more extensive ones in Appendices A.9 to A.12.

**Effect of Sparsity.** To examine the effect of sparsity, we train additional models with the capacity of the MoD layer increased to 20%, 30%, and 40%, respectively, on MTMDVRP50. The results are shown in Table 2. Specifically, as the sparsity moves from 10% to 20%, the model's bias improves—the in-task in-distribution optimality gap reduces, while the out-task in-distribution performance remains relatively stable. However, we observe that for both task types, the out-distribution performance starts to degrade. Increasing the number of tokens to 30% also improves the in-task in-distribution optimality gaps, but we see the decline in performance for out-task and out-distribution settings. This degradation continues with the 40% model, where overall performance deteriorates. The results clearly indicate that sparsity is crucial in generalization across both task and distribution.

**Effect of Clustering.** In the latent space, the soft clustering mechanism facilitates information exchange among dynamic clusters, enabling the model to capture high-level, generalizable features from neighboring hidden representations. This improves the model's understanding of the node selection process and enhances decision-making. Limiting the number of clusters also promotes abstraction, encouraging the model to focus on broadly applicable patterns rather than overfitting to task-specific details. However, too many clusters dilute this effect, leading to over-segmentation and reduced generalization as the model prioritizes more complex patterns over shared structures. Table 3 supports this, whereby we vary the number of cluster centers in the model. Thus, maintaining sparsity in this aspect is crucial as well.

**Sparse Encoder.** Given the studies so far, a natural question arises: *Since sparsity is helpful for the decoder, does it have the same impact on the encoder?* Table 4 presents our findings on this question. While preserving the same number of encoder layers and keeping a fixed capacity of 10% each layer, we find that the model's performance degrades significantly. Even after doubling the number of layers, the model fails to reach the original levels of performance. This suggests that in the encoder, it is essential for all tokens to be processed. The original MoE encoder plays a crucial role in the architecture—MoE efficiently scales and enables the model to leverage a variety of experts to capture a broad range of representations for various tasks. In contrast, the MoD introduces greater flexibility in the decoder, giving the model the ability to dynamically select layers for decision-making, which helps it adapt effectively to varying outputs.

**Patterns of Layer Selection.** We investigate how SHIELD behaves for a given problem compared to MVMoE. Figure 3 shows the final output of SHIELD and MVMoE for OVRPBTW on VM22775. The starred points indicate that SHIELD routes them more frequently during the problem-solving process. Consider route R5 for SHIELD and route R8 for MVMoE. SHIELD can recognize that such points are far away and that it is more advantageous to visit other points en route, whereas MVMoE merely visited one node first. Likewise, for route R4 in SHIELD and route R6 in MVMoE, SHIELD identifies the 2 starred points to be better served as connecting points, as opposed to making an entire loop, which results in back-tracking to a similar area. Since the problem is an open problem, we can see that SHIELD favors ending routes at faraway locations, whereas MVMoE tends to loop back and forth in many occurrences.

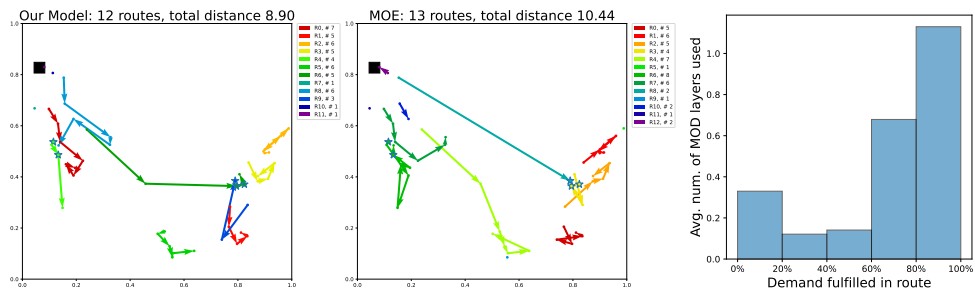

Figure 3: *Left two panels:* Plot of routes for OVRPBTW task between SHIELD (left) and MVMoE (middle). Points denoted with a star are the top few points that SHIELD identified and passed these embeddings through more layers. Note that the initial routes from the depot are masked away for a better view. *Right panel:* Average number of layers used as the demand is being met for CVRP.

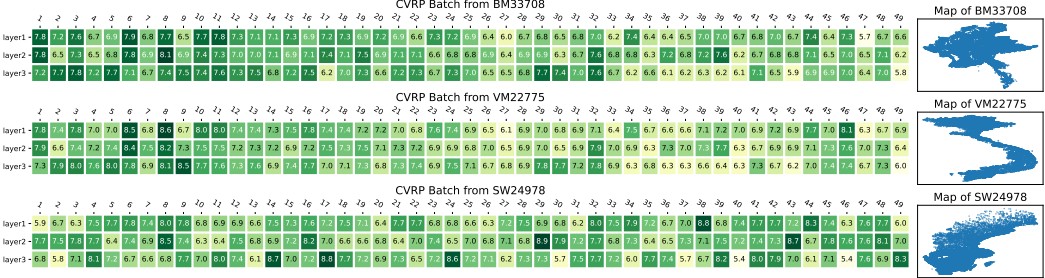

Figure 4: Plot of layer usage for CVRP samples across three maps, with the $x$-axis as node IDs, $y$-axis as layer numbers, and values as average usage frequency during decoding.

We conduct further analysis on the simpler CVRP to examine how the model generalizes across tasks and distributions. Figure 4 presents a heat map where we average the number of times a layer is used when the agent is positioned on a node. Note that the $x$-axis denotes the node ID, while the $y$-axis denotes the layer number, with the value indicating the average number of times that combination is called. For this analysis, we sort the nodes in anticlockwise order based on their $x$ and $y$ coordinates to impose a spatial ordering. We observe that for maps with similar top density and curved shapes, such as BM33708 and VM22775, the MoD layers tend to exhibit a similar pattern in layer usage, whereas a map like SW24978 has a much different sort of distribution.

Furthermore, the right panel of Figure 3 illustrates how the use of layers is distributed as the agent starts to address the demands of the problem. The $x$-axis represents the percentage of the sub-tour solved, while the $y$-axis denotes the average number of MoD layers being used by the agent. Thus, the plot indicates how the network is being used as the route is formed. As shown, when the sequence is still fairly early, the model uses some processing power to find a good set of initial nodes. In the middle, fewer layers are being used, and finally, as the problem comes to a close, more layers are activated to finalize the selection of appropriate ending points.

## 6 CONCLUSION

The push toward unified generic solvers is an important step in building foundation models for neural combinatorial optimization. In this paper, we propose to extend such solvers to the Multi-Task Multi-Distribution VRP, a significantly more practical representation of industrial problems. With this problem setting, we further propose SHIELD, a neural architecture that is designed to handle generalization across both task and distribution dimensions, making it a powerful solver for practical problems. Extensive experiments and thorough analysis of the empirical results demonstrate that *sparsity* and *hierarchy*, two key techniques in SHIELD, substantially influence the generalization ability of the model. We believe that this forms a stepping stone towards other forms of foundation models, such as generalizing across various sizes.

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

# A APPENDIX

## A.1 ADDITIONAL RELATED WORK

**Single-task VRP Solver.** Most research focuses on developing single-task VRP solvers, which primarily follows two key paradigms: constructive solvers and improvement solvers. *Constructive solvers* learn policies that generate solutions from scratch in an end-to-end fashion. Early works proposed Pointer Networks (Vinyals et al., 2015) to approximate optimal solutions for the TSP (Bello et al., 2017) and CVRP (Nazari et al., 2018) in an autoregressive (AR) way. A major breakthrough in AR-based methods came with the Attention Model (AM) (Kool et al., 2018), which became a foundational approach for solving VRPs. The policy optimization with multiple optima (POMO) (Kwon et al., 2020) improved upon AM by considering the symmetry property of VRP solutions. More recently, a wave of studies has focused on further boosting either the performance (Kim et al., 2022; Drakulic et al., 2023; Chalumeau et al., 2023; Grinsztajn et al., 2023; Luo et al., 2023; Hottung et al., 2024) or versatility (Kwon et al., 2021; Berto et al., 2023) of these solvers to handle more complex and varied problem instances. We refer the reader to Appendix A.1 for details on non-autoregressive (NAR) constructive solvers and improvement solvers in the single-task VRP. Beyond AR methods, non-autoregressive (NAR) constructive approaches (Joshi et al., 2019; Fu et al., 2021; Kool et al., 2022; Qiu et al., 2022; Sun & Yang, 2023; Min et al., 2023; Ye et al., 2023; Kim et al., 2024; Xia et al., 2024) construct matrices, such as heatmaps representing the probability of each edge being part of the optimal solution, to solve VRPs through complex post-hoc search. In contrast, *improvement solvers* (Chen & Tian, 2019; Lu et al., 2020; Hottung & Tierney, 2020; Costa et al., 2020; Wu et al., 2021; Ma et al., 2021; Xin et al., 2021; Hudson et al., 2022; Ma et al., 2023) typically learn more efficient and effective search components, often within the framework of classic heuristics or meta-heuristics, to iteratively refine an initial feasible solution. While constructive solvers can efficiently achieve desirable performance, improvement solvers have the potential to find near-optimal solutions given a longer time budget. There are also studies that focus on the scalability (Li et al., 2021; Hou et al., 2023; Ye et al., 2024) and robustness (Geisler et al., 2022; Lu et al., 2023) of neural VRP solvers, which are less directly related to our work. For those interested, we refer readers to Bogyrbayeva et al. (2024).

## A.2 GENERATION OF VRP VARIANTS

As mentioned in Section 3, we consider four additional constraints on top of the CVRP, resulting in 16 different variants in total. Note that unlike (Liu et al., 2024; Zhou et al., 2024), we do not generate node coordinates from a uniform distribution. Instead, we sample a set of fixed points from a given map. Here, we detail the generation of the five total constraints.

**Capacity (C):** We adopt the settings from (Kool et al., 2018), whereby each node's demand $\delta_i$ is randomly sampled from a discrete distribution set, $\{1, 2, ..., 9\}$. For $N = 50$, the vehicle capacity $Q$ is set to 40, and for $N = 100$, the vehicle capacity is set to 50. All demands are first normalized to their vehicle capacities, so that $\delta'_i = \delta_i/Q$.

**Open route (O):** For open routes, we set $o_t = 1$ in the dynamic feature set received by the decoder. Apart from this, we remove the constraint that the vehicle has to return to the depot when it has completed the route or is unable to proceed further due to other constraints. Suppose the problem has both open routes (O) and duration limit (L), then we mask all nodes $v_j$ such that $l_t + d_{ij} > L$, whereby $d_{ij}$ is the distance between node $v_i$ and the potentially masked node $v_j$, and $L$ is the duration limit constraint. For problems with both open routes (O) and time windows (TW), we mask all nodes $v_j$ such that $t_t + d_{ij} > w_j^c$, where $t_t$ is the current time after servicing the current node. Finally, suppose a route has both open routes (O) and backhauls (B), no special masking considerations are required as the vehicle does not return to the origin.

**Backhaul (B):** We adopt the approach from (Liu et al., 2024) by randomly selecting 20% of customer nodes to be backhauls, thus changing their demand to be negative instead. We also follow the same setup as (Zhou et al., 2024) whereby routes can have a mix of linehauls and backhauls without any strict precedence. To ensure feasible solutions, we ensure that all starting points are linehauls only unless all remaining nodes are backhauls.

**Duration limit (L):** The duration limit is fixed such that the maximum length of the vehicle, $L = 3$, which ensures that a feasible route can be found as all points are normalized to a unit square.

**Time window (TW):** For time windows, we follow the methodology in (Li et al., 2021). The depot node $v_0$ has a time window of $[0, 3]$ with no service time. As for other nodes, each node has a service time of $s_i = 0.2$, and the time windows are obtained as following: (1) first we sample a time window center given by $\gamma_i \ U(w_0^o + d_{0i}, w_i^c - d_{i0} - s_i)$, whereby $d_{0i} = d_{i0}$ is the distance or travel time between depot $v_0$ and node $v_i$, (2) then we sample a time window half-width $h_i$ uniformly from $[s_i/2, w_0^c/3] = [0.1, 1]$, (3) then we set the time window as $[w_i^o, w_i^c] = [\text{MAX}(w_i^o, \gamma_i - h_i), \text{MIN}(w_i^c, \gamma_i + h_i)]$.

## A.3 NEURAL COMBINATORIAL OPTIMIZATION MODEL DETAILS

Neural constructive solvers are typically parameterized by a neural network, whereby a policy, $\pi_\theta$, is trained by reinforcement learning so as to construct a solution sequentially (Kool et al., 2018; Kwon et al., 2020). The attention-based mechanism (Vaswani, 2017) is popularly used, whereby attention scores govern the decision-making process in an autoregressive fashion. The overall feasibility of solution can be managed by the use of masking, whereby invalid moves are masked away during the construction process. Classically, neural constructive solvers employ an encoder-decoder architecture and are trained as sequence-to-sequence models (Sutskever, 2014). The probability of a sequence can be factorized using the chain-rule of probability, such that

$$p_\theta(\tau|\mathcal{G}) = \prod_{t=1}^{T} p_\theta(\tau_t|\mathcal{G}, \tau_{1:t-1}) \tag{7}$$

The encoder tends employ a typical transformer layer, whereby

$$\tilde{\mathbf{h}} = \text{LN}^l(\mathbf{h}_i^{l-1} + \text{MHA}_i^l(\mathbf{h}_i^{l-1}, ..., \mathbf{h}_N^{l-1})) \tag{8}$$

$$\mathbf{h}_i^l = \text{LN}^l(\tilde{\mathbf{h}}_i + \text{FF}(\tilde{\mathbf{h}}_i)) \tag{9}$$

where $h_i^l$ is the embedding of the $i$-th node at the $l$-th layer, MHA is the multi-headed attention layer, LN the layer normalization function, and FF a feed-forward multi-layer perceptron (MLP). All embeddings are passed through $L$ layers before reaching the decoder.

The decoder produces the solutions autoregressively, whereby a contextual embedding combines the embeddings from the starting and current location as follows

$$\mathbf{h}_{(c)} = \mathbf{h}_{\text{LAST}}^L + \mathbf{h}_{\text{START}}^L \tag{10}$$

Then, the attention mechanism is used to produce the attention scores. Notably, the context vectors $\mathbf{h}_{(c)}$ are denoted as query vectors, while keys and values are the set of $N$ node embeddings. This is mathematically represented as

$$a_j = \begin{cases} U \cdot \text{TANH}(\frac{\mathbf{Q}\mathbf{K}^\top}{\sqrt{\text{DIM}}}) & j \neq \tau_{t'}, \forall t' < t \\ -\infty & \text{otherwise} \end{cases} \tag{11}$$

whereby $U$ is a clipping function and DIM the dimension of the latent vector. These attention scores are then normalized using a softmax function to generate the following selection probability

$$p_i = p_\theta(\tau_t = i|s, \tau_{1:t-1}) = \frac{e^{a_j}}{\sum_j e^{a_j}} \tag{12}$$

Finally, given a baseline function $b(\cdot)$, the policy is trained with the REINFORCE algorithm (Williams, 1992) and gradient ascent, with the expected return $J$

$$\nabla_\theta J(\theta) \approx \mathbb{E}\Big[(R(\tau^i) - b^i(s))\nabla_\theta \log p_\theta(\tau^i|s)\Big] \tag{13}$$

The reward of each solution $R$ is the length of the solution tour.

## A.4 SOFT-CLUSTERING ALGORITHM DETAILS

---

**Algorithm 1** Psuedo code of soft clustering algorithm

---

1: **procedure** CLUSTER(encoder embeddings $H$, constraints vector $\gamma_k$, number of centers $N_c$, number of iterations $B$, initial embeddings $C$, embedding size $d$)
2:      $\alpha_d = \mathbf{W}_\theta^\top \gamma_k$
3:      **for** $b \leftarrow 1$ to $B$ **do**
4:          $\hat{H} \leftarrow W_H(H)$
5:          $\hat{C} \leftarrow W_C([C, \alpha_d])$
6:          $\psi = \text{SOFTMAX}(\frac{\hat{H}\hat{C}^\top}{\sqrt{d}})$          ▷ Compute attention scores
7:          $C = \sum_i \psi_i h_i$          ▷ Update the centers with data
8:          $C_{\text{OUT}} = \hat{C} + C$          ▷ Residual connection
9:          $C = \text{NORM}(C_{\text{OUT}})$          ▷ Layer normalization
10:     **end for**
11:     **return** $C$
12: **end procedure**

---

## A.5 MODEL SIZES AND AVERAGE RUNTIMES

Table 5: Overall number of parameters and average runtimes for all models.

| Model | Num. Parameters | Runtime on MTMDVRP50 | Runtime on MTMDVRP100 |
|---|---|---|---|
| POMO-MTVRP | 1.25M | 2.74s | 8.30s |
| MVMoE | 3.68M | 3.72s | 11.21s |
| MVMoE-Light | 3.70M | 3.45s | 10.38s |
| MVMoE-Deeper | 4.46M | 9.23s | OOM |
| SHIELD-MoD | 4.37M | 5.43s | 17.70s |
| SHIELD | 4.59M | 6.16s | 20.07s |

## A.6  MATHEMATICAL NOTATIONS

| | |
|---|---|
| $\mathcal{S}_i$ | A problem instance $i$ |
| $\mathcal{D}_t$ | Set of dynamic features at decoding time-step $t$ |
| $t$ | Decoding time-step |
| $x_i$ | $x$-coordinate of problem instance $i$ |
| $y_i$ | $y$-coordinate of problem instance $i$ |
| $\delta_i$ | Demand of node $i$ |
| $w_i^o$ | Opening timing of time-window for node $i$ |
| $w_i^c$ | Closing timing of time-window for node $i$ |
| $z_t$ | Capacity of vehicle at decoding time-step $t$ |
| $t_t$ | Current time-step |
| $o_t$ | Presence of open route at time-step $t$ |
| $l_t$ | Current length of partial route at time-step $t$ |
| $\mathcal{K}$ | Set of all possible VRP tasks |
| $\mathcal{Q}$ | Set of all possible distributions |
| $\beta$ | The percentage of tokens allowed through a MoD layer |
| $r_i$ | Router score for node $i$ |
| $\gamma_k$ | One-hot encoded vector of constraints for task $k$ |
| $o_t$ | Presence of open route at time-step $t$ |
| $B$ | Number of iterations of clustering |
| $N_c$ | Number of cluster centers |
| $\psi_{ij}$ | Mixing coefficient between node $i$ and cluster $j$ |

## A.7  METRIC DETAILS

We utilize 8x augmentations on the $(x, y)$-coordinates for the test set as proposed by (Kwon et al., 2020). The following table details the various transformations applied.

Table 6: List of augmentations suggested by Kwon et al. (2020)

| $f(x, y)$ | |
|---|---|
| $(x, y)$ | $(y, x)$ |
| $(x, 1 - y)$ | $(y, 1 - x)$ |
| $(1 - x, y)$ | $(1 - y, x)$ |
| $(1 - x, 1 - y)$ | $(1 - y, 1 - x)$ |

The optimality gap is measured as the percentage gap between the neural solver's tour length and the traditional solver. This is defined as

$$O = \left( \frac{\frac{1}{N} \sum_i^N R_i}{\frac{1}{N} \sum_i^N L_i} - 1 \right) * 100 \tag{14}$$

where $L_i$ is the tour length of test instance $i$ computed by the traditional solver, HGS or OR-Tools.

## A.8 Detailed hyperparameter and training settings

- Number of MoE encoder layers: 6
- Total number of experts: 4
- Number of experts used per layer: 2
- Number of MoD decoder layers: 3
- Capacity of MoD layer (number of tokens allowed): 10%
- Number of single-headed attention decision-making layer: 1
- Latent dimension size: 128
- Number of heads per transformer layer: 8
- Feedforward MLP size: 512
- Logit clipping $U$: 10
- Learning rate: 1e-4
- Number of clustering layers: 1
- Number of iterations for clustering: 5
- Number of learnable cluster embeddings: 5
- Number of episodes per epoch: 20,000
- Number of epochs: 1,000
- Batch size: 128

## A.9 ADDITIONAL EXPERIMENTS - GENERALIZATION TO CVRPLIB

Table 7: Performance on CVRPLib data Set-X-1. Instances vary from 101 to 251 nodes.

| Set-X-1 | | POMO-MTL | | MVMoE | | MVMoE-Light | | SHIELD-MoD | | SHIELD | | SHIELD-Ep400 | |
|---|---|---|---|---|---|---|---|---|---|---|---|---|---|
| Instance | Opt. | Obj. | Gap | Obj. | Gap | Obj. | Gap | Obj. | Gap | Obj. | Gap | Obj. | Gap |
| X-n101-k25 | 27591 | 29875 | 8.2781% | 29189 | 5.7917% | 29445 | 6.7196% | 28967 | 4.9871% | 28678 | 3.9397% | 29346 | 6.3608% |
| X-n106-k14 | 26362 | 27158 | 3.0195% | 27061 | 2.6515% | 27356 | 3.7706% | 26909 | 2.0750% | 27076 | 2.7084% | 27192 | 3.1485% |
| X-n110-k13 | 14971 | 15420 | 2.9991% | 15379 | 2.7253% | 15387 | 2.7787% | 15450 | 3.1995% | 15316 | 2.3045% | 15312 | 2.2777% |
| X-n115-k10 | 12747 | 13680 | 7.3194% | 13368 | 4.8717% | 13536 | 6.1897% | 13245 | 3.9068% | 13290 | 4.2598% | 13472 | 5.6876% |
| X-n120-k6 | 13332 | 13939 | 4.5530% | 14082 | 5.6256% | 13980 | 4.8605% | 13901 | 4.2679% | 13724 | 2.9403% | 13971 | 4.7930% |
| X-n125-k30 | 55539 | 58929 | 6.1038% | 58443 | 5.2288% | 59056 | 6.3325% | 58648 | 5.5979% | 57426 | 3.3976% | 58277 | 4.9299% |
| X-n129-k18 | 28940 | 30114 | 4.0567% | 29905 | 3.3345% | 29970 | 3.5591% | 29802 | 2.9786% | 29540 | 2.0733% | 29695 | 2.6088% |
| X-n134-k13 | 10916 | 11637 | 6.6050% | 11658 | 6.7974% | 11612 | 6.3760% | 11519 | 5.5240% | 11274 | 3.2796% | 11447 | 4.8644% |
| X-n139-k10 | 13590 | 14295 | 5.1876% | 14155 | 4.1575% | 14121 | 3.9073% | 13988 | 2.9286% | 14004 | 3.0464% | 14152 | 4.1354% |
| X-n143-k7 | 15700 | 17091 | 8.8599% | 16710 | 6.4331% | 16744 | 6.6497% | 16621 | 5.8662% | 16548 | 5.4013% | 16792 | 6.9554% |
| X-n148-k46 | 43448 | 47317 | 8.9049% | 45621 | 5.0014% | 45794 | 5.3996% | 45728 | 5.2477% | 44739 | 2.9714% | 45082 | 3.7608% |
| X-n153-k22 | 21220 | 23689 | 11.6352% | 23267 | 9.6466% | 23510 | 10.7917% | 23541 | 10.9378% | 23252 | 9.5759% | 23392 | 10.2356% |
| X-n157-k13 | 16876 | 17730 | 5.0604% | 17698 | 4.8708% | 17713 | 4.9597% | 17386 | 3.0220% | 17366 | 2.9035% | 17583 | 4.1894% |
| X-n162-k11 | 14138 | 14845 | 5.0007% | 14884 | 5.2766% | 14746 | 4.3005% | 14703 | 3.9963% | 14767 | 4.4490% | 14804 | 4.7107% |
| X-n167-k10 | 20557 | 21863 | 6.3531% | 21898 | 6.5233% | 21827 | 6.1779% | 21644 | 5.2877% | 21326 | 3.7408% | 21566 | 4.9083% |
| X-n172-k51 | 45607 | 50381 | 10.4677% | 48863 | 7.1393% | 48686 | 6.7512% | 48434 | 6.1986% | 48091 | 5.4465% | 48613 | 6.5911% |
| X-n176-k26 | 47812 | 53848 | 12.6244% | 52302 | 9.3909% | 51433 | 7.5734% | 52313 | 9.4140% | 51811 | 8.3640% | 50887 | 6.4314% |
| X-n181-k23 | 25569 | 26480 | 3.5629% | 26661 | 4.2708% | 26490 | 3.6020% | 26156 | 2.2957% | 26237 | 2.6125% | 26333 | 2.9880% |
| X-n186-k15 | 24145 | 25900 | 7.2686% | 25695 | 6.4195% | 25613 | 6.0799% | 25409 | 5.2350% | 25503 | 5.6244% | 25372 | 5.0818% |
| X-n190-k8 | 16980 | 17826 | 4.9823% | 18121 | 6.7197% | 18125 | 6.7432% | 17417 | 2.5736% | 17802 | 4.8410% | 17846 | 5.1001% |
| X-n195-k51 | 44225 | 49703 | 12.3867% | 47834 | 8.1605% | 47704 | 7.8666% | 47608 | 7.6495% | 46509 | 5.1645% | 47731 | 7.9276% |
| X-n200-k36 | 58578 | 61857 | 5.5977% | 62039 | 5.9084% | 61871 | 5.6216% | 61384 | 4.7902% | 61375 | 4.7748% | 61729 | 5.3792% |
| X-n209-k16 | 30656 | 32754 | 6.8437% | 32725 | 6.7491% | 32605 | 6.7491% | 32157 | 4.8963% | 32244 | 5.1801% | 32083 | 4.6549% |
| X-n219-k73 | 117595 | 120795 | 2.7212% | 119924 | 1.9805% | 121201 | 3.0665% | 119679 | 1.7722% | 119847 | 1.9150% | 119560 | 1.6710% |
| X-n228-k23 | 25742 | 30042 | 16.7042% | 28629 | 11.2151% | 28754 | 11.7007% | 28206 | 9.5719% | 28118 | 9.2301% | 28119 | 9.2339% |
| X-n237-k14 | 27042 | 29217 | 8.0430% | 29252 | 8.1725% | 29003 | 7.2517% | 28560 | 5.6135% | 28743 | 6.2902% | 28880 | 6.7968% |
| X-n247-k50 | 37274 | 43111 | 15.6597% | 40868 | 9.6421% | 41735 | 11.9681% | 41556 | 11.4879% | 40676 | 9.1270% | 41266 | 10.7099% |
| X-n251-k28 | 38684 | 41321 | 6.8168% | 40874 | 5.6613% | 40854 | 5.6096% | 40316 | 4.2188% | 40410 | 4.4618% | 40602 | 4.9581% |
| Averages | 31280 | 33601 | 7.4148% | 33111 | 6.0845% | 33174 | 6.1773% | 32902 | 5.1979% | 32703 | 4.6437% | 32897 | 5.3961% |

Table 8: Performance on CVRPLib data Set-X-2. Instances vary from 502 to 1001 nodes.

| Set-X-2 | | POMO-MTL | | MVMoE | | MVMoE-Light | | SHIELD-MoD | | SHIELD | | SHIELD-Ep400 | |
|---|---|---|---|---|---|---|---|---|---|---|---|---|---|
| Instance | Opt. | Obj. | Gap | Obj. | Gap | Obj. | Gap | Obj. | Gap | Obj. | Gap | Obj. | Gap |
| X-n502-k39 | 69226 | 73599 | 6.3170% | 75113 | 8.5040% | 75679 | 9.3216% | 73184 | 5.7175% | 73062 | 5.5413% | 73445 | 6.0945% |
| X-n513-k21 | 24201 | 27955 | 15.5118% | 29444 | 21.6644% | 28483 | 17.6935% | 27478 | 13.5408% | 27217 | 12.4623% | 27373 | 13.1069% |
| X-n524-k153 | 154593 | 175923 | 13.7975% | 174409 | 12.8182% | 170334 | 10.1822% | 167380 | 8.2714% | 169715 | 9.7818% | 166660 | 7.8057% |
| X-n536-k96 | 94846 | 104866 | 10.5645% | 105896 | 11.6505% | 104408 | 10.0816% | 102157 | 7.7083% | 102237 | 7.7926% | 103042 | 8.6414% |
| X-n548-k50 | 86700 | 94290 | 8.7543% | 93623 | 7.9850% | 92798 | 7.0334% | 91483 | 5.5167% | 91726 | 5.7970% | 92055 | 6.1765% |
| X-n561-k42 | 42717 | 48781 | 14.1958% | 49953 | 16.9394% | 48678 | 13.9546% | 47328 | 10.7943% | 47639 | 11.5223% | 47485 | 11.1618% |
| X-n573-k30 | 50673 | 57151 | 12.7839% | 55796 | 10.1099% | 55870 | 10.2560% | 54664 | 7.8760% | 53936 | 6.4393% | 55204 | 8.9416% |
| X-n586-k159 | 190316 | 208217 | 9.4059% | 209038 | 9.8373% | 208510 | 9.5599% | 205408 | 7.9300% | 205487 | 7.9715% | 208175 | 9.3839% |
| X-n599-k92 | 108451 | 118994 | 9.7214% | 119879 | 10.5375% | 118864 | 9.6016% | 117615 | 8.4499% | 116950 | 7.8367% | 118514 | 9.2788% |
| X-n613-k62 | 59535 | 68882 | 15.7000% | 72992 | 22.6035% | 69091 | 16.0511% | 66657 | 11.9627% | 66715 | 12.0601% | 66419 | 11.5629% |
| X-n627-k43 | 62164 | 69756 | 12.2129% | 69197 | 11.3136% | 68302 | 9.8739% | 67125 | 7.9805% | 67494 | 8.5741% | 67059 | 7.8743% |
| X-n641-k35 | 63682 | 72638 | 14.0636% | 72348 | 13.6082% | 71041 | 11.5559% | 69425 | 9.0182% | 69156 | 8.5958% | 69617 | 9.3197% |
| X-n655-k131 | 106780 | 115083 | 7.7758% | 113186 | 5.9993% | 113610 | 6.3963% | 111711 | 4.6179% | 110508 | 3.4913% | 111542 | 4.4596% |
| X-n670-k130 | 146332 | 177344 | 21.1929% | 173046 | 18.2557% | 170328 | 16.3983% | 164820 | 12.6343% | 166737 | 13.9443% | 164140 | 12.1696% |
| X-n685-k75 | 68205 | 79362 | 16.3580% | 84485 | 23.8692% | 79502 | 16.5633% | 76224 | 11.7572% | 76676 | 12.4199% | 76195 | 11.7147% |
| X-n701-k44 | 81923 | 90163 | 10.0582% | 92522 | 12.9378% | 89812 | 9.6298% | 88608 | 8.1601% | 87959 | 7.3679% | 88603 | 8.1540% |
| X-n716-k35 | 43373 | 50636 | 16.7454% | 51003 | 17.5916% | 49429 | 13.9626% | 47821 | 10.2552% | 47996 | 10.6587% | 47586 | 9.7134% |
| X-n733-k159 | 136187 | 158694 | 16.5265% | 156545 | 14.9486% | 156747 | 15.0969% | 148203 | 8.8232% | 149217 | 9.5677% | 153664 | 12.8331% |
| X-n749-k98 | 77269 | 88333 | 14.3188% | 91569 | 18.5068% | 88438 | 14.4547% | 84651 | 9.5536% | 85367 | 10.4803% | 85824 | 11.0717% |
| X-n766-k71 | 114417 | 135772 | 18.6642% | 133725 | 16.8751% | 129996 | 13.6160% | 128128 | 11.9834% | 128052 | 11.9169% | 127179 | 11.1539% |
| X-n783-k48 | 72386 | 84162 | 16.2683% | 85094 | 17.5559% | 82690 | 14.2348% | 80855 | 11.6998% | 80521 | 11.2384% | 80358 | 11.0132% |
| X-n801-k40 | 73305 | 85008 | 15.9648% | 84025 | 14.6238% | 83210 | 13.5120% | 81070 | 10.5927% | 80637 | 10.0020% | 81015 | 10.5177% |
| X-n819-k171 | 158121 | 177282 | 12.1179% | 178589 | 12.9445% | 175340 | 10.8898% | 171630 | 8.5435% | 172020 | 8.7901% | 175820 | 11.1933% |
| X-n837-k142 | 193737 | 213908 | 10.4115% | 214165 | 10.5442% | 211521 | 9.1795% | 208552 | 7.6470% | 209350 | 8.0589% | 210464 | 8.6339% |
| X-n856-k95 | 88965 | 99911 | 12.3037% | 102485 | 15.1970% | 98990 | 11.2685% | 99014 | 11.2955% | 96889 | 8.9069% | 97602 | 9.7083% |
| X-n876-k59 | 99299 | 110191 | 10.9689% | 111857 | 12.6467% | 111044 | 11.8279% | 106826 | 7.5801% | 106180 | 6.9296% | 107710 | 8.4704% |
| X-n895-k37 | 53860 | 65277 | 21.1975% | 66353 | 23.1953% | 64716 | 20.1569% | 62114 | 15.3243% | 62101 | 15.3008% | 61552 | 14.2815% |
| X-n916-k207 | 329179 | 360052 | 9.3788% | 362596 | 10.1516% | 359444 | 9.1941% | 354793 | 7.7812% | 353567 | 7.4087% | 355423 | 7.9726% |
| X-n936-k151 | 132715 | 173297 | 30.5783% | 167723 | 26.3783% | 163193 | 22.9650% | 158308 | 19.2842% | 159965 | 20.5327% | 156897 | 18.2210% |
| X-n957-k87 | 85465 | 98132 | 14.8213% | 99442 | 16.3541% | 97109 | 13.6243% | 94209 | 10.2311% | 93672 | 9.6028% | 94118 | 10.1246% |
| X-n979-k58 | 118976 | 132128 | 11.0543% | 132449 | 11.3241% | 131752 | 10.7383% | 128765 | 8.2277% | 129968 | 9.2388% | 127952 | 7.5444% |
| X-n1001-k43 | 72355 | 87428 | 20.8320% | 87802 | 21.3489% | 86285 | 19.2523% | 82866 | 14.5270% | 82407 | 13.8926% | 82253 | 13.6798% |
| Averages | 101874 | 115725 | 14.0802% | 116136 | 14.9631% | 114225 | 12.7539% | 111534 | 9.8527% | 111598 | 9.8164% | 111905 | 10.0618% |

Tables 7 and 8 showcase various models applied to data from the CVRPLib Set-X-1 and Set-X-2. These instances have varying sizes from 101 to 1001 nodes. Additionally, we include SHIELD-Ep400, the 400th epoch of training SHIELD, which has similar in-task in-dist performance compared to MVMoE. Evidently, SHIELD is a significantly superior model in terms of size generalization.

## A.10 ADDITIONAL EXPERIMENTS - GENERALIZATION OF SHIELD

Table 9: Performance of SHIELD-Ep400, the 400th epoch of SHIELD, to MVMoE. Both models have similar in-task in-dist performance and can be viewed as equivalents.

| | Model | MTMDVRP50 | | | | MTMDVRP100 | | | |
| | | In-dist | | Out-dist | | In-dist | | Out-dist | |
| | | Obj | Gap | Obj | Gap | Obj | Gap | Obj | Gap |
|---|---|---|---|---|---|---|---|---|---|
| In-task | MVMoE | 6.0557 | 3.1479% | 6.3924 | 3.5071% | 9.3722 | 3.5969% | 10.0827 | 4.6855% |
| | SHIELD-400Ep | 6.0597 | 3.1495% | 6.3830 | 3.2730% | **9.3785** | **3.5993%** | **10.0559** | **4.3562%** |
| Out-task | MVMoE | 5.8328 | 7.1553% | 6.2196 | 7.5174% | 9.3811 | 7.4092% | 10.1665 | 8.5140% |
| | SHIELD-400Ep | **5.8290** | **7.1064%** | **6.2085** | **7.2927%** | **9.3499** | **6.9578%** | **10.1202** | **7.8332%** |

Table 10: Performance of SHIELD-Ep600, the 600th epoch of SHIELD, to MVMoE-Deeper. Both models have similar in-task in-dist performance and can be viewed as equivalents.

| | Model | MTMDVRP50 | | | | MTMDVRP100 | | | |
| | | In-dist | | Out-dist | | In-dist | | Out-dist | |
| | | Obj | Gap | Obj | Gap | Obj | Gap | Obj | Gap |
|---|---|---|---|---|---|---|---|---|---|
| In-task | MVMoE-Deeper | 6.0337 | 2.7343% | 6.3677 | 3.1333% | OOM | OOM | OOM | OOM |
| | SHIELD-600Ep | 6.0333 | 2.7089% | 6.3653 | 2.9993% | **9.3194** | **2.9498%** | **10.0113194** | **3.8262%** |
| Out-task | MVMoE-Deeper | 5.8206 | 6.7924% | 6.2136 | 7.2962% | OOM | OOM | OOM | OOM |
| | SHIELD-600Ep | **5.8039** | **6.6539%** | **6.1823** | **6.8736%** | **9.3105** | **6.4308%** | **10.0764533** | **7.2549%** |

Table 9 and 10 showcase SHIELD at the 400-th and 600-th epoch. These models have similar in-task and in-dist performance compared to MVMoE and MVMoE-Deeper, respectively, and can be viewed as equivalent models. Comparatively, SHIELD has better generalization across tasks and distribution, suggesting that the architecture is superior.

## A.11 ADDITIONAL EXPERIMENTS - IMPORTANCE OF VARIED DISTRIBUTIONS

Table 11: Performance of all models when trained on only Uniform data. We retain a similar layout to Table 1 but all distributions are considered out-of-distribution in this case.

| | Model | MTMDVRP50 | | | |
| | | In-dist | | Out-dist | |
| | | Obj | Gap | Obj | Gap |
|---|---|---|---|---|---|
| In-task | POMO-MTVRP (Uniform) | 6.0932 | 3.8834% | 6.4104 | 4.0007% |
| | MVMoE (Uniform) | 6.0779 | 3.6000% | 6.3930 | 3.6710% |
| | MVMoE-Light (Uniform) | 6.0926 | 3.8418% | 6.4061 | 3.8254% |
| | MVMoE-Deeper (Uniform) | 6.0580 | 3.1964% | 6.3822 | 3.5062% |
| | SHIELD-MoD (Uniform) | 6.0482 | 3.0379% | 6.3666 | 3.2037% |
| | SHIELD (Uniform) | **6.0414** | **2.9223%** | **6.3596** | **3.0832%** |
| Out-task | POMO-MTVRP (Uniform) | 5.8762 | 8.1526% | 6.2457 | 8.3681% |
| | MVMoE (Uniform) | 5.8602 | 7.7505% | 6.2251 | 7.8788% |
| | MVMoE-Light (Uniform) | 5.8802 | 8.1328% | 6.2414 | 8.0983% |
| | MVMoE-Deeper (Uniform) | 5.8292 | 7.0524% | 6.2034 | 7.4642% |
| | SHIELD-MoD (Uniform) | 5.8103 | 6.7257% | 6.1769 | 6.9455% |
| | SHIELD (Uniform) | **5.8035** | **6.6394%** | **6.1712** | **6.8616%** |

Table 11 displays the performance of all models when trained purely on uniform data. Note that while we retain the same table layout as Table 1, all distributions are considered as out-of-distribution in such a case as the model does not see them at all. Evidently, all models degrade in their predictive performance, even though SHIELD still retains its overall superior performance.

## A.12  ADDITIONAL EXPERIMENTS - SINGLE-TASK MULTI-DISTRIBUTION

Table 12: Performance of various models trained on the CVRP task with multiple distributions.

| Model | CVRP50 | | | |
| | In-dist | | Out-dist | |
| | Obj | Gap | Obj | Gap |
|---|---|---|---|---|
| POMO-MTVRP | 6.6511 | 1.2260% | 6.9763 | 1.4689% |
| MVMoE | 6.6454 | 1.1401% | 6.9709 | 1.3858% |
| MVMoE-Light | 6.6482 | 1.1814% | 6.9723 | 1.4112% |
| MVMoE-Deeper | 6.6313 | 0.9207% | 6.9628 | 1.2731% |
| SHIELD-MoD | 6.6284 | 0.8798% | 6.9552 | 1.1623% |
| SHIELD | **6.6269** | **0.8570%** | **6.9474** | **1.0338%** |

Table 12 displays the performance of various models when trained in a single-task multi-distribution setting. Here, we choose CVRP to be the task at hand. SHIELD remains the best-performing model in such a scenario, suggesting that its architecture is not catered purely to a multi-task multi-distribution problem only.

## A.13 DETAILED EXPERIMENTAL RESULTS

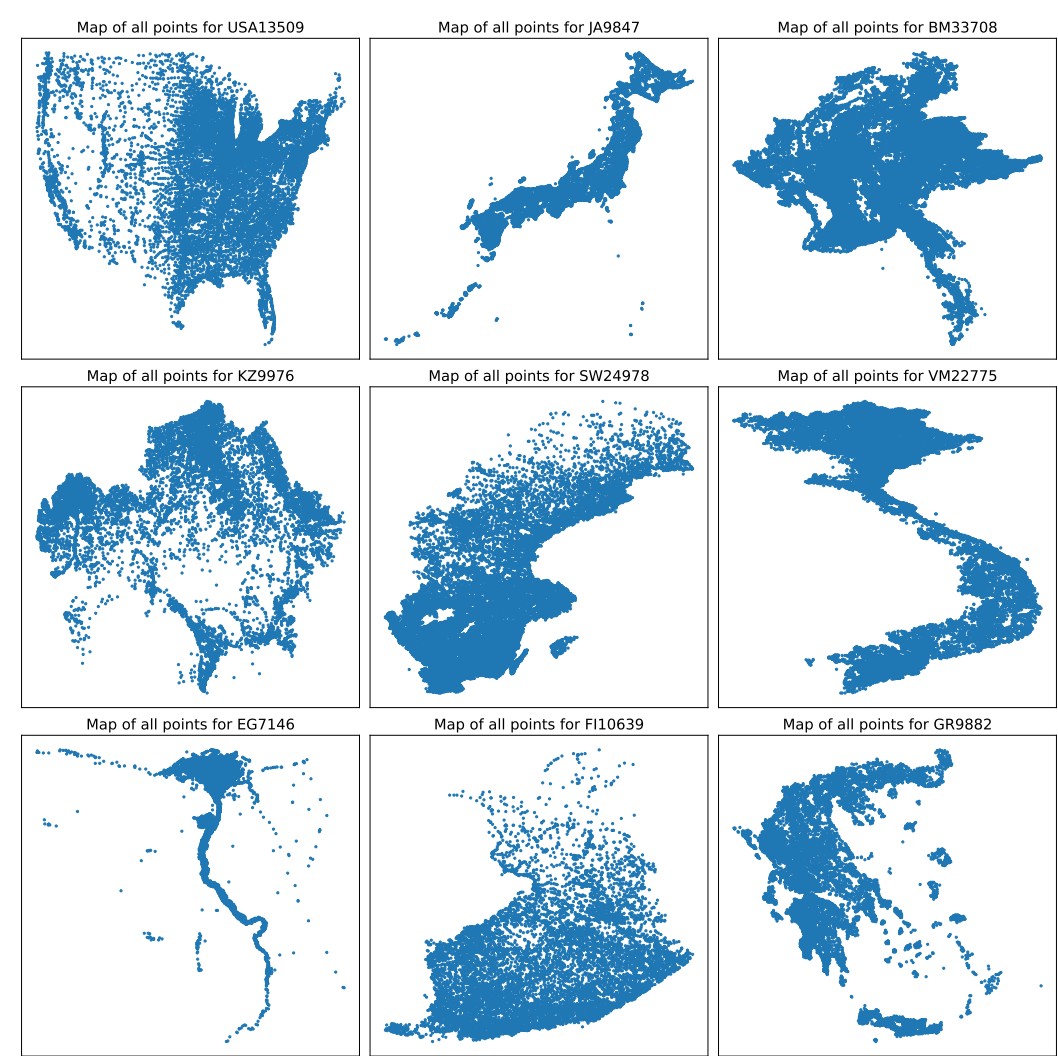

Figure 5: Plot of all 9 World Maps and their points

Table 13: Performance of models on USA13509

**In-task**

| USA13509 Problem | Solver | MTMDVRP50 Obj | MTMDVRP50 Gap | MTMDVRP50 Time | MTMDVRP100 Obj | MTMDVRP100 Gap | MTMDVRP100 Time |
|---|---|---|---|---|---|---|---|
| CVRP | HGS | 7.4382 | - | 1m34s | 11.0281 | - | 2m30s |
| | POMO-MTVRP | 7.5879 | 2.0132% | 3.22s | 11.3655 | 3.0940% | 8.71s |
| | MVMoE | 7.5507 | 1.5086% | 4.16s | 11.3352 | 2.8201% | 11.32s |
| | MVMoE-Light | 7.5570 | 1.5887% | 3.88s | 11.3493 | 2.9397% | 10.65s |
| | MVMoE-Deeper | 7.5411 | 1.3839% | 9.93s | - | - | - |
| | SHIELD-MoD | 7.5295 | 1.2313% | 6.09s | 11.2797 | 2.3098% | 17.93s |
| | SHIELD | 7.5221 | 1.1229% | 6.69s | 11.2701 | 2.2196% | 20.33s |
| OVRP | OR-tools | 4.5943 | - | 1m10s | 6.8727 | - | 2m38s |
| | POMO-MTVRP | 4.7904 | 4.2675% | 2.31s | 7.2755 | 5.9343% | 7.46s |
| | MVMoE | 4.7759 | 3.9312% | 3.35s | 7.2222 | 5.1669% | 10.24s |
| | MVMoE-Light | 4.7868 | 4.2013% | 3.16s | 7.2498 | 5.5529% | 9.38s |
| | MVMoE-Deeper | 4.7453 | 3.2923% | 8.87s | - | - | - |
| | SHIELD-MoD | 4.7290 | 2.9359% | 5.21s | 7.1346 | 3.8933% | 17.00s |
| | SHIELD | 4.7168 | 2.6767% | 6.15s | 7.0954 | 3.3129% | 19.46s |
| VRPB | OR-tools | 5.8325 | - | 1m8s | 8.5742 | - | 2m27s |
| | POMO-MTVRP | 5.9595 | 2.1771% | 2.16s | 8.7183 | 1.7366% | 6.71s |
| | MVMoE | 5.9322 | 1.7267% | 3.03s | 8.6799 | 1.2943% | 8.96s |
| | MVMoE-Light | 5.9474 | 1.9976% | 2.90s | 8.6976 | 1.4965% | 8.28s |
| | MVMoE-Deeper | 5.9275 | 1.6446% | 7.37s | - | - | - |
| | SHIELD-MoD | 5.9091 | 1.3368% | 4.69s | 8.6145 | 0.5298% | 15.12s |
| | SHIELD | 5.9019 | 1.2105% | 5.26s | 8.5929 | 0.2755% | 16.95s |
| OVRPB | OR-tools | 4.0952 | - | 1m7s | 5.9434 | - | 2m25s |
| | POMO-MTVRP | 4.4200 | 7.9311% | 2.28s | 6.5687 | 10.5536% | 6.99s |
| | MVMoE | 4.4023 | 7.4408% | 3.29s | 6.4989 | 9.3525% | 9.46s |
| | MVMoE-Light | 4.4276 | 8.0644% | 3.03s | 6.5524 | 10.2667% | 8.69s |
| | MVMoE-Deeper | 4.3726 | 6.7749% | 8.31s | - | - | - |
| | SHIELD-MoD | 4.3544 | 6.3291% | 4.97s | 6.3933 | 7.5796% | 15.75s |
| | SHIELD | 4.3542 | 6.2802% | 5.69s | 6.3535 | 6.9218% | 17.70s |
| OVRPL | OR-tools | 4.3923 | - | 1m12s | 6.9599 | - | 2m26s |
| | POMO-MTVRP | 4.8005 | 4.5344% | 2.37s | 7.3699 | 5.9831% | 7.91s |
| | MVMoE | 4.7809 | 4.0828% | 3.44s | 7.3170 | 5.2231% | 10.73s |
| | MVMoE-Light | 4.7959 | 4.4275% | 3.17s | 7.3444 | 5.6140% | 9.79s |
| | MVMoE-Deeper | 4.7667 | 3.7970% | 9.13s | - | - | - |
| | SHIELD-MoD | 4.7383 | 3.1656% | 5.27s | 7.2257 | 3.9050% | 17.57s |
| | SHIELD | 4.7267 | 2.9169% | 6.13s | 7.1917 | 3.4300% | 19.87s |
| VRPBL | OR-tools | 5.8225 | - | 1m11s | 8.5809 | - | 2m22s |
| | POMO-MTVRP | 5.9697 | 2.5288% | 2.31s | 8.7249 | 1.7249% | 7.40s |
| | MVMoE | 5.9362 | 1.9635% | 3.24s | 8.6827 | 1.2302% | 9.64s |
| | MVMoE-Light | 5.9479 | 2.1643% | 3.06s | 8.7017 | 1.4516% | 8.91s |
| | MVMoE-Deeper | 5.9425 | 2.0614% | 7.78s | - | - | - |
| | SHIELD-MoD | 5.9058 | 1.4473% | 4.77s | 8.6238 | 0.5440% | 15.81s |
| | SHIELD | 5.8952 | 1.2590% | 5.37s | 8.5988 | 0.2555% | 17.60s |
| VRPBTW | OR-tools | 9.2271 | - | 1m14s | 15.4369 | - | 2m40s |
| | POMO-MTVRP | 10.1169 | 9.6432% | 2.75s | 16.6193 | 8.0943% | 8.58s |
| | MVMoE | 10.0372 | 8.8942% | 3.88s | 16.6193 | 7.9169% | 11.39s |
| | MVMoE-Light | 10.0612 | 9.1353% | 3.61s | 16.6460 | 8.0804% | 10.69s |
| | MVMoE-Deeper | 10.0460 | 8.8750% | 9.58s | - | - | - |
| | SHIELD-MoD | 10.0014 | 8.4870% | 5.62s | 16.4699 | 6.9561% | 17.99s |
| | SHIELD | 9.9621 | 8.0901% | 6.24s | 16.3999 | 6.4802% | 20.10s |
| VRPLTW | OR-tools | 9.2840 | - | 1m18s | 15.8230 | - | 2m36s |
| | POMO-MTVRP | 9.6555 | 4.0016% | 2.81s | 16.2667 | 3.0087% | 9.90s |
| | MVMoE | 9.6052 | 3.5599% | 3.92s | 16.2093 | 2.6499% | 12.93s |
| | MVMoE-Light | 9.6202 | 3.6970% | 3.70s | 16.2532 | 2.9117% | 12.05s |
| | MVMoE-Deeper | 9.6122 | 3.5352% | 10.2s | - | - | - |
| | SHIELD-MoD | 9.5605 | 3.0615% | 5.74s | 16.0760 | 1.8171% | 20.01s |
| | SHIELD | 9.5422 | 2.8639% | 6.52s | 16.0317 | 1.5234% | 22.48s |

**Out-task**

| Problem | Solver | MTMDVRP50 Obj | MTMDVRP50 Gap | MTMDVRP50 Time | MTMDVRP100 Obj | MTMDVRP100 Gap | MTMDVRP100 Time |
|---|---|---|---|---|---|---|---|
| VRPL | OR-tools | 7.5719 | - | 1m10s | 11.5478 | - | 2m40s |
| | POMO-MTVRP | 7.6238 | 0.6848% | 2.38s | 11.4109 | -1.1490% | 7.96s |
| | MVMoE | 7.5835 | 0.1899% | 3.28s | 11.3828 | -1.3885% | 10.71s |
| | MVMoE-Light | 7.5922 | 0.2923% | 3.07s | 11.3988 | -1.2536% | 9.82s |
| | MVMoE-Deeper | 7.5709 | 0.0221% | 8.56s | - | - | - |
| | SHIELD-MoD | 7.5612 | -0.1102% | 5.02s | 11.3256 | -1.8875% | 17.29s |
| | SHIELD | 7.5547 | -0.1943% | 5.79s | 11.3108 | -2.0178% | 19.70s |
| VRPTW | OR-tools | 9.2000 | - | 1m19s | 14.9649 | - | 6m36s |
| | POMO-MTVRP | 9.7027 | 5.4640% | 2.82s | 16.1244 | 7.7828% | 9.14s |
| | MVMoE | 9.6755 | 5.1134% | 3.77s | 16.0672 | 7.3912% | 12.07s |
| | MVMoE-Light | 9.6941 | 5.3042% | 3.53s | 16.1132 | 7.6911% | 11.37s |
| | MVMoE-Deeper | 9.6401 | 4.7493% | 9.97s | - | - | - |
| | SHIELD-MoD | 9.6332 | 4.6368% | 5.71s | 15.9441 | 6.5794% | 19.06s |
| | SHIELD | 9.6035 | 4.3388% | 6.50s | 15.8862 | 6.1796% | 21.62s |
| OVRPTW | OR-tools | 5.9178 | - | 1m15s | 9.4305 | - | 2m43s |
| | POMO-MTVRP | 6.2256 | 5.2015% | 2.75s | 10.0576 | 6.7325% | 8.82s |
| | MVMoE | 6.2274 | 5.1953% | 3.91s | 10.0188 | 6.3105% | 11.80s |
| | MVMoE-Light | 6.2421 | 5.4416% | 3.56s | 10.0490 | 6.6435% | 10.85s |
| | MVMoE-Deeper | 6.1870 | 4.5388% | 10.32s | - | - | - |
| | SHIELD-MoD | 6.1663 | 4.1985% | 5.86s | 9.8820 | 4.8706% | 19.04s |
| | SHIELD | 6.1656 | 4.1838% | 6.74s | 9.8557 | 4.5972% | 21.67s |
| OVRPBL | OR-tools | 4.0893 | - | 1m9s | 5.9119 | - | 2m39s |
| | POMO-MTVRP | 4.4099 | 7.8395% | 2.35s | 6.5330 | 10.5271% | 7.41s |
| | MVMoE | 4.3940 | 7.4030% | 3.39s | 6.4591 | 9.2685% | 9.86s |
| | MVMoE-Light | 4.4193 | 8.0245% | 3.12s | 6.5207 | 10.3082% | 9.05s |
| | MVMoE-Deeper | 4.3668 | 6.7854% | 8.32s | - | - | - |
| | SHIELD-MoD | 4.3469 | 6.2984% | 5.05s | 6.3605 | 7.6109% | 16.12s |
| | SHIELD | 4.3464 | 6.2536% | 5.77s | 6.3191 | 6.9081% | 18.05s |
| OVRPBTW | OR-tools | 5.8937 | - | 1m12s | 9.3848 | - | 2m45s |
| | POMO-MTVRP | 6.5213 | 10.6484% | 2.83s | 10.5023 | 12.0111% | 8.58s |
| | MVMoE | 6.5206 | 10.5319% | 3.86s | 10.4519 | 11.4622% | 11.21s |
| | MVMoE-Light | 6.5279 | 10.6596% | 3.75s | 10.4753 | 11.7195% | 10.41s |
| | MVMoE-Deeper | 6.4880 | 10.0836% | 10.09s | - | - | - |
| | SHIELD-MoD | 6.4706 | 9.6962% | 5.89s | 10.3176 | 10.0337% | 18.34s |
| | SHIELD | 6.4546 | 9.4721% | 6.72s | 10.2672 | 9.5015% | 20.58s |
| OVRPLTW | OR-tools | 5.8319 | - | 1m15s | 9.4364 | - | 2m55s |
| | POMO-MTVRP | 6.1337 | 5.1747% | 3.15s | 10.0586 | 6.6666% | 9.24s |
| | MVMoE | 6.1313 | 5.0931% | 4.02s | 10.0215 | 6.2699% | 12.22s |
| | MVMoE-Light | 6.1423 | 5.2860% | 3.74s | 10.0512 | 6.5819% | 11.34s |
| | MVMoE-Deeper | 6.1102 | 4.7727% | 10.61s | - | - | - |
| | SHIELD-MoD | 6.0811 | 4.2453% | 6.02s | 9.8844 | 4.8210% | 19.54s |
| | SHIELD | 6.0750 | 4.1655% | 6.93s | 9.8580 | 4.5416% | 22.05s |
| VRPBLTW | OR-tools | 9.0613 | - | 1m22s | 15.4038 | - | 2m49s |
| | POMO-MTVRP | 9.9196 | 9.4717% | 3.28s | 16.6092 | 8.0791% | 9.20s |
| | MVMoE | 9.8671 | 8.9353% | 3.92s | 16.5711 | 7.8168% | 11.82s |
| | MVMoE-Light | 9.8688 | 8.9561% | 3.64s | 16.6005 | 8.0151% | 11.26s |
| | MVMoE-Deeper | 9.8627 | 8.8446% | 9.85s | - | - | - |
| | SHIELD-MoD | 9.8222 | 8.4543% | 5.73s | 16.4395 | 6.9872% | 18.61s |
| | SHIELD | 9.7819 | 7.9903% | 6.34s | 16.3696 | 6.4873% | 20.74s |
| OVRPBLTW | OR-tools | 5.8173 | - | 1m18s | 9.4629 | - | 2m38s |
| | POMO-MTVRP | 6.4355 | 10.6271% | 3.18s | 10.5841 | 11.9099% | 8.92s |
| | MVMoE | 6.4230 | 10.3227% | 3.97s | 10.5359 | 11.3903% | 11.57s |
| | MVMoE-Light | 6.4289 | 10.4245% | 3.84s | 10.5717 | 11.7672% | 10.84s |
| | MVMoE-Deeper | 6.3975 | 9.9740% | 10.25s | - | - | - |
| | SHIELD-MoD | 6.3813 | 9.6120% | 5.99s | 10.4016 | 9.9961% | 18.64s |
| | SHIELD | 6.3687 | 9.4311% | 6.82s | 10.3608 | 9.5539% | 20.80s |

Table 14: Performance of models on JA9847

| | **JA9847** Problem | Solver | MTMDVRP50 Obj | MTMDVRP50 Gap | MTMDVRP50 Time | MTMDVRP100 Obj | MTMDVRP100 Gap | MTMDVRP100 Time | Problem | Solver | MTMDVRP50 Obj | MTMDVRP50 Gap | MTMDVRP50 Time | MTMDVRP100 Obj | MTMDVRP100 Gap | MTMDVRP100 Time |
|---|---|---|---|---|---|---|---|---|---|---|---|---|---|---|---|---|
| In-task | CVRP | HGS | 5.9347 | 1m 21s | 8.7045 | 2m 12s | | | VRPL | OR-tools | 5.9291 | 1m 9s | 8.9750 | 2m 39s | | |
| | | POMO-MTVRP | 5.9686 | 1.8080% | 3.15s | 8.9352 | 2.7145% | 8.67s | | POMO-MTVRP | 5.9665 | 0.6312% | 2.35s | 8.8637 | -1.2338% | 8.00s |
| | | MVMoE | 5.9429 | 1.3723% | 4.28s | 8.9055 | 2.3673% | 11.60s | | MVMoE | 5.9350 | 0.1612% | 3.58s | 8.8337 | -1.5670% | 11.07s |
| | | MVMoE-Light | 5.9479 | 1.4661% | 4.44s | 8.9223 | 2.5692% | 10.59s | | MVMoE-Light | 5.9393 | 0.2371% | 3.07s | 8.8521 | -1.3629% | 9.86s |
| | | MVMoE-Deeper | 5.9328 | 1.2084% | 9.66s | | | | | MVMoE-Deeper | 5.9257 | 0.0021% | 8.34s | | | |
| | | SHIELD-MoD | 5.9249 | 1.0679% | 5.96s | 8.8645 | 1.8902% | 18.02s | | SHIELD-MoD | 5.9177 | -0.1350% | 5.02s | 8.7924 | -2.0387% | 17.32s |
| | | SHIELD | 5.9207 | 0.9989% | 6.63s | 8.8611 | 1.8524% | 20.42s | | SHIELD | 5.9140 | -0.2034% | 5.79s | 8.7904 | -2.0540% | 19.69s |
| | OVRP | OR-tools | 3.3709 | 1m 8s | 5.1676 | 2m 40s | | | VRPTW | OR-tools | 6.9905 | 1m 18s | 11.3101 | 6m 33s | | |
| | | POMO-MTVRP | 3.5699 | 5.9032% | 2.33s | 5.5171 | 6.9041% | 7.55s | | POMO-MTVRP | 7.2449 | 5.9169% | 2.80s | 12.1846 | 7.9817% | 9.14s |
| | | MVMoE | 3.5610 | 5.6759% | 3.63s | 5.4689 | 5.9189% | 10.99s | | MVMoE | 7.2083 | 5.4147% | 4.21s | 12.1293 | 7.4828% | 12.19s |
| | | MVMoE-Light | 3.5860 | 6.4499% | 3.22s | 5.4955 | 6.4637% | 9.31s | | MVMoE-Light | 7.2174 | 5.5424% | 3.53s | 12.1740 | 7.8991% | 11.34s |
| | | MVMoE-Deeper | 3.5076 | 4.0898% | 8.54s | | | | | MVMoE-Deeper | 7.1708 | 4.9030% | 9.88s | | | |
| | | SHIELD-MoD | 3.4963 | 3.7566% | 5.32s | 5.3515 | 3.6637% | 17.31s | | SHIELD-MoD | 7.1688 | 4.7880% | 5.66s | 12.0365 | 6.6742% | 19.06s |
| | | SHIELD | 3.4910 | 3.6111% | 6.08s | 5.3300 | 3.2637% | 19.51s | | SHIELD | 7.1579 | 4.7112% | 6.43s | 12.0109 | 6.4377% | 21.62s |
| | VRPB | OR-tools | 4.4164 | 1m 3s | 6.4448 | 2m 36s | | | OVRPTW | OR-tools | 4.1882 | 1m 12s | 6.7764 | 2m 44s | | |
| | | POMO-MTVRP | 4.5219 | 2.3878% | 2.13s | 6.5417 | 1.6225% | 6.69s | | POMO-MTVRP | 4.4770 | 6.8958% | 2.69s | 7.2570 | 7.3553% | 8.79s |
| | | MVMoE | 4.4959 | 1.8598% | 3.23s | 6.5100 | 1.1357% | 8.99s | | MVMoE | 4.4652 | 6.6707% | 4.25s | 7.2159 | 6.7570% | 12.29s |
| | | MVMoE-Light | 4.5026 | 2.0176% | 2.91s | 6.5309 | 1.4577% | 8.25s | | MVMoE-Light | 4.4809 | 7.0366% | 3.51s | 7.2523 | 7.2667% | 10.92s |
| | | MVMoE-Deeper | 4.4856 | 1.6420% | 7.02s | | | | | MVMoE-Deeper | 4.4289 | 5.8712% | 10.29s | | | |
| | | SHIELD-MoD | 4.4747 | 1.3822% | 4.69s | 6.4567 | 0.3003% | 15.08s | | SHIELD-MoD | 4.4245 | 5.6412% | 5.71s | 7.0928 | 4.9476% | 18.80s |
| | | SHIELD | 4.4667 | 1.2035% | 5.28s | 6.4494 | 0.1965% | 16.93s | | SHIELD | 4.4182 | 5.5804% | 6.58s | 7.0919 | 4.9260% | 21.33s |
| Out-task | OVRPB | OR-tools | 2.6854 | 1m 11s | 3.9796 | 2m 39s | | | OVRPBL | OR-tools | 2.7264 | 1m 7s | 3.9870 | 2m 35s | | |
| | | POMO-MTVRP | 2.9943 | 11.5013% | 2.25s | 4.5688 | 15.0151% | 7.02s | | POMO-MTVRP | 3.0326 | 11.2299% | 2.37s | 4.5780 | 15.0175% | 7.40s |
| | | MVMoE | 2.9814 | 10.9755% | 3.63s | 4.5028 | 13.2704% | 9.56s | | MVMoE | 3.0226 | 10.8299% | 3.69s | 4.5206 | 13.5069% | 10.00s |
| | | MVMoE-Light | 3.0220 | 12.5473% | 3.01s | 4.5577 | 14.6921% | 8.61s | | MVMoE-Light | 3.0648 | 12.4295% | 3.11s | 4.5720 | 14.8258% | 8.94s |
| | | MVMoE-Deeper | 2.9348 | 9.2875% | 8.1s | | | | | MVMoE-Deeper | 2.9763 | 9.1647% | 8.22s | | | |
| | | SHIELD-MoD | 2.9243 | 8.8657% | 4.91s | 4.3780 | 10.1884% | 15.83s | | SHIELD-MoD | 2.9640 | 8.7163% | 5.04s | 4.3889 | 10.2606% | 16.27s |
| | | SHIELD | 2.9195 | 8.7000% | 5.65s | 4.3530 | 9.5759% | 17.64s | | SHIELD | 2.9638 | 8.6944% | 5.70s | 4.3578 | 9.4704% | 18.05s |
| | OVRPL | OR-tools | 3.3761 | 1m 14s | 5.1001 | 2m 55s | | | OVRPBTW | OR-tools | 4.1148 | 1m 15s | 6.8126 | 2m 41s | | |
| | | POMO-MTVRP | 3.5789 | 6.0070% | 2.36s | 5.4586 | 7.1472% | 8.00s | | POMO-MTVRP | 4.5988 | 11.7626% | 2.80s | 7.5900 | 11.7303% | 8.45s |
| | | MVMoE | 3.5694 | 5.6644% | 3.78s | 5.4084 | 6.1290% | 11.59s | | MVMoE | 4.5813 | 11.3687% | 4.20s | 7.5476 | 11.1123% | 11.20s |
| | | MVMoE-Light | 3.5895 | 6.4082% | 3.17s | 5.4262 | 6.4924% | 9.77s | | MVMoE-Light | 4.5881 | 11.5339% | 3.67s | 7.5759 | 11.5058% | 10.45s |
| | | MVMoE-Deeper | 3.5249 | 4.4067% | 9.01s | | | | | MVMoE-Deeper | 4.5493 | 10.5600% | 9.91s | | | |
| | | SHIELD-MoD | 3.5010 | 3.7652% | 5.25s | 5.2866 | 3.7530% | 17.71s | | SHIELD-MoD | 4.5259 | 10.0152% | 5.74s | 7.4264 | 9.3600% | 17.94s |
| | | SHIELD | 3.4964 | 3.6483% | 6.12s | 5.2775 | 3.5944% | 19.93s | | SHIELD | 4.5198 | 9.9372% | 6.52s | 7.3886 | 8.7518% | 20.13s |
| | VRPBL | OR-tools | 4.3894 | 1m 13s | 6.4010 | 2m 49s | | | OVRPLTW | OR-tools | 4.1520 | 1m 17s | 6.8440 | 2m 51s | | |
| | | POMO-MTVRP | 4.4933 | 2.3667% | 2.25s | 6.4699 | 1.1785% | 9.64s | | POMO-MTVRP | 4.4365 | 6.8511% | 2.80s | 7.3216 | 7.3075% | 9.15s |
| | | MVMoE | 4.4657 | 1.7842% | 3.63s | 6.4997 | 1.6473% | 7.38s | | MVMoE | 4.4265 | 6.6381% | 4.20s | 7.2802 | 6.6918% | 12.62s |
| | | MVMoE-Light | 4.4737 | 1.9700% | 3.02s | 6.4901 | 1.4975% | 8.87s | | MVMoE-Light | 4.4423 | 7.0211% | 3.68s | 7.3130 | 7.1614% | 11.48s |
| | | MVMoE-Deeper | 4.4728 | 1.9007% | 8.69s | | | | | MVMoE-Deeper | 4.4021 | 6.0241% | 10.39s | | | |
| | | SHIELD-MoD | 4.4469 | 1.3514% | 4.75s | 6.4182 | 0.3668% | 15.77s | | SHIELD-MoD | 4.3809 | 5.5133% | 5.88s | 7.1516 | 4.8308% | 19.20s |
| | | SHIELD | 4.4357 | 1.1004% | 5.35s | 6.4115 | 0.002659 | 17.64s | | SHIELD | 4.3734 | 5.4095% | 6.75s | 7.1484 | 4.7587% | 21.69s |
| | VRPBTW | OR-tools | 6.7862 | 1m 20s | 11.8462 | 2m 42s | | | VRPBLTW | OR-tools | 6.8945 | 1m 22s | 12.1613 | 2m 46s | | |
| | | POMO-MTVRP | 7.3740 | 8.6621% | 2.73s | 12.6045 | 6.8206% | 8.52s | | POMO-MTVRP | 7.4997 | 8.7784% | 2.95s | 12.9318 | 6.8330% | 9.23s |
| | | MVMoE | 7.3203 | 8.1467% | 4.28s | 12.5412 | 6.2745% | 11.42s | | MVMoE | 7.4382 | 8.1638% | 4.18s | 12.8821 | 6.3975% | 12.09s |
| | | MVMoE-Light | 7.3267 | 8.2462% | 3.59s | 12.5954 | 6.7859% | 10.59s | | MVMoE-Light | 7.4476 | 8.2867% | 3.59s | 12.9371 | 6.8322% | 11.29s |
| | | MVMoE-Deeper | 7.3205 | 7.8732% | 9.44s | | | | | MVMoE-Deeper | 7.4418 | 7.9377% | 9.76s | | | |
| | | SHIELD-MoD | 7.2765 | 7.4651% | 5.57s | 12.4460 | 5.4695% | 17.90s | | SHIELD-MoD | 7.3976 | 7.5407% | 5.65s | 12.7737 | 5.5222% | 18.69s |
| | | SHIELD | 7.2522 | 7.1540% | 6.12s | 12.3966 | 5.0097% | 20.18s | | SHIELD | 7.3750 | 7.2483% | 6.26s | 12.7383 | 5.1504% | 20.94s |
| | VRPLTW | OR-tools | 7.0420 | 1m 24s | 12.0881 | 2m 50s | | | OVRPBLTW | OR-tools | 4.0716 | 1m 19s | 6.8237 | 2m 33s | | |
| | | POMO-MTVRP | 7.3305 | 4.0966% | 2.84s | 12.3506 | 2.5592% | 9.92s | | POMO-MTVRP | 4.5587 | 11.9622% | 2.85s | 7.6099 | 11.8026% | 8.82s |
| | | MVMoE | 7.2767 | 3.5074% | 4.22s | 12.3045 | 2.1596% | 12.99s | | MVMoE | 4.5427 | 11.5344% | 4.33s | 7.5505 | 10.9459% | 11.62s |
| | | MVMoE-Light | 7.2805 | 3.5980% | 3.67s | 12.3385 | 2.4430% | 12.01s | | MVMoE-Light | 4.5505 | 11.7197% | 3.72s | 7.5902 | 11.5239% | 10.79s |
| | | MVMoE-Deeper | 7.2765 | 3.3301% | 10.02s | | | | | MVMoE-Deeper | 4.5088 | 10.7374% | 10.03s | | | |
| | | SHIELD-MoD | 7.2300 | 2.8305% | 5.71s | 12.1935 | 1.2312% | 19.96s | | SHIELD-MoD | 4.4880 | 10.1687% | 5.82s | 7.4267 | 9.1647% | 18.30s |
| | | SHIELD | 7.2230 | 2.7590% | 6.50s | 12.1775 | 1.0937% | 22.52s | | SHIELD | 4.4808 | 10.0961% | 6.56s | 7.4043 | 8.7984% | 20.45s |

## Table 15: Performance of models on BM33708

**In-task**

| BM33708 Problem | Solver | MTMDVRP50 Obj | Gap | Time | MTMDVRP100 Obj | Gap | Time |
|---|---|---|---|---|---|---|---|
| CVRP | HGS | 6.5032 | - | 1m25s | 9.5205 | - | 2m11s |
| | POMO-MTVRP | 6.5373 | 1.9983% | 3.28s | 9.8019 | 2.9725% | 8.71s |
| | MVMoE | 6.5072 | 1.5219% | 4.46s | 9.7728 | 2.6616% | 11.44s |
| | MVMoE-Light | 6.5137 | 1.6263% | 3.87s | 9.7950 | 2.8929% | 10.58s |
| | MVMoE-Deeper | 6.4984 | 1.3845% | 9.58s | - | - | - |
| | SHIELD-MoD | 6.4897 | 1.2503% | 5.76s | 9.7312 | 2.2300% | 18.39s |
| | SHIELD | 6.4843 | 1.1648% | 6.47s | 9.7160 | 2.0607% | 20.45s |
| OVRP | OR-tools | 3.9920 | - | 1m12s | 5.9998 | - | 2m39s |
| | POMO-MTVRP | 4.1641 | 4.3105% | 2.59s | 6.3549 | 5.9698% | 7.47s |
| | MVMoE | 4.1523 | 3.9851% | 3.67s | 6.3028 | 5.0992% | 10.18s |
| | MVMoE-Light | 4.1634 | 4.2685% | 3.08s | 6.3316 | 5.5745% | 9.39s |
| | MVMoE-Deeper | 4.1270 | 3.3616% | 8.62s | 6.2323 | 3.9265% | 17.05s |
| | SHIELD-MoD | 4.1111 | 2.9609% | 5.21s | 6.1989 | 3.3706% | 19.51s |
| | SHIELD | 4.1012 | 2.7202% | 5.94s | - | - | - |
| VRPB | OR-tools | 5.1214 | - | 1m8s | 7.5311 | - | 2m37s |
| | POMO-MTVRP | 5.2323 | 2.1659% | 2.30s | 7.6426 | 1.5415% | 6.72s |
| | MVMoE | 5.2088 | 1.7189% | 3.08s | 7.6019 | 1.0042% | 8.96s |
| | MVMoE-Light | 5.2203 | 1.9473% | 3.04s | 7.6283 | 1.3498% | 8.27s |
| | MVMoE-Deeper | 5.1985 | 1.5129% | 7.03s | - | - | - |
| | SHIELD-MoD | 5.1872 | 1.2899% | 4.70s | 7.5605 | 0.4521% | 15.07s |
| | SHIELD | 5.1774 | 1.1052% | 5.24s | 7.5432 | 0.2217% | 16.96s |
| OVRPB | OR-tools | 3.5304 | - | 1m14s | 5.1150 | - | 2m40s |
| | POMO-MTVRP | 3.8075 | 7.8483% | 2.45s | 5.6545 | 10.5678% | 6.98s |
| | MVMoE | 3.7854 | 7.1879% | 3.30s | 5.5802 | 9.1051% | 9.63s |
| | MVMoE-Light | 3.8044 | 7.7261% | 3.23s | 5.6390 | 10.2535% | 8.66s |
| | MVMoE-Deeper | 3.7667 | 6.6935% | 7.98s | - | - | - |
| | SHIELD-MoD | 3.7513 | 6.2563% | 4.96s | 5.4924 | 7.4002% | 15.71s |
| | SHIELD | 3.7476 | 6.1222% | 5.59s | 5.4591 | 6.7559% | 17.68s |
| OVRPL | OR-tools | 3.9981 | - | 1m18s | 5.9357 | - | 2m53s |
| | POMO-MTVRP | 4.1679 | 4.5092% | 2.53s | 6.2854 | 5.9394% | 7.87s |
| | MVMoE | 4.1430 | 3.8587% | 3.46s | 6.2359 | 5.1059% | 10.72s |
| | MVMoE-Light | 4.1571 | 4.2170% | 3.25s | 6.2623 | 5.5449% | 9.83s |
| | MVMoE-Deeper | 4.1379 | 3.7570% | 9.04s | - | - | - |
| | SHIELD-MoD | 4.1054 | 2.9310% | 5.21s | 6.1603 | 3.8383% | 17.47s |
| | SHIELD | 4.0957 | 2.6807% | 6.02s | 6.1346 | 3.4063% | 19.90s |
| VRPBL | OR-tools | 5.1312 | - | 1m16s | 7.5768 | - | 2m35s |
| | POMO-MTVRP | 5.2568 | 2.4473% | 2.42s | 7.6932 | 1.5959% | 7.38s |
| | MVMoE | 5.2191 | 1.7541% | 3.26s | 7.6563 | 1.1044% | 9.59s |
| | MVMoE-Light | 5.2303 | 1.9696% | 3.06s | 7.6777 | 1.3801% | 8.90s |
| | MVMoE-Deeper | 5.2357 | 2.0373% | 8.81s | - | - | - |
| | SHIELD-MoD | 5.1972 | 1.3226% | 4.74s | 7.6094 | 0.4846% | 15.75s |
| | SHIELD | 5.1873 | 1.1250% | 5.34s | 7.5898 | 0.2250% | 17.67s |
| VRPBTW | OR-tools | 7.4449 | - | 1m21s | 12.4088 | - | 2m35s |
| | POMO-MTVRP | 8.1811 | 9.8882% | 3.17s | 13.4097 | 8.2633% | 8.46s |
| | MVMoE | 8.1267 | 9.2211% | 3.89s | 13.3707 | 7.9318% | 11.23s |
| | MVMoE-Light | 8.1238 | 9.1848% | 3.59s | 13.4078 | 8.2289% | 10.47s |
| | MVMoE-Deeper | 8.1340 | 9.2566% | 9.53s | - | - | - |
| | SHIELD-MoD | 8.0782 | 8.5745% | 5.57s | 13.2689 | 7.1352% | 17.61s |
| | SHIELD | 8.0547 | 8.2630% | 6.17s | 13.2026 | 6.5746% | 19.68s |
| VRPLTW | OR-tools | 7.6281 | - | 1m33s | 12.4766 | - | 2m47s |
| | POMO-MTVRP | 7.9617 | 4.3739% | 3.50s | 12.8902 | 3.4968% | 9.72s |
| | MVMoE | 7.9210 | 3.8770% | 3.91s | 12.8550 | 3.1971% | 12.71s |
| | MVMoE-Light | 7.9339 | 4.0594% | 3.71s | 12.8826 | 3.4163% | 11.81s |
| | MVMoE-Deeper | 7.9339 | 4.0089% | 10.1s | - | - | - |
| | SHIELD-MoD | 7.8808 | 3.3677% | 5.68s | 12.7513 | 2.3939% | 19.45s |
| | SHIELD | 7.8676 | 3.1980% | 6.43s | 12.7150 | 2.0844% | 21.88s |

**Out-task**

| Problem | Solver | MTMDVRP50 Obj | Gap | Time | MTMDVRP100 Obj | Gap | Time |
|---|---|---|---|---|---|---|---|
| VRPL | OR-tools | 6.5389 | - | 1m15s | 9.9236 | - | 2m37s |
| | POMO-MTVRP | 6.5722 | 0.5091% | 2.51s | 9.8052 | -1.1388% | 7.97s |
| | MVMoE | 6.5382 | 0.0282% | 3.35s | 9.7744 | -1.4515% | 10.71s |
| | MVMoE-Light | 6.5459 | 0.1420% | 3.39s | 9.7952 | -1.2381% | 9.83s |
| | MVMoE-Deeper | 6.5289 | -0.1156% | 8.41s | - | - | - |
| | SHIELD-MoD | 6.5195 | -0.2635% | 5.02s | 9.7327 | -1.8684% | 17.31s |
| | SHIELD | 6.5148 | -0.3338% | 5.76s | 9.7200 | -2.0014% | 19.70s |
| VRPTW | OR-tools | 7.6658 | - | 1m22s | 12.0249 | - | 6m25s |
| | POMO-MTVRP | 7.9788 | 5.7045% | 2.96s | 12.9875 | 8.0100% | 9.00s |
| | MVMoE | 7.9591 | 5.3968% | 3.86s | 12.9415 | 7.6221% | 12.20s |
| | MVMoE-Light | 7.9703 | 5.5481% | 3.76s | 12.9698 | 7.8572% | 11.18s |
| | MVMoE-Deeper | 7.9357 | 5.0955% | 9.86s | - | - | - |
| | SHIELD-MoD | 7.9158 | 4.8116% | 5.66s | 12.8390 | 6.7800% | 18.72s |
| | SHIELD | 7.9004 | 4.6212% | 6.39s | 12.7998 | 6.4508% | 21.21s |
| OVRPTW | OR-tools | 5.0201 | - | 1m16s | 8.0463 | - | 2m34s |
| | POMO-MTVRP | 5.2763 | 5.1035% | 2.90s | 8.5785 | 6.6527% | 8.82s |
| | MVMoE | 5.2834 | 5.1907% | 3.89s | 8.5365 | 6.1197% | 11.72s |
| | MVMoE-Light | 5.2918 | 5.3632% | 3.73s | 8.5722 | 6.5587% | 10.85s |
| | MVMoE-Deeper | 5.2542 | 4.6315% | 10.23s | - | - | - |
| | SHIELD-MoD | 5.2383 | 4.2937% | 5.83s | 8.4346 | 4.8748% | 19.04s |
| | SHIELD | 5.2333 | 4.2265% | 6.68s | 8.4155 | 4.6225% | 21.62s |
| OVRPBL | OR-tools | 3.5357 | - | 1m20s | 5.1156 | - | 2m34s |
| | POMO-MTVRP | 3.8204 | 8.0509% | 3.10s | 5.6544 | 10.5263% | 7.38s |
| | MVMoE | 3.7958 | 7.2984% | 3.41s | 5.5812 | 9.1010% | 9.81s |
| | MVMoE-Light | 3.8204 | 7.9910% | 3.11s | 5.6376 | 10.2014% | 9.02s |
| | MVMoE-Deeper | 3.7805 | 6.9246% | 8.33s | - | - | - |
| | SHIELD-MoD | 3.7656 | 6.5012% | 5.04s | 5.4930 | 7.3890% | 16.09s |
| | SHIELD | 3.7616 | 6.3349% | 5.77s | 5.4609 | 6.7607% | 18.04s |
| OVRPBTW | OR-tools | 4.9702 | - | 1m20s | 7.9711 | - | 2m44s |
| | POMO-MTVRP | 5.4885 | 10.4286% | 3.47s | 8.9031 | 11.7278% | 8.53s |
| | MVMoE | 5.4754 | 10.1045% | 3.94s | 8.8646 | 11.2500% | 11.16s |
| | MVMoE-Light | 5.4791 | 10.2090% | 3.75s | 8.8888 | 11.5506% | 10.34s |
| | MVMoE-Deeper | 5.4665 | 9.9856% | 10.01s | - | - | - |
| | SHIELD-MoD | 5.4432 | 9.4653% | 5.84s | 8.7594 | 9.9495% | 18.19s |
| | SHIELD | 5.4334 | 9.2902% | 6.44s | 8.7285 | 9.5463% | 20.42s |
| OVRPLTW | OR-tools | 4.9822 | - | 1m30s | 8.0416 | - | 2m48s |
| | POMO-MTVRP | 5.2483 | 5.3405% | 3.51s | 8.5824 | 6.7651% | 9.23s |
| | MVMoE | 5.2444 | 5.2144% | 4.04s | 8.5412 | 6.2512% | 12.14s |
| | MVMoE-Light | 5.2263 | 5.4492% | 3.73s | 8.5730 | 6.6407% | 11.32s |
| | MVMoE-Deeper | 5.2335 | 5.0438% | 10.5s | - | - | - |
| | SHIELD-MoD | 5.2019 | 4.3630% | 6.01s | 8.4339 | 4.9271% | 19.43s |
| | SHIELD | 5.1979 | 4.3038% | 6.88s | 8.4186 | 4.7310% | 22.07s |
| VRPBLTW | OR-tools | 7.4143 | - | 1m40s | 12.4970 | - | 2m41s |
| | POMO-MTVRP | 8.1315 | 9.6728% | 3.73s | 13.5021 | 8.2643% | 9.07s |
| | MVMoE | 8.0779 | 8.9961% | 3.90s | 13.4496 | 7.8306% | 11.85s |
| | MVMoE-Light | 8.0827 | 9.0685% | 3.62s | 13.4871 | 8.1257% | 11.09s |
| | MVMoE-Deeper | 8.0907 | 9.1229% | 9.82s | - | - | - |
| | SHIELD-MoD | 8.0423 | 8.5104% | 5.67s | 13.3439 | 7.0136% | 18.27s |
| | SHIELD | 8.0099 | 8.0833% | 6.29s | 13.2892 | 6.5421% | 20.34s |
| OVRPBLTW | OR-tools | 4.9601 | - | 1m32s | 8.0296 | - | 2m34s |
| | POMO-MTVRP | 5.4895 | 10.6725% | 3.57s | 8.9600 | 11.6180% | 8.91s |
| | MVMoE | 5.4732 | 10.3069% | 4.02s | 8.9232 | 11.1516% | 11.44s |
| | MVMoE-L | 5.4830 | 10.5132% | 3.83s | 8.9511 | 11.5117% | 10.73s |
| | MVMoE-Deeper | 5.4668 | 10.2162% | 10.21s | - | - | - |
| | MVMoD | 5.4426 | 9.6962% | 5.95s | 8.8254 | 9.9594% | 18.62s |
| | Ours | 5.4346 | 9.5587% | 6.75s | 8.7979 | 9.6128% | 20.86s |

Table 16: Performance of models on KZ9976

**KZ9976 (left block)**

| | Problem | Solver | MTMDVRP50 Obj | MTMDVRP50 Gap | MTMDVRP50 Time | MTMDVRP100 Obj | MTMDVRP100 Gap | MTMDVRP100 Time |
|---|---|---|---|---|---|---|---|---|
| In-task | CVRP | HGS | 8.4217 | - | 1m 17s | 12.4181 | - | 2m 14s |
| | | POMO-MTVRP | 8.4796 | 2.1707% | 2.98s | 12.8288 | 3.3197% | 8.66s |
| | | MVMoE | 8.4334 | 1.6093% | 4.36s | 12.7846 | 2.9640% | 11.31s |
| | | MVMoE-Light | 8.441 | 1.7004% | 4.13s | 12.8041 | 3.1223% | 10.98s |
| | | MVMoE-Deeper | 8.4149 | 1.3910% | 9.56s | - | - | - |
| | | SHIELD-MoD | 8.4057 | 1.2745% | 5.79s | 12.7248 | 2.4846% | 18.03s |
| | | SHIELD | 8.3991 | 1.1865% | 6.46s | 12.7058 | 2.3312% | 20.42s |
| | OVRP | OR-tools | 5.0798 | - | 1m 5s | 7.6637 | - | 2m 37s |
| | | POMO-MTVRP | 5.314 | 4.6202% | 2.37s | 8.1047 | 5.8375% | 7.44s |
| | | MVMoE | 5.2892 | 4.1392% | 3.42s | 8.0450 | 5.0610% | 10.16s |
| | | MVMoE-Light | 5.2966 | 4.2863% | 3.04s | 8.0662 | 5.3313% | 9.33s |
| | | MVMoE-Deeper | 5.2511 | 3.3950% | 8.63s | - | - | - |
| | | SHIELD-MoD | 5.239 | 3.1536% | 5.13s | 7.9500 | 3.8202% | 17.04s |
| | | SHIELD | 5.2274 | 2.9232% | 5.92s | 7.8905 | 3.0341% | 19.49s |
| | VRPB | OR-tools | 6.332 | - | 1m 4s | 9.3879 | - | 2m 38s |
| | | POMO-MTVRP | 6.4841 | 2.4023% | 2.20s | 9.5585 | 1.2055% | 6.74s |
| | | MVMoE | 6.4416 | 1.7613% | 3.02s | 9.4946 | 1.5526% | 8.96s |
| | | MVMoE-Light | 6.459 | 2.0400% | 2.83s | 9.5275 | | 8.30s |
| | | MVMoE-Deeper | 6.4264 | 1.5232% | 7.04s | - | - | - |
| | | SHIELD-MoD | 6.4131 | 1.3130% | 4.71s | 9.4364 | 0.5879% | 15.04s |
| | | SHIELD | 6.4203 | 1.1505% | 5.25s | 9.4097 | 0.2961% | 16.92s |
| | OVRPB | OR-tools | 4.2834 | - | 1m 10s | 6.2087 | - | 2m 39s |
| | | POMO-MTVRP | 4.6591 | 8.7721% | 2.25s | 6.9177 | 11.4873% | 6.98s |
| | | MVMoE | 4.6161 | 7.7478% | 3.36s | 6.7990 | 9.5636% | 9.40s |
| | | MVMoE-Light | 4.6559 | 8.6910% | 2.98s | 6.8691 | 10.71103% | 8.66s |
| | | MVMoE-Deeper | 4.5958 | 7.2939% | 7.99s | - | - | - |
| | | SHIELD-MoD | 4.5812 | 6.9517% | 4.91s | 6.6961 | 7.9126% | 15.67s |
| | | SHIELD | 4.5743 | 6.7822% | 5.57s | 6.6557 | 7.2445% | 17.61s |
| Out-task | OVRPL | OR-tools | 5.0382 | - | 1m 14s | 7.6885 | - | 2m 54s |
| | | POMO-MTVRP | 5.2716 | 4.6326% | 2.36s | 8.1227 | 5.7422% | 7.88s |
| | | MVMoE | 5.2428 | 4.0808% | 3.48s | 8.0570 | 4.8769% | 10.76s |
| | | MVMoE-Light | 5.2548 | 4.3204% | 3.16s | 8.0883 | 5.2854% | 9.78s |
| | | MVMoE-Deeper | 5.2318 | 3.8432% | 9.03s | - | - | - |
| | | SHIELD-MoD | 5.1909 | 3.0462% | 5.23s | 7.9654 | 3.6885% | 17.51s |
| | | SHIELD | 5.1812 | 2.8560% | 6.04s | 7.9175 | 3.0511% | 19.83s |
| | VRPBL | OR-tools | 6.3024 | - | 1m 13s | 9.4149 | - | 2m 33s |
| | | POMO-MTVRP | 6.4771 | 2.7726% | 2.26s | 9.6073 | 2.1055% | 7.43s |
| | | MVMoE | 6.4204 | 1.8865% | 3.28s | 9.5380 | 1.3777% | 9.59s |
| | | MVMoE-Light | 6.4364 | 2.1453% | 3.02s | 9.5682 | 1.6922% | 8.91s |
| | | MVMoE-Deeper | 6.4305 | 2.0327% | 8.8s | - | - | - |
| | | SHIELD-MoD | 6.3904 | 1.4119% | 4.76s | 9.4791 | 0.7505% | 15.72s |
| | | SHIELD | 6.3788 | 1.2310% | 5.32s | 9.4541 | 0.4835% | 17.63s |
| | VRPBTW | OR-tools | 10.6457 | - | 1m 20s | 18.3619 | - | 2m 44s |
| | | POMO-MTVRP | 11.7415 | 10.2929% | 2.77s | 19.8107 | 8.0818% | 8.62s |
| | | MVMoE | 11.6367 | 9.4073% | 4.00s | 19.7718 | 7.8616% | 11.26s |
| | | MVMoE-Light | 11.6477 | 9.6440% | 3.61s | 19.7959 | 7.9818% | 10.65s |
| | | MVMoE-Deeper | 11.6333 | 9.2771% | 9.52s | - | - | - |
| | | SHIELD-MoD | 11.5870 | 8.9121% | 5.62s | 19.5695 | 6.7684% | 17.94s |
| | | SHIELD | 11.5423 | 8.5047% | 6.22s | 19.4954 | 6.4336% | 20.06s |
| | VRPLTW | OR-tools | 10.6950 | - | 1m 23s | 18.2887 | - | 2m 49s |
| | | POMO-MTVRP | 11.1707 | 4.4476% | 2.82s | 18.8087 | 3.0163% | 9.86s |
| | | MVMoE | 11.0690 | 3.5796% | 3.95s | 18.7728 | 2.8171% | 12.71s |
| | | MVMoE-Light | 11.1070 | 3.9295% | 3.68s | 18.8008 | 2.9582% | 11.99s |
| | | MVMoE-Deeper | 11.0888 | 3.6817% | 10s | - | - | - |
| | | SHIELD-MoD | 11.0282 | 3.1870% | 5.78s | 18.5787 | 1.7581% | 19.89s |
| | | SHIELD | 10.9948 | 2.8804% | 6.50s | 18.5216 | 1.4410% | 22.30s |

**KZ9976 (right block)**

| Problem | Solver | MTMDVRP50 Obj | MTMDVRP50 Gap | MTMDVRP50 Time | MTMDVRP100 Obj | MTMDVRP100 Gap | MTMDVRP100 Time |
|---|---|---|---|---|---|---|---|
| VRPL | OR-tools | 8.4633 | - | 1m 10s | 12.8865 | - | 2m 39s |
| | POMO-MTVRP | 8.5304 | 0.7927% | 2.40s | 12.7791 | -0.7886% | 8.01s |
| | MVMoE | 8.4747 | 0.1676% | 3.35s | 12.7344 | -1.1332% | 10.71s |
| | MVMoE-Light | 8.4820 | 0.2478% | 3.05s | 12.7580 | -0.9518% | 9.84s |
| | MVMoE-Deeper | 8.4577 | -0.0367% | 8.44s | - | - | - |
| | SHIELD-MoD | 8.4511 | -0.1117% | 5.01s | 12.6752 | -1.5891% | 17.34s |
| | SHIELD | 8.4423 | -0.2183% | 5.71s | 12.6599 | -1.7116% | 19.68s |
| VRPTW | OR-tools | 10.6491 | - | 1m 19s | 17.3625 | - | 6m 32s |
| | POMO-MTVRP | 11.1016 | 6.1918% | 2.82s | 18.8165 | 8.4175% | 9.10s |
| | MVMoE | 11.0366 | 5.5490% | 3.90s | 18.7844 | 8.2241% | 11.95s |
| | MVMoE-Light | 11.0857 | 5.9973% | 3.50s | 18.8030 | 8.3233% | 11.31s |
| | MVMoE-Deeper | 10.9993 | 5.1885% | 9.9s | - | - | - |
| | SHIELD-MoD | 10.9963 | 5.1533% | 5.74s | 18.6051 | 7.1926% | 19.07s |
| | SHIELD | 10.9675 | 4.8838% | 6.46s | 18.5330 | 6.7691% | 21.51s |
| OVRPTW | OR-tools | 6.4917 | - | 1m 13s | 10.6668 | - | 2m 42s |
| | POMO-MTVRP | 6.8737 | 5.8447% | 2.73s | 11.3865 | 6.8257% | 8.79s |
| | MVMoE | 6.8558 | 5.6048% | 3.91s | 11.3429 | 6.4039% | 11.67s |
| | MVMoE-Light | 6.8743 | 5.8860% | 3.52s | 11.3584 | 6.5555% | 10.71s |
| | MVMoE-Deeper | 6.8098 | 4.9103% | 10.22s | - | - | - |
| | SHIELD-MoD | 6.8059 | 4.8336% | 5.83s | 11.1768 | 4.8610% | 19.00s |
| | SHIELD | 6.7798 | 4.4422% | 6.63s | 11.1398 | 4.5167% | 21.61s |
| OVRPBL | OR-tools | 4.2813 | - | 1m 6s | 6.1967 | - | 2m 31s |
| | POMO-MTVRP | 4.6503 | 8.6179% | 2.34s | 6.9034 | 11.4733% | 7.35s |
| | MVMoE | 4.6112 | 7.6726% | 3.36s | 6.7926 | 9.6704% | 9.76s |
| | MVMoE-Light | 4.6486 | 8.5437% | 3.07s | 6.8705 | 10.9271% | 9.04s |
| | MVMoE-Deeper | 4.5877 | 7.1562% | 8.29s | - | - | - |
| | SHIELD-MoD | 4.5717 | 6.7819% | 5.03s | 6.6910 | 8.0347% | 16.11s |
| | SHIELD | 4.5686 | 6.6880% | 5.65s | 6.6351 | 7.1246% | 17.96s |
| OVRPBTW | OR-tools | 6.4426 | - | 1m 14s | 10.6121 | - | 2m 42s |
| | POMO-MTVRP | 7.2019 | 11.7856% | 2.77s | 11.9287 | 12.4815% | 8.50s |
| | MVMoE | 7.1797 | 11.4104% | 3.92s | 11.8841 | 12.0447% | 11.14s |
| | MVMoE-Light | 7.1893 | 11.5716% | 3.69s | 11.8949 | 12.1569% | 10.30s |
| | MVMoE-Deeper | 7.1516 | 11.0056% | 10.04s | - | - | - |
| | SHIELD-MoD | 7.1353 | 10.7090% | 5.84s | 11.7189 | 10.4973% | 18.09s |
| | SHIELD | 7.1020 | 10.2196% | 6.59s | 11.6645 | 9.9822% | 20.31s |
| OVRPLTW | OR-tools | 6.5074 | - | 1m 17s | 10.5746 | - | 2m 54s |
| | POMO-MTVRP | 6.8985 | 6.0097% | 2.82s | 11.2864 | 6.7992% | 9.22s |
| | MVMoE | 6.8964 | 5.5594% | 4.11s | 11.2550 | 6.4918% | 12.12s |
| | MVMoE-Light | 6.8832 | 5.7811% | 3.70s | 11.2703 | 6.6409% | 11.17s |
| | MVMoE-Deeper | 6.8490 | 5.2494% | 10.46s | - | - | - |
| | SHIELD-MoD | 6.8213 | 4.8220% | 5.97s | 11.0853 | 4.9005% | 19.42s |
| | SHIELD | 6.7949 | 4.4390% | 6.79s | 11.0480 | 4.5534% | 21.95s |
| VRPBLTW | OR-tools | 10.5947 | - | 1m 22s | 18.3014 | - | 2m 47s |
| | POMO-MTVRP | 11.7074 | 10.5025% | 2.94s | 19.7894 | 8.3381% | 9.25s |
| | MVMoE | 11.5911 | 9.5324% | 4.00s | 19.7494 | 8.1026% | 11.78s |
| | MVMoE-Light | 11.6260 | 9.8357% | 3.63s | 19.7794 | 8.2802% | 11.22s |
| | MVMoE-Deeper | 11.6011 | 9.4993% | 9.79s | - | - | - |
| | SHIELD-MoD | 11.5585 | 9.2067% | 5.74s | 19.5707 | 7.1332% | 18.54s |
| | SHIELD | 11.5051 | 8.6889% | 6.32s | 19.4922 | 6.6791% | 20.70s |
| OVRPBLTW | OR-tools | 6.4313 | - | 1m 19s | 10.6460 | - | 2m 33s |
| | POMO-MTVRP | 7.1961 | 11.8922% | 2.90s | 11.9586 | 12.3982% | 8.81s |
| | MVMoE | 7.1622 | 11.3643% | 4.05s | 11.9137 | 11.9667% | 11.41s |
| | MVMoE-L | 7.1742 | 11.5651% | 3.76s | 11.9164 | 11.9921% | 10.68s |
| | MVMoE-Deeper | 7.1340 | 10.9255% | 10.24s | - | - | - |
| | MVMoD | 7.1239 | 10.7710% | 5.93s | 11.7414 | 10.3568% | 18.46s |
| | Ours | 7.0845 | 10.1846% | 6.67s | 11.6952 | 9.9203% | 20.71s |

Table 17: Performance of models on SW24978

**SW24978 — Left block**

| | Problem | Solver | MTMDVRP50 Obj | Gap | Time | MTMDVRP100 Obj | Gap | Time |
|---|---|---|---|---|---|---|---|---|
| In-task | CVRP | HGS | 6.6979 | - | 1m 18s | 9.8826 | - | 2m 11s |
| | | POMO-MTVRP | 6.7538 | 2.2739% | 3.06s | 10.2519 | 3.78760% | 8.66s |
| | | MVMoE | 6.7181 | 1.7447% | 4.43s | 10.2290 | 3.56040% | 12.50s |
| | | MVMoE-Light | 6.7260 | 1.8586% | 3.89s | 10.2507 | 3.77250% | 10.58s |
| | | MVMoE-Deeper | 6.7072 | 1.5831% | 9.7s | - | - | - |
| | | SHIELD-MoD | 6.6937 | 1.3636% | 5.81s | 10.1538 | 2.79320% | 18.11s |
| | | SHIELD | 6.6842 | 1.2169% | 6.54s | 10.1386 | 2.62190% | 20.38s |
| | OVRP | OR-tools | 4.0521 | - | 1m 9s | 6.1626 | - | 2m 38s |
| | | POMO-MTVRP | 4.2564 | 5.0417% | 2.39s | 6.9952 | 7.13210% | 7.47s |
| | | MVMoE | 4.2382 | 4.6192% | 3.49s | 6.5459 | 6.33510% | 11.83s |
| | | MVMoE-Light | 4.2492 | 4.8916% | 3.09s | 6.5766 | 6.81880% | 9.36s |
| | | MVMoE-Deeper | 4.2075 | 3.8666% | 8.72s | - | - | - |
| | | SHIELD-MoD | 4.1888 | 3.3901% | 5.30s | 6.4406 | 4.62700% | 17.16s |
| | | SHIELD | 4.1733 | 3.0110% | 6.05s | 6.3878 | 3.75130% | 19.49s |
| | VRPB | OR-tools | 5.2139 | - | 1m 2s | 7.6890 | - | 2m 40s |
| | | POMO-MTVRP | 5.3608 | 2.8184% | 2.18s | 7.8945 | 2.77250% | 6.73s |
| | | MVMoE | 5.3331 | 2.3264% | 3.30s | 7.8535 | 2.24830% | 10.36s |
| | | MVMoE-Light | 5.3395 | 2.4519% | 3.06s | 7.8847 | 2.64580% | 8.30s |
| | | MVMoE-Deeper | 5.3178 | 2.0248% | 7.02s | - | - | - |
| | | SHIELD-MoD | 5.2987 | 1.6618% | 4.70s | 7.7733 | 1.21340% | 15.13s |
| | | SHIELD | 5.2861 | 1.4189% | 5.27s | 7.7549 | 0.95640% | 16.94s |
| | OVRPB | OR-tools | 3.5427 | - | 1m 12s | 5.1907 | - | 2m 53s |
| | | POMO-MTVRP | 3.8655 | 9.1129% | 2.30s | 5.8643 | 13.03590% | 7.91s |
| | | MVMoE | 3.8442 | 8.4761% | 3.28s | 5.7911 | 11.60920% | 12.37s |
| | | MVMoE-Light | 3.8671 | 9.1296% | 3.25s | 5.8531 | 12.81730% | 9.79s |
| | | MVMoE-Deeper | 3.8229 | 7.9091% | 8.02s | - | - | - |
| | | SHIELD-MoD | 3.7960 | 7.1164% | 4.95s | 5.6669 | 9.24450% | 15.71s |
| | | SHIELD | 3.7910 | 6.9758% | 5.69s | 5.6204 | 8.33150% | 17.62s |
| | OVRPL | OR-tools | 4.0512 | - | 1m 13s | 6.1671 | - | 2m 49s |
| | | POMO-MTVRP | 4.2591 | 5.1319% | 2.40s | 6.6124 | 7.36220% | 8.73s |
| | | MVMoE | 4.2415 | 4.7335% | 3.44s | 6.5529 | 6.37850% | 12.06s |
| | | MVMoE-Light | 4.2534 | 5.0199% | 3.44s | 6.5929 | 7.01370% | 10.93s |
| | | MVMoE-Deeper | 4.2251 | 4.2921% | 9s | - | - | - |
| | | SHIELD-MoD | 4.1890 | 3.4200% | 5.25s | 6.4510 | 4.74400% | 18.25s |
| | | SHIELD | 4.1754 | 3.0814% | 6.08s | 6.4068 | 4.00120% | 20.56s |
| Out-task | VRPBL | OR-tools | 5.1909 | - | 1m 13s | 7.6594 | - | 2m 33s |
| | | POMO-MTVRP | 5.3371 | 2.8163% | 2.29s | 7.8639 | 2.75730% | 7.42s |
| | | MVMoE | 5.3057 | 2.2667% | 3.22s | 7.8236 | 2.23730% | 10.92s |
| | | MVMoE-Light | 5.3141 | 2.4299% | 3.24s | 7.8552 | 2.64200% | 8.95s |
| | | MVMoE-Deeper | 5.3113 | 2.3187% | 8.85s | - | - | - |
| | | SHIELD-MoD | 5.2689 | 1.5530% | 4.80s | 7.7445 | 1.20130% | 15.81s |
| | | SHIELD | 5.2597 | 1.3699% | 5.35s | 7.7291 | 0.98540% | 17.58s |
| | VRPBTW | OR-tools | 8.0886 | - | 1m 19s | 14.1676 | - | 2m 49s |
| | | POMO-MTVRP | 8.8803 | 9.7879% | 2.77s | 15.3142 | 8.49130% | 8.73s |
| | | MVMoE | 8.8044 | 9.0737% | 3.92s | 15.2694 | 8.15410% | 12.06s |
| | | MVMoE-Light | 8.8173 | 9.1602% | 3.83s | 15.3335 | 8.61220% | 10.93s |
| | | MVMoE-Deeper | 8.8276 | 9.1366% | 9.49s | - | - | - |
| | | SHIELD-MoD | 8.7607 | 8.4928% | 5.69s | 15.1656 | 7.47000% | 18.25s |
| | | SHIELD | 8.7359 | 8.1860% | 6.29s | 15.0792 | 6.77220% | 20.56s |
| | VRPLTW | OR-tools | 8.1532 | - | 1m 23s | 13.9665 | - | 2m 52s |
| | | POMO-MTVRP | 8.5131 | 4.4146% | 2.85s | 14.4345 | 3.63460% | 10.13s |
| | | MVMoE | 8.4670 | 4.0004% | 3.95s | 14.3885 | 3.26980% | 14.01s |
| | | MVMoE-Light | 8.4753 | 4.0737% | 3.97s | 14.4490 | 3.68940% | 12.49s |
| | | MVMoE-Deeper | 8.4764 | 3.9635% | 10.08s | - | - | - |
| | | SHIELD-MoD | 8.4117 | 3.2956% | 5.84s | 14.2713 | 2.45880% | 20.38s |
| | | SHIELD | 8.3926 | 3.0792% | 6.55s | 14.2191 | 2.05060% | 23.08s |

**SW24978 — Right block**

| Problem | Solver | MTMDVRP50 Obj | Gap | Time | MTMDVRP100 Obj | Gap | Time |
|---|---|---|---|---|---|---|---|
| VRPL | OR-tools | 6.7721 | - | 1m 10s | 10.3234 | - | 2m 38s |
| | POMO-MTVRP | 6.8296 | 0.8497% | 2.43s | 10.2774 | -0.4057% | 8.00s |
| | MVMoE | 6.7881 | 0.2730% | 3.32s | 10.2491 | -0.6711% | 11.64s |
| | MVMoE-Light | 6.7941 | 0.3548% | 3.60s | 10.2727 | -0.4486% | 9.87s |
| | MVMoE-Deeper | 6.7762 | 0.0947% | 8.45s | - | - | - |
| | SHIELD-MoD | 6.7623 | -0.1163% | 5.04s | 10.1749 | -1.4005% | 17.44s |
| | SHIELD | 6.7533 | -0.2187% | 5.78s | 10.1589 | -1.5678% | 19.74s |
| VRPTW | OR-tools | 8.3232 | - | 1m 17s | 13.3531 | - | 2m 36s |
| | POMO-MTVRP | 8.6793 | 6.1415% | 2.88s | 14.4825 | 8.5406% | 9.26s |
| | MVMoE | 8.6465 | 5.7162% | 3.86s | 14.4327 | 8.1655% | 13.26s |
| | MVMoE-Light | 8.6542 | 5.8053% | 3.97s | 14.4849 | 8.5695% | 11.65s |
| | MVMoE-Deeper | 8.6081 | 5.2484% | 9.91s | - | - | - |
| | SHIELD-MoD | 8.5945 | 5.0838% | 5.81s | 14.3061 | 7.2326% | 19.33s |
| | SHIELD | 8.5729 | 4.8030% | 6.55s | 14.2664 | 6.9055% | 22.03s |
| OVRPTW | OR-tools | 5.2057 | - | 1m 11s | 8.4320 | - | 2m 42s |
| | POMO-MTVRP | 5.5109 | 5.8629% | 2.77s | 9.0652 | 7.6573% | 8.81s |
| | MVMoE | 5.5111 | 5.8568% | 3.84s | 9.0238 | 7.1596% | 12.81s |
| | MVMoE-Light | 5.5221 | 6.0649% | 3.87s | 9.0742 | 7.7730% | 10.96s |
| | MVMoE-Deeper | 5.4697 | 5.1083% | 10.39s | - | - | - |
| | SHIELD-MoD | 5.4527 | 4.7333% | 5.85s | 8.9004 | 5.7226% | 18.90s |
| | SHIELD | 5.4445 | 4.6356% | 6.67s | 8.8643 | 5.2987% | 21.56s |
| OVRPBL | OR-tools | 3.5320 | - | 1m 8s | 5.2096 | - | 2m 36s |
| | POMO-MTVRP | 3.8526 | 9.0782% | 2.36s | 5.8816 | 12.9788% | 7.40s |
| | MVMoE | 3.8357 | 8.5557% | 3.41s | 5.8117 | 11.6128% | 10.78s |
| | MVMoE-Light | 3.8564 | 9.1395% | 3.21s | 5.8700 | 12.7548% | 9.04s |
| | MVMoE-Deeper | 3.8095 | 7.8556% | 8.34s | - | - | - |
| | SHIELD-MoD | 3.7870 | 7.2199% | 5.08s | 5.6836 | 9.1828% | 16.07s |
| | SHIELD | 3.7777 | 6.9288% | 5.68s | 5.6304 | 8.1570% | 18.05s |
| OVRPBTW | OR-tools | 5.1779 | - | 1m 8s | 8.4308 | - | 2m 41s |
| | POMO-MTVRP | 5.7623 | 11.2859% | 2.80s | 9.4775 | 12.5687% | 8.48s |
| | MVMoE | 5.7434 | 10.9114% | 3.88s | 9.4291 | 11.9843% | 11.52s |
| | MVMoE-Light | 5.7527 | 11.0847% | 3.83s | 9.4815 | 12.6198% | 10.50s |
| | MVMoE-Deeper | 5.7286 | 10.6357% | 10.09s | - | - | - |
| | SHIELD-MoD | 5.6986 | 10.0178% | 5.84s | 9.3156 | 10.6768% | 17.99s |
| | SHIELD | 5.6828 | 9.7971% | 6.64s | 9.2617 | 10.0060% | 20.27s |
| OVRPLTW | OR-tools | 5.1469 | - | 1m 15s | 8.4292 | - | 2m 49s |
| | POMO-MTVRP | 5.4627 | 6.1352% | 2.81s | 9.0423 | 7.4062% | 9.20s |
| | MVMoE | 5.4512 | 5.9274% | 4.01s | 9.0008 | 6.9011% | 12.92s |
| | MVMoE-Light | 5.4605 | 6.0997% | 3.80s | 9.0501 | 7.4929% | 11.48s |
| | MVMoE-Deeper | 5.4396 | 5.6877% | 10.51s | - | - | - |
| | SHIELD-MoD | 5.4001 | 4.9193% | 6.01s | 8.8812 | 5.5065% | 19.38s |
| | SHIELD | 5.3864 | 4.7154% | 6.84s | 8.8487 | 5.1093% | 22.00s |
| VRPBLTW | OR-tools | 8.1677 | - | 1m 22s | 13.6276 | - | 2m 49s |
| | POMO-MTVRP | 8.9615 | 9.7182% | 2.96s | 14.7188 | 8.3795% | 9.28s |
| | MVMoE | 8.8913 | 9.0623% | 3.96s | 14.6809 | 8.0858% | 12.80s |
| | MVMoE-Light | 8.8890 | 9.0142% | 3.79s | 14.7363 | 8.5020% | 11.41s |
| | MVMoE-Deeper | 8.9035 | 9.0086% | 9.87s | - | - | - |
| | SHIELD-MoD | 8.8410 | 8.4141% | 5.83s | 14.5760 | 7.3226% | 18.69s |
| | SHIELD | 8.8013 | 7.9695% | 6.43s | 14.5032 | 6.7412% | 20.92s |
| OVRPBLTW | OR-tools | 5.1245 | - | 1m 18s | 8.4572 | - | 2m 31s |
| | POMO-MTVRP | 5.7114 | 11.4524% | 2.88s | 9.5106 | 12.5946% | 8.87s |
| | MVMoE | 5.6997 | 11.2344% | 4.03s | 9.4532 | 11.8813% | 12.13s |
| | MVMoE-L | 5.7007 | 11.2689% | 3.86s | 9.5087 | 12.5619% | 10.84s |
| | MVMoE-Deeper | 5.6842 | 10.9230% | 10.3s | - | - | - |
| | SHIELD-MoD | 5.6483 | 10.2214% | 5.95s | 9.3427 | 10.6239% | 18.44s |
| | Ours | 5.6312 | 9.9603% | 6.75s | 9.3013 | 10.0938% | 20.74s |

Table 18: Performance of models on VM22775

**VM22775 (left block)**

| | Problem | Solver | MTMDVRP50 Obj | Gap | Time | MTMDVRP100 Obj | Gap | Time |
|---|---|---|---|---|---|---|---|---|
| In-task | CVRP | HGS | 8.2120 | - | 1m35s | 12.1714 | - | 2m15s |
| | | POMO-MTVRP | 8.2974 | 2.2454% | 4.30s | 12.5856 | 3.4193% | 8.70s |
| | | MVMoE | 8.2459 | 1.6115% | 4.29s | 12.5450 | 3.0942% | 11.37s |
| | | MVMoE-Light | 8.2554 | 1.7229% | 4.02s | 12.5657 | 3.2645% | 10.62s |
| | | MVMoE-Deeper | 8.2352 | 1.4836% | 9.66s | 12.4767 | 2.5224% | 18.05s |
| | | SHIELD-MoD | 8.2229 | 1.3205% | 5.78s | 12.4608 | 2.3879% | 20.38s |
| | | SHIELD | 8.2143 | 1.2193% | 6.48s | | | |
| | OVRP | OR-tools | 4.8138 | - | 1m7s | 7.3689 | - | 2m39s |
| | | POMO-MTVRP | 5.0672 | 5.2636% | 2.38s | 7.8238 | 6.2490% | 7.49s |
| | | MVMoE | 5.0433 | 4.7859% | 3.38s | 7.7843 | 5.7257% | 10.39s |
| | | MVMoE-Light | 5.0557 | 5.0703% | 3.04s | 7.7975 | 5.8929% | 9.38s |
| | | MVMoE-Deeper | 4.9992 | 3.8900% | 8.65s | 7.6502 | 3.8971% | 17.11s |
| | | SHIELD-MoD | 4.9870 | 3.6258% | 5.15s | 7.6047 | 3.2896% | 19.63s |
| | | SHIELD | 4.9697 | 3.2797% | 5.98s | | | |
| | VRPB | OR-tools | 6.0429 | - | 1m1s | 9.0476 | - | 2m35s |
| | | POMO-MTVRP | 6.2125 | 2.8072% | 2.23s | 9.2501 | 2.3584% | 6.74s |
| | | MVMoE | 6.1694 | 2.1187% | 3.11s | 9.2009 | 1.8153% | 9.18s |
| | | MVMoE-Light | 6.1849 | 2.3749% | 2.91s | 9.2190 | 2.0173% | 8.27s |
| | | MVMoE-Deeper | 6.1576 | 1.9315% | 6.99s | 9.1118 | 0.8220% | 15.08s |
| | | SHIELD-MoD | 6.1402 | 1.6367% | 4.69s | 9.0876 | 0.5547% | 16.97s |
| | | SHIELD | 6.1326 | 1.5173% | 5.23s | | | |
| | OVRPB | OR-tools | 3.8870 | - | 1m8s | 5.7542 | - | 2m39s |
| | | POMO-MTVRP | 4.2505 | 9.3515% | 2.42s | 6.4354 | 11.9283% | 7.05s |
| | | MVMoE | 4.2141 | 8.4012% | 3.35s | 6.3647 | 10.6831% | 9.52s |
| | | MVMoE-Light | 4.2512 | 9.3733% | 3.03s | 6.4183 | 11.6274% | 8.69s |
| | | MVMoE-Deeper | 4.1841 | 7.6443% | 7.94s | 6.2023 | 7.8597% | 15.65s |
| | | SHIELD-MoD | 4.1656 | 7.1677% | 4.95s | 6.1568 | 7.0782% | 17.63s |
| | | SHIELD | 4.1613 | 7.0535% | 5.59s | | | |
| | OVRPL | OR-tools | 4.8097 | - | 1m19s | 7.3550 | - | 2m55s |
| | | POMO-MTVRP | 5.0597 | 5.1971% | 2.54s | 7.8041 | 6.1984% | 7.94s |
| | | MVMoE | 5.0372 | 4.7388% | 3.48s | 7.7679 | 5.7154% | 10.82s |
| | | MVMoE-Light | 5.0494 | 5.0200% | 3.17s | 7.7777 | 5.8223% | 9.78s |
| | | MVMoE-Deeper | 5.0157 | 4.2837% | 8.98s | 7.6389 | 3.9467% | 17.53s |
| | | SHIELD-MoD | 4.9793 | 3.5419% | 5.26s | 7.5889 | 3.2589% | 19.97s |
| | | SHIELD | 4.9624 | 3.1907% | 6.08s | | | |
| Out-task | VRPBL | OR-tools | 6.0258 | - | 1m16s | 8.9724 | - | 2m45s |
| | | POMO-MTVRP | 6.1987 | 2.8686% | 2.45s | 9.1670 | 2.2641% | 7.42s |
| | | MVMoE | 6.1500 | 2.1044% | 3.29s | 9.1213 | 1.7682% | 9.63s |
| | | MVMoE-Light | 6.1670 | 2.3844% | 3.05s | 9.1414 | 1.9859% | 8.91s |
| | | MVMoE-Deeper | 6.1722 | 2.4288% | 8.87s | 9.0330 | 0.7832% | 15.73s |
| | | SHIELD-MoD | 6.1227 | 1.6549% | 4.78s | 9.0043 | 0.4597% | 17.66s |
| | | SHIELD | 6.1111 | 1.4675% | 5.34s | | | |
| | VRPBTW | OR-tools | 10.7055 | - | 1m23s | 18.7523 | - | 2m45s |
| | | POMO-MTVRP | 11.7038 | 9.3248% | 2.94s | 20.0852 | 7.3516% | 8.63s |
| | | MVMoE | 11.6157 | 8.7550% | 3.96s | 20.0964 | 7.4046% | 11.29s |
| | | MVMoE-Light | 11.6391 | 8.9638% | 3.62s | 20.0970 | 7.4093% | 10.65s |
| | | MVMoE-Deeper | 11.6580 | 8.8974% | 9.48s | 19.8598 | 6.1625% | 17.96s |
| | | SHIELD-MoD | 11.5819 | 8.3991% | 5.66s | 19.7931 | 5.7595% | 20.13s |
| | | SHIELD | 11.5264 | 7.8871% | 6.25s | | | |
| | VRPLTW | OR-tools | 10.6738 | - | 1m28s | 18.6939 | - | 2m49s |
| | | POMO-MTVRP | 11.1114 | 4.0993% | 2.98s | 19.1206 | 2.4516% | 9.93s |
| | | MVMoE | 11.0270 | 3.4150% | 3.95s | 19.1122 | 2.3953% | 12.81s |
| | | MVMoE-Light | 11.0698 | 3.8246% | 3.70s | 19.1288 | 2.4781% | 12.05s |
| | | MVMoE-Deeper | 11.0612 | 3.6296% | 10.03s | 18.8883 | 1.1980% | 20.02s |
| | | SHIELD-MoD | 11.0021 | 3.1596% | 5.80s | 18.8243 | 0.8404% | 22.38s |
| | | SHIELD | 10.9673 | 2.8632% | 6.50s | | | |

**VM22775 (right block)**

| Problem | Solver | MTMDVRP50 Obj | Gap | Time | MTMDVRP100 Obj | Gap | Time |
|---|---|---|---|---|---|---|---|
| VRPL | OR-tools | 8.2151 | - | 1m11s | 12.5283 | - | 2m39s |
| | POMO-MTVRP | 8.3078 | 1.1279% | 2.55s | 12.4811 | -0.3508% | 8.01s |
| | MVMoE | 8.2539 | 0.5085% | 3.33s | 12.4472 | -0.6200% | 10.84s |
| | MVMoE-Light | 8.2593 | 0.5735% | 3.12s | 12.4618 | -0.5008% | 9.88s |
| | MVMoE-Deeper | 8.2412 | 0.3507% | 8.4s | - | - | - |
| | SHIELD-MoD | 8.2272 | 0.1836% | 5.03s | 12.3739 | -1.2047% | 17.29s |
| | SHIELD | 8.2167 | 0.0535% | 5.73s | 12.3579 | -1.3374% | 19.71s |
| VRPTW | OR-tools | 10.5525 | - | 1m16s | 17.7378 | - | 6m34s |
| | POMO-MTVRP | 10.9940 | 6.2437% | 3.03s | 19.9257 | 8.4620% | 9.17s |
| | MVMoE | 10.9227 | 5.5759% | 3.81s | 19.2077 | 8.3633% | 12.03s |
| | MVMoE-Light | 10.9546 | 5.8847% | 3.55s | 19.2231 | 8.4491% | 11.34s |
| | MVMoE-Deeper | 10.8784 | 5.1612% | 9.87s | - | - | - |
| | SHIELD-MoD | 10.8878 | 5.2251% | 5.77s | 18.9746 | 7.0508% | 19.14s |
| | SHIELD | 10.8471 | 4.8543% | 6.50s | 18.9211 | 6.7445% | 21.64s |
| OVRPTW | OR-tools | 6.0966 | - | 1m19s | 10.1562 | - | 2m44s |
| | POMO-MTVRP | 6.5159 | 6.8769% | 2.93s | 10.8685 | 7.1113% | 8.74s |
| | MVMoE | 6.4816 | 6.3126% | 3.84s | 10.8348 | 6.7774% | 11.72s |
| | MVMoE-Light | 6.5058 | 6.7237% | 3.55s | 10.8369 | 6.7987% | 10.73s |
| | MVMoE-Deeper | 6.3464 | 5.6127% | 10.32s | - | - | - |
| | SHIELD-MoD | 6.4294 | 5.4647% | 5.78s | 10.6427 | 4.8878% | 18.91s |
| | SHIELD | 6.3982 | 4.9368% | 6.60s | 10.5970 | 4.4416% | 21.43s |
| OVRPBL | OR-tools | 3.8906 | - | 1m5s | 5.7679 | - | 2m35s |
| | POMO-MTVRP | 4.2454 | 9.1193% | 2.51s | 6.4539 | 11.9664% | 7.39s |
| | MVMoE | 4.2179 | 8.4004% | 3.41s | 6.3792 | 10.6523% | 9.91s |
| | MVMoE-Light | 4.2518 | 9.2892% | 3.08s | 6.4421 | 11.7448% | 9.06s |
| | MVMoE-Deeper | 4.1888 | 7.6644% | 8.33s | - | - | - |
| | SHIELD-MoD | 4.1716 | 7.2214% | 5.02s | 6.2237 | 7.9653% | 16.06s |
| | SHIELD | 4.1646 | 7.0469% | 5.67s | 6.1758 | 7.1334% | 18.05s |
| OVRPBTW | OR-tools | 6.0530 | - | 1m15s | 10.1574 | - | 2m40s |
| | POMO-MTVRP | 6.8045 | 12.4156% | 2.89s | 11.3495 | 12.3227% | 8.49s |
| | MVMoE | 6.7815 | 12.0607% | 3.90s | 11.3082 | 11.9073% | 11.10s |
| | MVMoE-Light | 6.7831 | 12.0955% | 3.69s | 11.3072 | 11.8808% | 10.30s |
| | MVMoE-Deeper | 6.7538 | 11.5779% | 10.01s | - | - | - |
| | SHIELD-MoD | 6.7272 | 11.1773% | 5.77s | 11.1283 | 10.1297% | 18.06s |
| | SHIELD | 6.6794 | 10.3960% | 6.58s | 11.0622 | 9.4637% | 20.30s |
| OVRPLTW | OR-tools | 6.0521 | - | 1m18s | 10.1576 | - | 2m50s |
| | POMO-MTVRP | 6.4593 | 6.7289% | 2.92s | 10.8730 | 7.1771% | 9.17s |
| | MVMoE | 6.4319 | 6.2823% | 4.02s | 10.8447 | 6.9019% | 12.17s |
| | MVMoE-Light | 6.4508 | 6.5993% | 3.72s | 10.8427 | 6.8716% | 11.20s |
| | MVMoE-Deeper | 6.3882 | 5.5533% | 10.45s | - | - | - |
| | SHIELD-MoD | 6.3805 | 5.4425% | 5.95s | 10.6515 | 4.9986% | 19.33s |
| | SHIELD | 6.3456 | 4.8817% | 6.79s | 10.6088 | 4.5639% | 21.87s |
| VRPBLTW | OR-tools | 10.6434 | - | 1m19s | 18.5622 | - | 2m45s |
| | POMO-MTVRP | 11.6674 | 9.6210% | 3.08s | 19.8427 | 7.1626% | 9.27s |
| | MVMoE | 11.5700 | 8.9490% | 3.98s | 19.8255 | 7.0499% | 11.88s |
| | MVMoE-Light | 11.6111 | 9.2697% | 3.64s | 19.8335 | 7.0997% | 11.22s |
| | MVMoE-Deeper | 11.6039 | 9.0243% | 9.76s | - | - | - |
| | SHIELD-MoD | 11.5523 | 8.6835% | 5.80s | 19.5935 | 5.8187% | 18.70s |
| | SHIELD | 11.4789 | 8.0243% | 6.32s | 19.5566 | 5.5655% | 20.76s |
| OVRPBLTW | OR-tools | 10.0628 | - | 1m20s | 10.0760 | - | 2m33s |
| | POMO-MTVRP | 6.8095 | 12.3158% | 2.95s | 11.3098 | 12.3540% | 8.86s |
| | MVMoE | 6.7884 | 11.9893% | 4.06s | 11.2734 | 11.9732% | 11.48s |
| | MVMoE-L | 6.7993 | 12.1789% | 3.79s | 11.2539 | 11.7919% | 10.65s |
| | MVMoE-Deeper | 6.7635 | 11.5571% | 10.19s | - | - | - |
| | MVMoD | 6.7314 | 11.0334% | 5.90s | 11.0888 | 10.1594% | 18.44s |
| | Ours | 6.6856 | 10.3143% | 6.66s | 11.0178 | 9.4527% | 20.64s |

Table 19: Performance of models on EG7146

**Left table (EG7146 — In-task / Out-task)**

| Category | Problem | Solver | MTMDVRP50 Obj | Gap | Time | MTMDVRP100 Obj | Gap | Time |
|---|---|---|---|---|---|---|---|---|
| In-task | CVRP | HGS | 4.2261 | - | 1m 21s | 6.3233 | - | 2m 15s |
| | | POMO-MTVRP | 4.3335 | 2.6537% | 3.33s | 6.6029 | 4.7559% | 9.03s |
| | | MVMoE | 4.3018 | 2.0324% | 4.32s | 6.6075 | 4.8078% | 12.39s |
| | | MVMoE-Light | 4.3061 | 2.1268% | 4.17s | 6.6246 | 5.0781% | 11.98s |
| | | MVMoE-Deeper | 4.3061 | 2.1625% | 9.68s | - | - | - |
| | | SHIELD-MoD | 4.2876 | 1.6642% | 5.83s | 6.5535 | 3.9363% | 18.16s |
| | | SHIELD | 4.2802 | 1.4656% | 6.49s | 6.5367 | 3.6566% | 22.93s |
| | OVRP | OR-tools | 2.4397 | - | 1m 20s | 3.7510 | - | 2m 42s |
| | | POMO-MTVRP | 2.6045 | 6.7560% | 2.91s | 4.1674 | 11.8018% | 8.09s |
| | | MVMoE | 2.5861 | 6.2931% | 3.66s | 4.1673 | 11.8226% | 11.08s |
| | | MVMoE-Light | 2.5995 | 6.8360% | 3.26s | 4.1427 | 11.1366% | 10.74s |
| | | MVMoE-Deeper | 2.5787 | 6.0468% | 8.89s | - | - | - |
| | | SHIELD-MoD | 2.5514 | 4.7885% | 5.15s | 4.0474 | 8.4444% | 17.25s |
| | | SHIELD | 2.5357 | 4.1187% | 6.16s | 3.9849 | 6.7276% | 19.88s |
| | VRPB | OR-tools | 3.3731 | - | 1m 1s | 4.9564 | - | 2m 40s |
| | | POMO-MTVRP | 3.4892 | 3.4424% | 2.62s | 5.1741 | 4.6788% | 7.01s |
| | | MVMoE | 3.4641 | 2.8546% | 3.03s | 5.1634 | 4.4185% | 9.74s |
| | | MVMoE-Light | 3.4676 | 2.9329% | 3.02s | 5.1815 | 4.8007% | 9.44s |
| | | MVMoE-Deeper | 3.4652 | 2.8993% | 7.11s | - | - | - |
| | | SHIELD-MoD | 3.4455 | 2.2692% | 4.70s | 5.1077 | 3.2931% | 15.18s |
| | | SHIELD | 3.4347 | 1.9133% | 5.30s | 5.0930 | 3.0192% | 17.34s |
| | OVRPB | OR-tools | 2.0569 | - | 1m 20s | 3.0546 | - | 2m 41s |
| | | POMO-MTVRP | 2.2657 | 10.1491% | 2.76s | 3.5751 | 17.7022% | 7.32s |
| | | MVMoE | 2.2547 | 9.7586% | 3.32s | 3.5857 | 18.0985% | 10.33s |
| | | MVMoE-Light | 2.2747 | 10.7739% | 3.09s | 3.5722 | 17.6061% | 9.84s |
| | | MVMoE-Deeper | 2.2469 | 9.2368% | 7.97s | - | - | - |
| | | SHIELD-MoD | 2.2204 | 8.1100% | 4.90s | 3.4462 | 13.3146% | 15.74s |
| | | SHIELD | 2.2093 | 7.5305% | 5.59s | 3.3731 | 10.9531% | 17.60s |
| | OVRPL | OR-tools | 2.4504 | - | 1m 16s | 3.7508 | - | 2m 49s |
| | | POMO-MTVRP | 2.6115 | 6.5748% | 2.69s | 4.1693 | 11.8376% | 8.44s |
| | | MVMoE | 2.5969 | 6.2734% | 3.54s | 4.1683 | 11.8336% | 11.71s |
| | | MVMoE-Light | 2.6038 | 6.5720% | 3.33s | 4.1360 | 10.9600% | 11.30s |
| | | MVMoE-Deeper | 2.6103 | 6.5254% | 9.09s | - | - | - |
| | | SHIELD-MoD | 2.5630 | 4.7895% | 5.24s | 4.0473 | 8.4154% | 17.64s |
| | | SHIELD | 2.5450 | 4.0585% | 6.12s | 3.9838 | 6.7253% | 20.22s |
| Out-task | VRPBL | OR-tools | 3.2954 | - | 1m 19s | 4.9569 | - | 2m 33s |
| | | POMO-MTVRP | 3.4085 | 3.4311% | 2.55s | 5.1859 | 4.9025% | 7.72s |
| | | MVMoE | 3.3857 | 2.8847% | 3.32s | 5.1710 | 4.5591% | 10.50s |
| | | MVMoE-Light | 3.3891 | 2.9609% | 3.21s | 5.1941 | 5.0288% | 10.31s |
| | | MVMoE-Deeper | 3.4121 | 3.5403% | 8.91s | - | - | - |
| | | SHIELD-MoD | 3.3661 | 2.2564% | 4.78s | 5.1138 | 3.3937% | 15.87s |
| | | SHIELD | 3.3562 | 1.9297% | 5.33s | 5.0982 | 3.0999% | 18.12s |
| | VRPBTW | OR-tools | 4.7375 | - | 1m 23s | 7.9075 | - | 2m 43s |
| | | POMO-MTVRP | 5.1863 | 9.4734% | 3.03s | 8.6547 | 10.1123% | 8.67s |
| | | MVMoE | 5.1460 | 8.9711% | 4.00s | 8.6541 | 10.0303% | 12.47s |
| | | MVMoE-Light | 5.1448 | 8.9284% | 3.67s | 8.6629 | 10.1730% | 13.64s |
| | | MVMoE-Deeper | 5.1886 | 9.5212% | 9.51s | - | - | - |
| | | SHIELD-MoD | 5.1189 | 8.3411% | 5.58s | 8.5744 | 9.0841% | 18.31s |
| | | SHIELD | 5.1109 | 8.2421% | 6.15s | 8.5188 | 8.3280% | 21.80s |
| | VRPLTW | OR-tools | 4.8841 | - | 1m 31s | 8.0086 | - | 2m 55s |
| | | POMO-MTVRP | 5.1422 | 5.2845% | 3.11s | 8.4323 | 5.8016% | 10.10s |
| | | MVMoE | 5.0992 | 4.6517% | 4.12s | 8.4420 | 5.8510% | 14.30s |
| | | MVMoE-Light | 5.0994 | 4.6053% | 3.85s | 8.4599 | 6.1180% | 16.23s |
| | | MVMoE-Deeper | 5.1307 | 5.0499% | 10.22s | - | - | - |
| | | SHIELD-MoD | 5.0942 | 4.3019% | 5.80s | 8.3590 | 4.8426% | 20.45s |
| | | SHIELD | 5.0728 | 4.1259% | 6.94s | 8.3220 | 4.4071% | 24.94s |

**Right table (EG7146)**

| Problem | Solver | MTMDVRP50 Obj | Gap | Time | MTMDVRP100 Obj | Gap | Time |
|---|---|---|---|---|---|---|---|
| VRPL | OR-tools | 4.2562 | - | 1m 2s | 6.5015 | - | 2m 41s |
| | POMO-MTVRP | 4.3245 | 1.6041% | 2.93s | 6.5822 | 1.3993% | 8.39s |
| | MVMoE | 4.2965 | 1.0675% | 3.39s | 6.5868 | 1.4509% | 11.30s |
| | MVMoE-Light | 4.2979 | 1.0892% | 3.26s | 6.6053 | 1.7299% | 11.48s |
| | MVMoE-Deeper | 4.2990 | 1.1466% | 8.49s | - | - | - |
| | SHIELD-MoD | 4.2801 | 0.6453% | 5.07s | 6.5317 | 0.6028% | 17.47s |
| | SHIELD | 4.2717 | 0.4317% | 5.79s | 6.5171 | 0.3611% | 21.76s |
| VRPTW | OR-tools | 4.8840 | - | 1m 28s | 7.5872 | - | 6m 35s |
| | POMO-MTVRP | 5.1345 | 6.7583% | 3.36s | 8.3451 | 10.3102% | 9.29s |
| | MVMoE | 5.1021 | 6.1431% | 3.94s | 8.3413 | 10.1853% | 13.57s |
| | MVMoE-Light | 5.1049 | 6.1902% | 3.66s | 8.3665 | 10.5592% | 15.44s |
| | MVMoE-Deeper | 5.0940 | 6.0413% | 10.01s | - | - | - |
| | SHIELD-MoD | 5.1510 | 5.4674% | 5.74s | 8.2620 | 9.1747% | 19.75s |
| | SHIELD | 5.0787 | 5.6905% | 6.69s | 8.2334 | 8.7933% | 23.82s |
| OVRPTW | OR-tools | 3.0238 | - | 1m 40s | 4.9353 | - | 2m 50s |
| | POMO-MTVRP | 3.2700 | 8.1407% | 3.23s | 5.4417 | 10.9588% | 9.20s |
| | MVMoE | 3.2627 | 8.1766% | 3.99s | 5.4753 | 11.5536% | 14.07s |
| | MVMoE-Light | 3.2622 | 8.0924% | 3.56s | 5.4714 | 11.5304% | 13.93s |
| | MVMoE-Deeper | 3.2479 | 7.7094% | 10.52s | - | - | - |
| | SHIELD-MoD | 3.2360 | 7.0192% | 5.78s | 5.3584 | 9.1885% | 20.05s |
| | SHIELD | 3.2229 | 6.8535% | 6.77s | 5.3254 | 8.5817% | 23.67s |
| OVRPBL | OR-tools | 2.0523 | - | 1m 9s | 3.0685 | - | 2m 33s |
| | POMO-MTVRP | 2.2616 | 10.1999% | 2.63s | 3.5984 | 17.9380% | 7.70s |
| | MVMoE | 2.2526 | 9.8491% | 3.45s | 3.6028 | 18.0853% | 10.66s |
| | MVMoE-Light | 2.2652 | 10.5639% | 3.13s | 3.5860 | 17.5421% | 10.34s |
| | MVMoE-Deeper | 2.2397 | 9.1323% | 8.34s | - | - | - |
| | SHIELD-MoD | 2.2165 | 8.1623% | 5.01s | 3.4627 | 13.3097% | 16.21s |
| | SHIELD | 2.2037 | 7.5294% | 5.70s | 3.3874 | 10.8413% | 18.12s |
| OVRPBTW | OR-tools | 2.9200 | - | 1m 16s | 4.8008 | - | 2m 49s |
| | POMO-MTVRP | 3.2772 | 12.2321% | 3.09s | 5.5019 | 15.1401% | 8.85s |
| | MVMoE | 3.2692 | 12.1102% | 3.93s | 5.5239 | 15.5221% | 13.05s |
| | MVMoE-Light | 3.2664 | 12.0087% | 3.71s | 5.5144 | 15.3594% | 12.82s |
| | MVMoE-Deeper | 3.2752 | 12.1652% | 10.15s | - | - | - |
| | SHIELD-MoD | 3.2366 | 10.8410% | 5.80s | 5.4285 | 13.5184% | 18.52s |
| | SHIELD | 3.2274 | 10.7363% | 6.67s | 5.3650 | 12.2104% | 22.24s |
| OVRPLTW | OR-tools | 2.9926 | - | 1m 21s | 4.8134 | - | 2m 58s |
| | POMO-MTVRP | 3.2366 | 8.1519% | 3.09s | 5.3253 | 11.2289% | 9.66s |
| | MVMoE | 3.2305 | 8.2202% | 4.14s | 5.3475 | 11.6426% | 14.57s |
| | MVMoE-Light | 3.2343 | 8.2613% | 3.72s | 5.3524 | 11.7726% | 14.39s |
| | MVMoE-Deeper | 3.2433 | 8.3770% | 10.71s | - | - | - |
| | SHIELD-MoD | 3.2093 | 7.2411% | 5.95s | 5.2409 | 9.4280% | 20.35s |
| | SHIELD | 3.1930 | 6.9771% | 6.93s | 5.2088 | 8.7900% | 24.35s |
| VRPBLTW | OR-tools | 4.7699 | - | 1m 25s | 7.9676 | - | 2m 44s |
| | POMO-MTVRP | 5.2290 | 9.6259% | 3.23s | 8.6980 | 9.8474% | 9.28s |
| | MVMoE | 5.1827 | 8.9705% | 3.95s | 8.7020 | 9.8280% | 13.11s |
| | MVMoE-Light | 5.1899 | 9.1190% | 3.70s | 8.7208 | 10.0636% | 14.68s |
| | MVMoE-Deeper | 5.2269 | 9.5803% | 9.83s | - | - | - |
| | SHIELD-MoD | 5.1630 | 8.4985% | 5.65s | 8.6340 | 8.9862% | 18.95s |
| | SHIELD | 5.1542 | 8.3490% | 6.38s | 8.5772 | 8.2720% | 22.41s |
| OVRPBLTW | OR-tools | 2.9427 | - | 1m 25s | 4.8417 | - | 2m 39s |
| | POMO-MTVRP | 3.3049 | 12.3088% | 3.12s | 5.5503 | 15.2090% | 9.19s |
| | MVMoE | 3.3005 | 12.3844% | 4.02s | 5.5658 | 15.4365% | 13.38s |
| | MVMoE-L | 3.3004 | 12.3136% | 3.78s | 5.5611 | 15.4145% | 13.35s |
| | MVMoE-Deeper | 3.3061 | 12.3491% | 10.39s | 1.0000% | - | - |
| | MVMoD | 3.2702 | 11.1276% | 5.86s | 5.4697 | 13.4759% | 19.02s |
| | Ours | 3.2579 | 10.9948% | 6.79s | 5.4196 | 12.4838% | 22.03s |

Table 20: Performance of models on FI10639

**In-task**

| Problem | Solver | MTMDVRP50 Obj | Gap | Time | MTMDVRP100 Obj | Gap | Time |
|---|---|---|---|---|---|---|---|
| CVRP | HGS | 7.1789 | - | 1m21s | 10.6055 | - | 2m11s |
| | POMO-MTVRP | 7.2316 | 2.2536% | 3.18s | 10.9689 | 3.4421% | 8.69s |
| | MVMoE | 7.1891 | 1.6553% | 4.09s | 10.9438 | 3.2108% | 11.58s |
| | MVMoE-Light | 7.1959 | 1.7537% | 4.07s | 10.9590 | 3.3535% | 10.52s |
| | MVMoE-Deeper | 7.1775 | 1.4944% | 9.66s | 10.8778 | 2.5840% | 18.05s |
| | SHIELD-MoD | 7.1675 | 1.3514% | 5.97s | 10.8700 | 2.5115% | 20.48s |
| | SHIELD | 7.1578 | 1.2110% | 6.45s | | | |
| OVRP | OR-tools | 4.3654 | - | 1m7s | 6.6709 | - | 2m37s |
| | POMO-MTVRP | 4.5669 | 4.6148% | 2.32s | 7.0708 | 6.0585% | 7.44s |
| | MVMoE | 4.5476 | 4.7107% | 3.30s | 7.0215 | 5.3245% | 10.18s |
| | MVMoE-Light | 4.5643 | 4.5705% | 3.29s | 7.0384 | 5.5842% | 9.33s |
| | MVMoE-Deeper | 4.5261 | 3.6903% | 8.81s | 6.9334 | 4.0124% | 17.02s |
| | SHIELD-MoD | 4.5059 | 3.2248% | 5.29s | 6.8809 | 3.2150% | 19.42s |
| | SHIELD | 4.4901 | 2.8598% | 5.96s | | | |
| VRPB | OR-tools | 5.5089 | - | 1m3s | 8.2519 | - | 2m35s |
| | POMO-MTVRP | 5.6511 | 2.5818% | 2.19s | 8.4295 | 2.2114% | 6.73s |
| | MVMoE | 5.6148 | 1.9523% | 3.02s | 8.3929 | 1.7773% | 9.21s |
| | MVMoE-Light | 5.6260 | 2.1609% | 2.97s | 8.4094 | 1.9787% | 8.29s |
| | MVMoE-Deeper | 5.6035 | 1.7363% | 7.05s | 8.3212 | 0.9074% | 15.16s |
| | SHIELD-MoD | 5.5876 | 1.4523% | 4.70s | 8.3038 | 0.6926% | 16.92s |
| | SHIELD | 5.5745 | 1.2165% | 5.24s | | | |
| OVRPB | OR-tools | 3.8078 | - | 1m11s | 5.6014 | - | 2m38s |
| | POMO-MTVRP | 4.1457 | 8.8747% | 2.27s | 6.2396 | 11.4105% | 6.99s |
| | MVMoE | 4.1234 | 8.2539% | 3.32s | 6.1759 | 10.2748% | 9.40s |
| | MVMoE-Light | 4.1465 | 8.8749% | 3.03s | 6.2219 | 11.0947% | 8.70s |
| | MVMoE-Deeper | 4.0969 | 7.5936% | 8.01s | 6.0640 | 8.2826% | 15.65s |
| | SHIELD-MoD | 4.0743 | 6.9982% | 4.98s | 6.0129 | 7.3669% | 17.60s |
| | SHIELD | 4.0687 | 6.8295% | 5.62s | | | |
| OVRPL | OR-tools | 4.3703 | - | 1m15s | 6.6913 | - | 2m50s |
| | POMO-MTVRP | 4.5739 | 4.6592% | 2.38s | 7.0941 | 6.0842% | 7.85s |
| | MVMoE | 4.5514 | 4.1338% | 3.45s | 7.0483 | 5.4115% | 10.77s |
| | MVMoE-Light | 4.5653 | 4.4587% | 3.18s | 7.0665 | 5.6736% | 9.79s |
| | MVMoE-Deeper | 4.5394 | 3.8684% | 9.01s | 6.9520 | 3.9647% | 17.40s |
| | SHIELD-MoD | 4.5080 | 3.1491% | 5.24s | 6.9121 | 3.3698% | 19.82s |
| | SHIELD | 4.4958 | 2.8730% | 6.02s | | | |
| VRPBL | OR-tools | 5.4775 | - | 1m13s | 8.2521 | - | 2m35s |
| | POMO-MTVRP | 5.6231 | 2.6583% | 2.27s | 8.4291 | 2.2085% | 7.43s |
| | MVMoE | 5.5861 | 2.0085% | 3.25s | 8.3967 | 1.8150% | 9.92s |
| | MVMoE-Light | 5.5972 | 2.2190% | 3.06s | 8.3967 | 1.9562% | 8.92s |
| | MVMoE-Deeper | 5.5920 | 2.0900% | 8.88s | 8.3250 | 0.9400% | 15.80s |
| | SHIELD-MoD | 5.5587 | 1.5040% | 4.76s | 8.3091 | 0.7540% | 17.63s |
| | SHIELD | 5.5482 | 1.3205% | 5.34s | | | |
| VRPBTW | OR-tools | 8.3979 | - | 1m21s | 14.4940 | - | 2m46s |
| | POMO-MTVRP | 9.2380 | 10.0037% | 2.74s | 15.6453 | 8.1734% | 8.55s |
| | MVMoE | 9.1706 | 9.3132% | 3.87s | 15.6160 | 7.9340% | 11.31s |
| | MVMoE-Light | 9.1749 | 9.3735% | 3.64s | 15.6476 | 8.1580% | 11.25s |
| | MVMoE-Deeper | 9.1749 | 9.2528% | 9.49s | 15.4787 | 7.0071% | 17.85s |
| | SHIELD-MoD | 9.1328 | 8.8544% | 5.61s | 15.4095 | 6.5168% | 20.27s |
| | SHIELD | 9.0873 | 8.3285% | 6.19s | | | |
| VRPLTW | OR-tools | 8.5328 | - | 1m23s | 14.5948 | - | 2m46s |
| | POMO-MTVRP | 8.9175 | 4.5081% | 2.79s | 15.0812 | 3.5072% | 9.85s |
| | MVMoE | 8.8754 | 4.0915% | 3.90s | 15.0478 | 3.2597% | 12.90s |
| | MVMoE-Light | 8.8878 | 4.2350% | 3.76s | 15.0779 | 3.4699% | 12.45s |
| | MVMoE-Deeper | 8.8705 | 3.9576% | 10.12s | 14.9107 | 2.3365% | 19.73s |
| | SHIELD-MoD | 8.8237 | 3.4896% | 5.71s | 14.8641 | 2.0135% | 22.55s |
| | SHIELD | 8.8021 | 3.2202% | 6.43s | | | |

**Out-task**

| Problem | Solver | MTMDVRP50 Obj | Gap | Time | MTMDVRP100 Obj | Gap | Time |
|---|---|---|---|---|---|---|---|
| VRPL | OR-tools | 7.2655 | - | 1m11s | 11.0647 | - | 2m39 |
| | POMO-MTVRP | 7.3195 | 0.7427% | 2.44s | 10.9764 | -0.7391% | 8.01s |
| | MVMoE | 7.2732 | 0.1525% | 3.28s | 10.9476 | -0.9920% | 10.90s |
| | MVMoE-Light | 7.2799 | 0.2467% | 3.10s | 10.9619 | -0.8662% | 9.84s |
| | MVMoE-Deeper | 7.2567 | -0.0743% | 8.41s | | | |
| | SHIELD-MoD | 7.2485 | -0.1881% | 5.03s | 10.8826 | -1.5856% | 17.38s |
| | SHIELD | 7.2411 | -0.2947% | 5.74s | 10.8762 | -1.6419% | 19.71s |
| VRPTW | OR-tools | 8.6076 | - | 1m21s | 13.8303 | - | 6m32s |
| | POMO-MTVRP | 8.9835 | 6.1814% | 2.92s | 14.9881 | 8.4109% | 9.10s |
| | MVMoE | 8.9383 | 5.6687% | 3.77s | 14.9514 | 8.1368% | 12.12s |
| | MVMoE-Light | 8.9575 | 5.8823% | 3.63s | 14.9753 | 8.3111% | 12.44s |
| | MVMoE-Deeper | 8.9071 | 5.2897% | 9.91s | | | |
| | SHIELD-MoD | 8.8903 | 5.0787% | 5.70s | 14.8289 | 7.2670% | 18.93s |
| | SHIELD | 8.8706 | 4.8594% | 6.40s | 14.7707 | 6.8428% | 21.86s |
| OVRPTW | OR-tools | 5.5367 | - | 1m14s | 9.0269 | - | 2m44s |
| | POMO-MTVRP | 5.8542 | 5.7348% | 2.76s | 9.6772 | 7.2601% | 8.81s |
| | MVMoE | 5.8494 | 5.6378% | 3.81s | 9.6439 | 6.8831% | 11.85s |
| | MVMoE-Light | 5.8618 | 5.8692% | 3.57s | 9.6658 | 7.1204% | 11.19s |
| | MVMoE-Deeper | 5.8129 | 4.9966% | 10.41s | | | |
| | SHIELD-MoD | 5.8005 | 4.7565% | 5.83s | 9.4988 | 5.2872% | 18.86s |
| | SHIELD | 5.7889 | 4.5616% | 6.65s | 9.4746 | 5.0130% | 21.82s |
| OVRPBL | OR-tools | 3.7943 | - | 1m6s | 5.6060 | - | 2m35s |
| | POMO-MTVRP | 4.1217 | 8.6277% | 2.37s | 6.2443 | 11.3928% | 7.36s |
| | MVMoE | 4.1042 | 8.1365% | 3.40s | 6.1755 | 10.1709% | 9.77s |
| | MVMoE-Light | 4.1345 | 8.9410% | 3.11s | 6.2269 | 11.0891% | 9.01s |
| | MVMoE-Deeper | 4.0782 | 7.4829% | 8.35s | | | |
| | SHIELD-MoD | 4.0743 | 6.9563% | 5.04s | 6.0646 | 8.1956% | 16.03s |
| | SHIELD | 4.0511 | 6.7413% | 5.69s | 6.0163 | 7.3310% | 17.97s |
| OVRPBTW | OR-tools | 5.4856 | - | 1m14s | 9.0376 | - | 2m41s |
| | POMO-MTVRP | 6.0902 | 11.0224% | 2.75s | 10.1308 | 12.1891% | 8.48s |
| | MVMoE | 6.0783 | 10.7968% | 3.89s | 10.0940 | 11.7631% | 11.14s |
| | MVMoE-Light | 6.0859 | 10.9503% | 3.73s | 10.1191 | 12.0410% | 10.96s |
| | MVMoE-Deeper | 6.0537 | 10.3571% | 10.07s | | | |
| | SHIELD-MoD | 6.0311 | 9.9335% | 5.84s | 9.9550 | 10.2390% | 18.01s |
| | SHIELD | 6.0170 | 9.7035% | 6.57s | 9.9070 | 9.7087% | 20.45s |
| OVRPLTW | OR-tools | 5.5178 | - | 1m16s | 9.0627 | - | 2m51s |
| | POMO-MTVRP | 5.8370 | 5.7846% | 2.81s | 9.7068 | 7.1746% | 9.21s |
| | MVMoE | 5.8256 | 5.5633% | 4.09s | 9.6680 | 6.7480% | 12.31s |
| | MVMoE-Light | 5.8413 | 5.8436% | 3.72s | 9.6930 | 7.0206% | 11.58s |
| | MVMoE-Deeper | 5.8108 | 5.3092% | 10.6s | | | |
| | SHIELD-MoD | 5.7726 | 4.5990% | 5.97s | 9.5282 | 5.2071% | 19.32s |
| | SHIELD | 5.7654 | 4.4862% | 6.80s | 9.4963 | 4.8548% | 22.25s |
| VRPBLTW | OR-tools | 8.4892 | - | 1m23s | 14.3715 | - | 2m41s |
| | POMO-MTVRP | 9.3172 | 9.7539% | 2.94s | 15.5583 | 8.4426% | 9.20s |
| | MVMoE | 9.2484 | 9.0613% | 4.01s | 15.5064 | 8.0744% | 11.78s |
| | MVMoE-Light | 9.2586 | 9.1808% | 3.74s | 15.5511 | 8.3871% | 12.02s |
| | MVMoE-Deeper | 9.2484 | 8.9437% | 9.59s | | | |
| | SHIELD-MoD | 9.2078 | 8.5868% | 5.72s | 15.3800 | 7.2121% | 18.51s |
| | SHIELD | 9.1680 | 8.0919% | 6.28s | 15.3116 | 6.7035% | 20.90s |
| OVRPBLTW | OR-tools | 5.4777 | - | 1m19s | 9.0148 | - | 2m33s |
| | POMO-MTVRP | 6.0851 | 11.0885% | 2.86s | 10.1094 | 12.2402% | 8.87s |
| | MVMoE | 6.0701 | 10.8055% | 4.02s | 10.0706 | 11.8008% | 11.52s |
| | MVMoE-L | 6.0786 | 10.9700% | 3.79s | 10.0989 | 12.1151% | 10.88s |
| | MVMoE-Deeper | 6.0498 | 10.4445% | 10.31s | | | |
| | MVMoD | 6.0241 | 9.9658% | 5.94s | 9.9412 | 10.3847% | 18.39s |
| | Ours | 6.0146 | 9.8151% | 6.69s | 9.9001 | 9.9104% | 20.80s |

Table 21: Performance of models on GR9882

**In-task (GR9882)**

| Problem | Solver | MTMDVRP50 Obj | Gap | Time | MTMDVRP100 Obj | Gap | Time |
|---|---|---|---|---|---|---|---|
| CVRP | HGS | 6.9560 | - | 1m17s | 10.3936 | - | 2m13s |
| | POMO-MTVRP | 7.1084 | 2.1913% | 3.02s | 10.7649 | 3.6221% | 8.77s |
| | MVMoE | 7.0647 | 1.5660% | 4.72s | 10.7410 | 3.3953% | 11.51s |
| | MVMoE-Light | 7.0754 | 1.7233% | 3.89s | 10.7575 | 3.5564% | 10.66s |
| | MVMoE-Deeper | 7.0566 | 1.4537% | 9.65s | - | - | - |
| | SHIELD-MoD | 7.0445 | 1.2709% | 5.76s | 10.6632 | 2.6404% | 18.03s |
| | SHIELD | 7.0360 | 1.1565% | 6.47s | 10.6622 | 2.6295% | 20.47s |
| OVRP | OR-tools | 4.2741 | - | 1m9s | 6.4873 | - | 2m37s |
| | POMO-MTVRP | 4.4856 | 4.9486% | 2.32s | 6.9236 | 6.8885% | 7.50s |
| | MVMoE | 4.4670 | 4.5352% | 3.62s | 6.8612 | 5.9006% | 10.23s |
| | MVMoE-Light | 4.4821 | 4.9079% | 3.06s | 6.8992 | 6.5072% | 9.43s |
| | MVMoE-Deeper | 4.4342 | 3.7705% | 8.84s | - | - | - |
| | SHIELD-MoD | 4.4165 | 3.3587% | 5.13s | 6.7528 | 4.2265% | 17.09s |
| | SHIELD | 4.4039 | 3.0663% | 5.99s | 6.7109 | 3.5776% | 19.54s |
| VRPB | OR-tools | 5.3878 | - | 1m2s | 7.9488 | - | 2m43s |
| | POMO-MTVRP | 5.5305 | 2.6488% | 2.12s | 8.1316 | 2.3515% | 6.73s |
| | MVMoE | 5.4960 | 2.0273% | 3.22s | 8.1031 | 1.9936% | 9.11s |
| | MVMoE-Light | 5.5070 | 2.2479% | 3.05s | 8.1145 | 2.1470% | 8.28s |
| | MVMoE-Deeper | 5.4825 | 1.7840% | 7.04s | - | - | - |
| | SHIELD-MoD | 5.4659 | 1.4692% | 4.67s | 8.0045 | 0.7585% | 14.99s |
| | SHIELD | 5.4560 | 1.2933% | 5.25s | 8.0003 | 0.7073% | 16.98s |
| OVRPB | OR-tools | 3.6601 | - | 2s | 5.3017 | - | 2m40s |
| | POMO-MTVRP | 3.9849 | 8.8728% | 2.29s | 5.9619 | 12.5598% | 7.09s |
| | MVMoE | 3.9679 | 8.3625% | 3.31s | 5.8707 | 10.7826% | 9.42s |
| | MVMoE-Light | 4.0022 | 9.3077% | 3.06s | 5.9508 | 12.3350% | 8.69s |
| | MVMoE-Deeper | 3.9357 | 7.5287% | 8.04s | - | - | - |
| | SHIELD-MoD | 3.9929 | 6.8585% | 4.94s | 5.7386 | 8.3071% | 15.66s |
| | SHIELD | 3.9116 | 6.8401% | 5.61s | 5.7054 | 7.6825% | 17.64s |
| OVRPL | OR-tools | 4.2759 | - | 1m14s | 6.4665 | - | 2m54s |
| | POMO-MTVRP | 4.4924 | 5.0627% | 2.38s | 6.9109 | 7.0167% | 7.98s |
| | MVMoE | 4.4725 | 4.6183% | 3.44s | 6.8517 | 6.0897% | 10.81s |
| | MVMoE-Light | 4.4862 | 4.9660% | 3.21s | 6.8892 | 6.6897% | 9.81s |
| | MVMoE-Deeper | 4.4528 | 4.1370% | 9.01s | - | - | - |
| | SHIELD-MoD | 4.4226 | 3.4465% | 5.24s | 6.7470 | 4.4686% | 17.46s |
| | SHIELD | 4.4093 | 3.1437% | 6.05s | 6.7033 | 3.7798% | 19.89s |
| VRPBL | OR-tools | 5.4044 | - | 1m13s | 7.9259 | - | 2m46s |
| | POMO-MTVRP | 5.5466 | 2.6310% | 2.27s | 8.0977 | 2.2328% | 7.44s |
| | MVMoE | 5.5124 | 2.0251% | 3.26s | 8.0624 | 1.7844% | 9.60s |
| | MVMoE-Light | 5.5290 | 2.3316% | 3.07s | 8.0825 | 2.0334% | 8.91s |
| | MVMoE-Deeper | 5.5167 | 2.0785% | 8.91s | - | - | - |
| | SHIELD-MoD | 5.4844 | 1.4986% | 4.74s | 7.9756 | 0.6815% | 15.70s |
| | SHIELD | 5.4701 | 1.2376% | 5.32s | 7.9657 | 0.5610% | 17.65s |
| VRPBTW | OR-tools | 8.5591 | - | 1m23s | 14.7076 | - | 2m42s |
| | POMO-MTVRP | 9.3818 | 9.6117% | 2.77s | 15.8813 | 8.2210% | 8.55s |
| | MVMoE | 9.3229 | 9.0631% | 3.88s | 15.8378 | 7.8885% | 11.38s |
| | MVMoE-Light | 9.3409 | 9.2382% | 3.63s | 15.8744 | 8.1585% | 10.80s |
| | MVMoE-Deeper | 9.3261 | 8.9607% | 9.47s | - | - | - |
| | SHIELD-MoD | 9.2801 | 8.5275% | 5.58s | 15.6726 | 6.7998% | 17.77s |
| | SHIELD | 9.2497 | 8.1686% | 6.16s | 15.6150 | 6.3862% | 20.07s |
| VRPLTW | OR-tools | 8.7717 | - | 1m24s | 14.6818 | - | 2m49s |
| | POMO-MTVRP | 9.1521 | 4.3371% | 2.81s | 15.1587 | 3.4803% | 9.87s |
| | MVMoE | 9.1039 | 3.8812% | 3.93s | 15.1196 | 3.1988% | 13.51s |
| | MVMoE-Light | 9.1157 | 4.0234% | 3.73s | 15.1365 | 3.3103% | 12.05s |
| | MVMoE-Deeper | 9.0936 | 3.6702% | 10.08s | - | - | - |
| | SHIELD-MoD | 9.0525 | 3.2890% | 5.73s | 14.9741 | 2.2352% | 19.73s |
| | SHIELD | 9.0343 | 3.0922% | 6.40s | 14.9220 | 1.8556% | 22.28s |

**Out-task**

| Problem | Solver | MTMDVRP50 Obj | Gap | Time | MTMDVRP100 Obj | Gap | Time |
|---|---|---|---|---|---|---|---|
| VRPL | OR-tools | 7.0566 | - | 1m8s | 10.9621 | - | 2m39s |
| | POMO-MTVRP | 7.1025 | 0.6507% | 2.35s | 10.8673 | -0.8707% | 8.00s |
| | MVMoE | 7.0583 | 0.0504% | 3.42s | 10.8343 | -1.1747% | 10.86s |
| | MVMoE-Light | 7.0674 | 0.1778% | 3.13s | 10.8499 | -1.0253% | 9.87s |
| | MVMoE-Deeper | 7.0458 | -0.1276% | 8.4s | - | - | - |
| | SHIELD-MoD | 7.0342 | -0.2962% | 5.00s | 10.7588 | -1.8647% | 17.28s |
| | SHIELD | 7.0267 | -0.4024% | 5.73s | 10.7533 | -1.9215% | 19.75s |
| VRPTW | OR-tools | 8.7191 | - | 1m18s | 14.1579 | - | 6m33s |
| | POMO-MTVRP | 9.0838 | 6.0783% | 2.81s | 15.3650 | 8.6007% | 9.14s |
| | MVMoE | 9.0412 | 5.5405% | 3.81s | 15.3199 | 8.2577% | 12.73s |
| | MVMoE-Light | 9.0580 | 5.7197% | 3.56s | 15.3400 | 8.3956% | 11.40s |
| | MVMoE-Deeper | 9.0011 | 5.0722% | 9.93s | - | - | - |
| | SHIELD-MoD | 8.9955 | 4.9831% | 5.68s | 15.1639 | 7.1856% | 18.96s |
| | SHIELD | 8.9728 | 4.7571% | 6.39s | 15.1141 | 6.8234% | 21.64s |
| OVRPTW | OR-tools | 5.3713 | - | 1m14s | 8.7285 | - | 2m43s |
| | POMO-MTVRP | 5.6981 | 6.0840% | 2.74s | 9.3763 | 7.5389% | 8.85s |
| | MVMoE | 5.6898 | 5.9100% | 3.86s | 9.3344 | 7.0460% | 11.74s |
| | MVMoE-Light | 5.7075 | 6.2320% | 3.58s | 9.3682 | 7.4403% | 10.87s |
| | MVMoE-Deeper | 5.6383 | 4.9754% | 10.44s | - | - | - |
| | SHIELD-MoD | 5.6333 | 4.8490% | 5.81s | 9.1843 | 5.3396% | 18.81s |
| | SHIELD | 5.6179 | 4.5997% | 6.65s | 9.1565 | 5.0259% | 21.63s |
| OVRPBL | OR-tools | 3.6489 | - | 1m6s | 5.3628 | - | 2m35s |
| | POMO-MTVRP | 3.9788 | 9.0419% | 2.37s | 6.0290 | 12.5224% | 7.41s |
| | MVMoE | 3.9540 | 8.3113% | 3.42s | 5.9425 | 10.8406% | 9.80s |
| | MVMoE-Light | 3.9894 | 9.3004% | 3.12s | 6.0122 | 12.1885% | 9.05s |
| | MVMoE-Deeper | 3.9219 | 7.4804% | 8.33s | - | - | - |
| | SHIELD-MoD | 3.9083 | 7.1102% | 5.02s | 5.7983 | 8.1741% | 16.03s |
| | SHIELD | 3.9036 | 6.9255% | 5.69s | 5.7719 | 7.6776% | 18.07s |
| OVRPBTW | OR-tools | 5.3443 | - | 1m15s | 8.7557 | - | 2m38s |
| | POMO-MTVRP | 5.9343 | 11.0402% | 2.79s | 9.8206 | 12.5485% | 8.55s |
| | MVMoE | 5.9228 | 10.7895% | 3.87s | 9.7818 | 12.0651% | 11.11s |
| | MVMoE-Light | 5.9307 | 10.9458% | 3.76s | 9.8090 | 12.3796% | 10.40s |
| | MVMoE-Deeper | 5.8931 | 10.2686% | 10.09s | - | - | - |
| | SHIELD-MoD | 5.8669 | 9.7525% | 5.86s | 9.6345 | 10.4030% | 18.04s |
| | SHIELD | 5.8538 | 9.5448% | 6.57s | 9.5922 | 9.9189% | 20.40s |
| OVRPLTW | OR-tools | 5.4180 | - | 1m17s | 8.7467 | - | 2m50s |
| | POMO-MTVRP | 5.7484 | 6.0986% | 2.83s | 9.3965 | 7.5557% | 9.20s |
| | MVMoE | 5.7388 | 5.9032% | 4.01s | 9.3546 | 7.0644% | 12.16s |
| | MVMoE-Light | 5.7578 | 6.2420% | 3.74s | 9.3821 | 7.3816% | 11.28s |
| | MVMoE-Deeper | 5.7069 | 5.3321% | 10.62s | - | - | - |
| | SHIELD-MoD | 5.6815 | 4.8271% | 5.97s | 9.2118 | 5.4452% | 19.31s |
| | SHIELD | 5.6673 | 4.6124% | 6.81s | 9.1826 | 5.1003% | 22.02s |
| VRPBLTW | OR-tools | 8.5652 | - | 1m22s | 14.8707 | - | 2m43s |
| | POMO-MTVRP | 9.4066 | 9.8231% | 2.94s | 16.0036 | 0.0008% | 9.19s |
| | MVMoE | 9.3398 | 9.1785% | 3.92s | 15.9774 | 7.7261% | 12.09s |
| | MVMoE-Light | 9.3566 | 9.3505% | 3.66s | 16.0050 | 7.9140% | 11.49s |
| | MVMoE-Deeper | 9.3414 | 9.0618% | 9.64s | - | - | - |
| | SHIELD-MoD | 9.3011 | 8.7148% | 5.69s | 15.8097 | 6.6217% | 18.45s |
| | SHIELD | 9.2614 | 8.2500% | 6.30s | 15.7598 | 6.2376% | 20.62s |
| OVRPBLTW | OR-tools | 5.3472 | - | 1m18s | 8.7637 | - | 2m35s |
| | POMO-MTVRP | 5.9479 | 11.2345% | 2.90s | 9.8478 | 12.4518% | 8.93s |
| | MVMoE | 5.9443 | 11.0920% | 4.00s | 9.8019 | 11.9205% | 11.47s |
| | MVMoE-L | 5.9446 | 11.1131% | 3.84s | 9.8301 | 12.2460% | 10.77s |
| | MVMoE-Deeper | 5.9096 | 10.5184% | 10.35s | - | - | - |
| | MVMoD | 5.8912 | 10.0965% | 5.97s | 9.6544 | 10.2485% | 18.44s |
| | Ours | 5.8696 | 9.7509% | 6.69s | 9.6219 | 9.8720% | 20.74s |

