# OpenReview forum: "SHIELD: Multi-task Multi-distribution Vehicle Routing Solver with Sparsity & Hierarchy in Efficiently Layered Decoder"
_ICLR.cc/2025/Conference — Submitted to ICLR 2025_

### Official Review · Reviewer_wrAC · 2024-11-01

**Soundness:** 2
**Presentation:** 3
**Contribution:** 3
**Rating:** 6
**Confidence:** 3

**Summary:**

This paper propose a variant setting of VRP，which is called Multi-Task Multi-Distribution VRP (MTMDVRP). And it introduces an impressive learning model SHIELD, a neural solver that leverages sparsity through a customized NCO decoder with MoD layers and hierarchy through context-based cluster representation. At the same time, necessary experiments are conducted to support the model performance.

**Strengths:**

The authors of this paper consider the generalization of tasks and distribution in VRP at the same time, which is a good attempt. In addition, they proposed the innovative network learning architecture SHIELD, introduced the clustering layer to enhance the hierarchical expression ability of the model, and added the MoD layer in the decoding to take into account the sparsity, which is impressive in the field of machine learning to solve combinatorial optimization problems.

**Weaknesses:**

The new setting proposed in this paper adds additional expectations for distribution. I think this is not a very unique innovation, and it is essentially using different distribution instances of non-uniform distributions during training, which seems to naturally enable the model to perform well on other non-uniform distributions during the test stage. If it is only trained on uniform distribution, it will be more impressive to generalize to non-uniform distributions during the test phase. There is a crucial weakness that the solution time is not reflected in the experiment, which makes it difficult to understand the computational efficiency of the algorithm. What’s more, the baselines compared in the experiment are a bit small. There have been many works on VRP in top conferences in recent years, but they are not reflected in the baselines in this experiment.

**Questions:**

1 What computational complexity will the introduction of MoD layers and context-based cluster representation cause? Can the time for model solution be supplemented in the experiment? This is crucial to understand the computational efficiency of the algorithm. It is unacceptable to have no time measurement at all.
2 Can the latest work on VRP solution in recent years be supplemented in the baseline in the experiment?
3 In the new setting proposed in this paper, for the expectation of distribution q, does q obey a specific distribution or have a specific classification?
4 If the model is trained only on a uniform distribution, will it still perform well on a cross-distribution test? Can this be supplemented in the experiment?

---

> ### Author Response · Authors · 2024-11-23
> **Response to Reviewer wrAC**
>
> We thank the reviewer for the positive recognition on the effectiveness of our method SHIELD as well as the contributions we have written. We would like to address your concerns as follows:
>
> **[Q1: Model runtime]:** Thanks for your comment. Actually, we reported the inference time in the tables 13-21 of Appendix A.8. Following the reviewer suggestions, we further averaged the runtime of the models reported in Appendix A.8 and updated Table 1 to reflect them. In specific, it refers to the average solving time on 1000 test instances over 9 countries. For convenience, we tabulate the number of parameters and runtimes here. From the table, we can see that utilizing SHIELD improves the inference time of a denser MVMoE-Deeper, and also renders MTMDVRP100 trainable.
>
> |Model|Num. Parameters|Runtime on MTMDVRP50|Runtime on MTMDVRP100|
> |:-:|:-:|:-:|:-:|
> |POMO-MTVRP|1.25M|2.74s|8.30s|
> |MVMoE|3.68M|3.72s|11.21s|
> |MVMoE-Light|3.70M|3.45s|10.38s|
> |MVMoE-Deeper|4.46M|9.23s|OOM|
> |SHIELD-MoD|4.37M|5.43s|17.70s|
> |SHIELD|4.59M|6.16s|20.07s|
>
> **[Q2: Other VRP Approaches]:** We acknowledge the reviewer's feedback regarding other VRP methods. However, we note that many of recent works are focused on the single-task VRP setting, and that the multi-task and single-task research directions are orthogonal to each other. Generalizing single-task VRP solvers to the Multi-task Multi-distribution setting is non-trivial, as shown in the case for POMO-MTVRP. Following the ICLR guidelines, for Multi-task VRP solvers, we have included all recent published works as baselines in the form of POMO-MTVRP and MVMoE. We have also provided multiple benchmark models that expand on MVMoE so that the overall model capacity as equivalent as possible. Please let us know if you have any other recommendations. We refer the reader to Appendix A.1 for a comprehensive literature review and discussion on neural single-task VRP solvers.
>
> **[Q3: Explanation on $\mathcal{Q}$]:** We have updated Figure 1 in the manuscript to reflect the overall process from sampling to training and finally inference. As shown, we designate USA13509, JA9847, and BM33708 as known maps (i.e., in-distribution). These form the set of distributions $\mathcal{Q}'$, which is a subset of $\mathcal{Q}$. Also, we have a set of known tasks (i.e., in-task) $\mathcal{K}'$, containing CVRP, VRPB, OVRP, VRPTW, VRPL, and OVRPTW, which is a subset of $\mathcal{K}$. At each step, we sample a map $q' \in \mathcal{Q}'$ uniformly, followed by sampling a set of coordinates (cities) from the map. Next, we sample a task $k'$ from the set of $\mathcal{K}'$. This forms a batch of training data and is passed through the model. Note that for the proposed MTMDVRP setting, we consider 9 countires map and 16 VRP tasks as $\mathcal{Q}$ and $\mathcal{K}$ in this paper, while $\mathcal{Q}$ and $\mathcal{K}$ could be any world maps or related-tasks in theory. During inference, we apply the model to the entire set of tasks $\mathcal{K}$ and distributions $\mathcal{Q}$, whereby the remaining 10 tasks and 6 distributions are considered to be out-task and out-distribution. For a formal definition, please see Section 4.1 in our revised paper.
>
> **[Q4: Training on Uniform only]:**
> We do appreciate the reviewer pointing out this key point and have further trained MTMDVRP50 models on uniform distribution, followed by evaluating them on the 9 distributions discussed in this paper. We retain the same table format as Table 1, but please note that in this case, all distributions are technically out-of-distribution, as the models are trained purely on uniform data. We exclude runtimes in the following table as we have updated Table 1 with overall runtimes for the various models. Also, due to time constraints, we were only available to train the models on MTMDVRP50 settings. As shown, when the models are not trained on any structured distributions and only on the uniform one, there is a degradation in performance all around. This necessitates the training of a sufficiently flexible model on varied and structured distributions, so as to train a strong foundation model capable of realistic applications.
>
> |||MTMDVRP50||||
> |:-:|:-:|:-:|:-:|:-:|:-:|
> ||Model|In-dist||Out-dist|
> |||Obj.|Gap|Obj.|Gap||Obj.|Gap|Obj.|Gap|
> ||POMO-MTVRP (Uniform)|6.0932|3.8834%|6.4104|4.0007%|
> ||MVMoE (Uniform)|6.0779|3.6000%|6.3930|3.6710%|
> |In-task|MVMoE-Light (Uniform)|6.0926|3.8418%|6.4061|3.8254%|
> ||MVMoE-Deeper (Uniform)|6.0580|3.1964%|6.3822|3.5062%|
> ||SHIELD-MoD (Uniform)|6.0482|3.0379%|6.3666|3.2037%|
> ||SHIELD (Uniform)|**6.0414**|**2.9223%**|**6.3596**|**3.0832%**|
> |||||||||||
> ||POMO-MTVRP (Uniform)|5.8762|8.1526%|6.2457|8.3681%|
> ||MVMoE (Uniform)|5.8602|7.7505%|6.2251|7.8788%|
> |Out-task|MVMoE-Light (Uniform)|5.8802|8.1328%|6.2414|8.0983%|
> ||MVMoE-Deeper (Uniform)|5.8292|7.0524%|6.2034|7.4642%|
> ||SHIELD-MoD (Uniform)|5.8103|6.7257%|6.1769|6.9455%|
> ||SHIELD (Uniform)|**5.8035**|**6.6394%**|**6.1712**|**6.8616%**|

---

> > ### Author Response · Authors · 2024-11-29
> >
> > Dear Reviewer wrAC,
> >
> > Thank you for the time and effort to review our manuscript. As the author-reviewer discussion period is coming to a conclusion, we hope that we have sufficiently satisfied your concerns with the work. In lieu of that, we hope that you will consider revising your score favourably. Otherwise, we are open and keen to discuss further so as to address any other issues. We sincerely thank you very much.

---

> > > ### Author Response · Authors · 2024-12-02
> > >
> > > Dear Reviewer wrAC,
> > >
> > > Thank you for the time and effort to review our manuscript. As the author-reviewer discussion period is coming to a conclusion in two days, we hope that we have sufficiently satisfied your concerns with the work. In lieu of that, we hope that you will consider revising your score favourably. Otherwise, we are open and keen to discuss further so as to address any other issues. We sincerely thank you very much.

---

### Official Review · Reviewer_Hw5y · 2024-11-03

**Soundness:** 2
**Presentation:** 2
**Contribution:** 2
**Rating:** 3
**Confidence:** 4

**Summary:**

This paper studies the multi-task multi-distribution scenario of Vehicle Routing Problems (VPRs), and propose a new method from the perspective of sparsity. Specifically, it employs the Mixture-of-Depths (MoD) architecture in the decoder of neural solvers, which can select active layers for each token. Meanwhile, inspired by previous works on the benefit of clustering nodes for instance representations, it further extends the input of the decoder with additional information of the cluster centroids. Experiments show that compared with MVMoE, a previous work on multi-task VRPs, the proposed method can achieve better performance.

**Strengths:**

The writing of this paper is generally good. Experiments demonstrates the superior performance compared with previous method MVMoE.

**Weaknesses:**

1. The motivation of this paper is weak. The main contribution of this work is the introduction of MoD to the neural solvers for VRPs. However, MoE and MoD share similar ideas from the perspective of sparsity (MoE achieves sparsity on width and MoD achieves sparsity on depth), and the advantages of employing MoD instead of MoE in sub section 4.2 are not convincing.
2. As the benefit of clustering nodes has already been discussed in previous works [1], the contribution of adapting it in the proposed method is very limited.
3. As multi-task VRPs and multi-distribution VRPs have already been studied in previous works respectively, the necessity and challenges of the proposed multi-task multi-distribution VRPs should be discussed. Meanwhile, the specific advantages of the proposed method SHIELD on the scenario of multi-task multi-distribution should be better clarified.

**Questions:**

1. As computational efficiency is one of the main advantages of the employed MoD, what about the comparison inference time?
2. Is the proposed method SHIELD tailored for the scenario of multi-task multi-distribution? What about its performance on the scenario of multi-distribution?
3. What about the performance on problem instances with more than 100 nodes?

---

> ### Author Response · Authors · 2024-11-23
> **Response to Reviewer Hw5y - Part 1**
>
> We thank the reviewer for acknowledging our good writing, and positive performance and impact of SHIELD. We understand that the main concern is regarding the motivation of MoD and MTMDVRP. We hope the following rebuttal could address your concerns.
>
> **[W1: Motivation of MoE vs MoD]:** While the reviewer highlights the differences in sparsity between MoE and MoD, we would like to clarify that the MoE mechanism used in MVMoE achieves only partial sparsity by selecting specific experts, that is, determining which sub-network to activate. Consequently, the forward computation in MVMoE requires a similar amount of compute as the POMO-MTVRP framework. This similarity arises because MVMoE replaces the dense multi-layer perceptron (MLP) blocks in POMO-MTVRP with an MoE layer, which retains the same input and output dimensions. Although only a portion of the MoE layer is activated during inference or training, the overall parameter usage remains comparable for each input token.
>
> In contrast, the MoD employed in SHIELD presents 3 layers in the decoder, each of which independently decides a fraction of the tokens to be processed. This encourages the network to prioritize tokens that are more important. Figure 3 (right panel) showcases this effect, whereby starting nodes and ending nodes in sub-routes utilize more layers, whereas those in the middle do not. As such, the computation can vary for each input token, enhancing the model's flexibility in representation learning and decision-making.
>
> Moreover, we argue that since all parameters are shared, the sparsity introduced by MoD forces the network to learn the best set of parameters that can be generalizable across all tasks and distributions. This behavior aligns with that of Information Bottleneck as mentioned in Section 4.2, where studies found that highly predictive representations with minimal complexity improved generalization as a whole, suggesting the need to balance the shared and task-specific information. Therefore, by introducing sparsity on the network through MoD, we force the network to learn a set of minimal representations capable of solving various tasks and distributions. Our empirical results in Table 2 justify the importance of enforcing sparsity: when we increase the number of tokens allowed, thereby reducing sparsity, the network's in-task in-distribution performance improves, but its OOD generalization starts to worsen. On the extreme end, in the case of MVMoE-Deeper where all tokens are available, the network has the worst performance compared to the sparser variants.
>
> **[W2: Contribution of Soft-clustering]:** While soft-clustering has been previously explored, its capabilities in learning foundation models for generalization in combinatorial optimization problems have not yet been studied, to the best of our knowledge. Note that the soft clustering approach in (Goh et el., 2024) was only centered on learning neural TSP solvers and the neural solver was trained specifically for one distribution. Differently, in our paper, we explore the generalization properties of foundation VRP models. With the introduction of the MTMDVRP scenario, we test the robustness of the soft clustering across 16 different problems and 9 distributions. Technically, we introduce the context-based prompt so as to make the soft clustering mechanism more suitable to the MTMDVRP. We believe this work is the first to identify the key benefits of having such a clustering mechanism in the space of foundational neural combinatorial optimization solvers.

---

> > ### Author Response · Authors · 2024-11-23
> > **Response to Reviewer Hw5y - Part 2**
> >
> > **[W3: Overall specific advantages of SHIELD]:** As discussed in the respone to the point W1, SHIELD presents an architecture that aims to learn a minimal representation set across both tasks and distributions. **Having a set of shared parameters and enforcing sparsity through MoD on the network forces it to learn strong representations that appear to be highly useful for both task and distribution generalization**.
> >
> > Specifically, we observe from Table 1 in the manuscript that having a deeper decoder model, as in MVMoE-Deeper, produces a better model compared to MVMoE. However, simply increasing the capacity for such an autoregressive model renders it untrainable. By introducing MoD to the decoder layers, we sparsify the compute and improve the overall speed of the model. In addition to compute benefits, MoD's inductive bias emulates the Information Bottleneck principle - there is a need to learn a minimal complex set of highly rich representations for generalization. Sparsity naturally invokes this principle by forcing the network to only process a handful of representations. Another approach to learning a minimal complex set of highly rich representations is via clustering. Here, the network is forced to condense global information into a handful of representations. The combination of these 2 contributions result in SHIELD, a flexible model tested on the MTMDVRP. We believe that all experiments holistically point towards SHIELD being the superior model in both predictive performance and generalization on the MTMDVRP.
> >
> > Moreover, SHIELD advances the recent development of foundation VRP models by embodying the underlying principle of learning general representations within a more practical MTMDVRP setting, thereby establishing it a solid foundation for future work.
> >
> > **[Q1: Runtime of inference]:** We averaged the runtime of the models reported in Appendix A.8 (Tables 13-21) and updated Table 1 to reflect them. In specific, it refers to the average solving time on 1000 test instances over 9 countries. For convenience, we tabulate the number of parameters and runtimes here. From the table, we can see that utilizing SHIELD improves the inference time of a denser MVMoE-Deeper, and also renders MTMDVRP100 trainable.
> >
> > |Model|Num. Parameters|Runtime on MTMDVRP50|Runtime on MTMDVRP100|
> > |:-:|:-:|:-:|:-:|
> > |POMO-MTVRP|1.25M|2.74s|8.30s|
> > |MVMoE|3.68M|3.72s|11.21s|
> > |MVMoE-Light|3.70M|3.45s|10.38s|
> > |MVMoE-Deeper|4.46M|9.23s|OOM|
> > |SHIELD-MoD|4.37M|5.43s|17.70s|
> > |SHIELD|4.59M|6.16s|20.07s|

---

> > > ### Author Response · Authors · 2024-11-23
> > > **Response to Reviewer Hw5y - Part 3**
> > >
> > > **[Q2: MTMDVRP vs MDVRP]:** We understand the reviewer's comment regarding the multi-distribution setting. In Table 1, generalization rightwards (in-dist vs out-dist) should cover such a case, where we have tested on the same set of tasks during training, but on the 6 out-of-distribution maps. In all cases, SHIELD presents itself as the superior model. Moreover, we have identified earlier checkpoints, SHIELD-Ep400 which denotes SHIELD at the 400th epoch, and SHIELD-Ep600 which denotes SHIELD at the 600th epoch. We find that SHIELD-Ep400 has similar in-task in-distribution performance to MVMoE, while SHIELD-Ep600 has similar in-task in-distribution performance to MVMoE-Deeper. We apply these models to our datasets and tabulated the results below. As shown, for both cases, SHIELD is still the superior model when it comes to both task and distribution generalization.
> > >
> > > |||MTMDVRP50|||||MTMDVRP100||||
> > > |:-:|:-:|:-:|:-:|:-:|:-:|:-:|:-:|:-:|:-:|:-:|
> > > ||Model|In-dist||Out-dist|||In-dist||Out-dist||
> > > |||Obj.|Gap|Obj.|Gap||Obj.|Gap|Obj.|Gap|
> > > |In-task|MVMoE|6.0557|3.1479%|6.3924|3.5071%||9.3722|3.5969%|10.0827|4.6855%|
> > > ||SHIELD-400Ep|6.0597|3.1495%|**6.3830**|**3.2730%**||9.3785|3.5993%|**10.0559**|**4.3562%**|
> > > |||||||||||
> > > |Out-task|MVMoE|5.8328|7.1553%|6.2196|7.5174%||9.3811|7.4092%|10.1665|8.5140%|
> > > ||SHIELD-400Ep|**5.8290**|**7.1064%**|**6.2085**|**7.2927%**||**9.3499**|**6.9578%**|**10.1202**|**7.8332%**|
> > >
> > >
> > > |||MTMDVRP50|||||MTMDVRP100||||
> > > |:-:|:-:|:-:|:-:|:-:|:-:|:-:|:-:|:-:|:-:|:-:|
> > > ||Model|In-dist||Out-dist|||In-dist||Out-dist||
> > > |||Obj.|Gap|Obj.|Gap||Obj.|Gap|Obj.|Gap|
> > > |In-task|MVMoE-Deeper|6.0337|2.7343%|6.3677|3.1333%||OOM|OOM|OOM|OOM|
> > > ||SHIELD-600Ep|6.0333|2.7089%|**6.3653**|**2.9993%**||**9.3194**|**2.9498%**|**10.0111**|**3.8262%**|
> > > |||||||||||
> > > |Out-task|MVMoE-Deeper|5.8206|6.7924%|6.2136|7.2962%||OOM|OOM|OOM|OOM|
> > > ||SHIELD-600Ep|**5.8039**|**6.6539%**|**6.1823**|**6.8736%**||**9.3105**|**6.4308%**|**10.0765**|**7.2594%**|
> > >
> > > Additionally, we have conducted experiments suggested by the reviewer. Specifically, we fixed the task to only CVRP, and trained all models on the 3 in-distribution maps (i.e., USA13509, JA9847, and BM33708). We then apply the models to the 6 out-distribution maps. The following table shows the overall performance across the distribution. **As shown, SHIELD displays superior performance in and out of distribution.**
> > > ||CVRP50||||
> > > |:-:|:-:|:-:|:-:|:-:|
> > > |Model|In-dist||Out-dist||
> > > ||Obj.|Gap|Obj.|Gap||Obj.|Gap|Obj.|Gap|
> > > |POMO-MTVRP|6.6511|1.2260%|6.9763|1.4689%|
> > > |MVMoE|6.6454|1.1401%|6.9709|1.3858%|
> > > |MVMoE-Light|6.6482|1.1814%|6.9723|1.4112%|
> > > |MVMoE-Deeper|6.6313|0.9207%|6.9628|1.2731%|
> > > |SHIELD-MOD|6.6284|0.8798%|6.9552|1.1623%|
> > > |SHIELD|**6.6269**|**0.8570%**|**6.9474**|**1.0338%**|
> > >
> > > **[Q3: Performance on larger problems]:** We recognize the reviewer's concern regarding generalization to large-scale problem instances. To address this, we evaluate our trained models MTMDVRP100 to Set-X in the CVRPLib instances. Set-X-1 contains problems of various distributions and sizes, ranging from 101 nodes to 251 nodes (28 instances). Set-X-2 contains problems of various distributions and sizes, ranging from 502 nodes to 1001 nodes (32 instances). Below are the average performance. Note that since we are using MTMDVRP100 models, MVMoE-Deeper could not be run as it is too large to be trained. We also include the average score from SHIELD-Ep400, an earlier training checkpoint where the in-task in-distribution performance is similar to that of MVMoE. As shown, SHIELD outperforms all baselines on large-scale instances. Additionally, SHIELD-Ep400 outperforms its MVMoE counterpart.
> > >
> > > |Set-X-1||POMO-MTVRP||MVMoE||MVMoE-Light||SHIELD-MoD||SHIELD||SHIELD-Ep400||
> > > |:-:|:-:|:-:|:-:|:-:|:-:|:-:|:-:|:-:|:-:|:-:|:-:|:-:|:-:|
> > > ||Opt.|Obj.|Gap|Obj.|Gap|Obj.|Gap|Obj.|Gap|Obj.|Gap|Obj.|Gap|
> > > |Avg.|31280|33601|7.4148%|33111|6.0845%|33174|6.1773%|32902|5.1979%|**32703**|**4.6437%**|32897|5.3961%
> > >
> > > |Set-X-2||POMO-MTVRP||MVMoE||MVMoE-Light||SHIELD-MoD||SHIELD||SHIELD-Ep400||
> > > |:-:|:-:|:-:|:-:|:-:|:-:|:-:|:-:|:-:|:-:|:-:|:-:|:-:|:-:|
> > > ||Opt.|Obj.|Gap|Obj.|Gap|Obj.|Gap|Obj.|Gap|Obj.|Gap|Obj.|Gap|
> > > |Avg.|101874|115725|14.0802%|116136|14.9631%|114225|12.7539%|111534|9.8527%|**111598**|**9.8164%**|111905|10.0618%|

---

> ### Comment · Reviewer_Hw5y · 2024-11-28
>
> Thank you to the authors for the responses. I do appreciate the authors’ effort during the discussion phase, but I still hold the opinion in the initial comments. Here are the detailed reasons that I decide to give my rating as 3.
>
> 1. As the generalization performance of neural solvers has always been an important issue in the community, both multi-task generalization and multi-distribution generalization have been studied before. I do not think that highlighting the scenario of multi-task multi-distribution generalization could become one of the strengths for a high-quality paper. Meanwhile, I do not think that there are some specific designs and advantages of the proposed method on the multi-task multi-distribution. In other words, it seems that employing MoD to potentially emphasizing important nodes, and employing soft-clustering can also benefit multi-scale or multi-distribution scenarios as well.
>
> 2. The main ideas in the proposed methods provide rare new insights. I agree that MoE and MoD are achieving sparsity from different perspectives, but I do not think that this can be regarded as a main reason for that MoD should be better. Even that there are some intuitive explanations with Figure 3 and Figure 4, the results are not significant and convincing enough. For example, in figure 3, the explanations about right panel are very weak. In figure 4, it is hard for me to recognize the relationship among 3 maps on emphasized nodes. Therefore, I tend to regard this part as an application of existing method MoD with weak motivation. Besides, with the consensus that the idea of soft-clustering has been studied before, I do not think that extending it to CVRP problems and multi-task multi-distribution scenarios with rare new findings is very interesting.
>
> 3. The supplemented results on set-X are very limited, which is only a simple test of the pretrained models on MTMDVRP. As there are various powerful solvers that can achieve better performance on set-X, more comparison with them under the same training settings would be interesting.

---

> > ### Author Response · Authors · 2024-11-29
> > **Response to Official Comment by Reviewer Hw5y - Part 1**
> >
> > Dear Reviewer #Hw5y,
> >
> > We thank you for your response and sharing details for your scores. We provide our responses to your points as follows.
> >
> > 1. While generalization has been an important issue within the community, most of the solutions provided involve fine-tuning a pre-trained model towards the task [1]. This still suggests that one requires data from that specific task, and that their proposed approaches are efficient architectures that can be fine-tuned quickly to them. POMO-MTVRP and MVMoE were the first to present training approaches and architectures for the multi-task setting. In the space of multi-distribution settings, previous works still focus on uniformly distributed data [2], whereby for clustered distributions, the number of clusters are specified **beforehand**. Also, more recent works involve some form of meta learning to quickly adapt a trained model towards various distributions or to train from scratch but with considerably more sample complexity [3]. While they are impactful works, their underlying distributions do not reflect real-world patterns or scenarios, with [3] stating the need for a sufficient variety in the data for any form of generalization (340 different Gaussian mixtures). Additionally, both [2,3] propose *frameworks* for enabling some form of cross-distribution / cross-task generalization, whereby the underlying models are still POMO at heart. In the case of the multi-task foundation model in [5, 6], the goal is to produce **new architectures capable of learning all forms of tasks**. In this work, we first introduce a significantly more practical scenario, where we ask models to cater their parameters to **both** tasks and distributions **concurrently**. Then, we identify **key aspects of generalization** and show which architectural changes can be made that **promote these aspects**. Ideally, a true Foundation Model should be trained on a variety of data, such as tasks, distributions, sizes, and more. In this work, we make a step forward by transitioning the multi-task scenario to multi-task multi-distributions, whereby the distributions are not arbitrary - they reflect a typical scenario that companies face. We believe that a model capable of addressing the large variety of the problem has to be sufficiently robust and that the significant strides made in our model as compared to the benchmarks showcase that.
> >
> > 2. While we both agree that MoE and MoD achieve sparsity differently, we respectfully disagree that it is not the main reason why MoD is better in this scenario. MoE, in MVMoE's case, provides multiple alternative paths in the decoder. This asks the networks to utilize the **same number of parameters as POMO-MTVRP**, just **different sets** of them to provide some flexibility in the network. Intuitively, this can provide different outputs based on different tasks, and empirical results shows that its beneficial for the multi-task scenario. In contrast, MoD asks the network to **select the required compute** by forcing only a limited number of tokens to work on. This means that if a node that is selected is important, it might get processed more by the network. Figure 3 shows this exact behavior, where **crucial decision-making steps** such as near the start or end of the sub-route, which could adversely affect the route length (as shown in the left panel of Figure 3), requires more overall compute for the token. As for Figure 4, we believe that both BM33708 and VM22775 share a similar right end tail in the distribution, with some heavy density towards the top and the right of the map, whereas for SW24978, its skewed towards the bottom left of the map. Looking at the overall. We numbered each node in sequential order and in an anti-clockwise fashion such that they are as close to each other in physical positions on the grid. Based on this, a plot of the layer use of each node during the total solving process highlights how similarly positioned nodes display some familiar patterns in the overall compute used.

---

> ### Author Response · Authors · 2024-11-29
> **Response to Official Comment by Reviewer Hw5y - Part 2**
>
> 3. While there are powerful solvers that can achieve better performance on Set-X, the models here are trained on a **multitude of different tasks and distributions**. This set of parameters do well on our scenario, **in addition to** performing well on a form of size generalization. Additionally, the feedback proposed to us in general was regarding size generalization, which we interpret to be an upward generalization in overall size. As such, we believe that the performance on Set-X, which contain most of the larger problems in CVRPLib, would satisfy such a scenario. In fact, powerful solvers such as LEHD, fail to train with reinforcement in a reasonable amount of time, as described in their paper [4]. Couple their high training cost with the addition of a multi-task multi-distribution scenario would result in a large increase in training time as convergence gets significantly harder due to the complexity of the scenario.
>
> [1] Lin, Zhuoyi, et al. "Cross-problem learning for solving vehicle routing problems." arXiv preprint arXiv:2404.11677 (2024).
>
> [2] Bi, Jieyi, et al. "Learning generalizable models for vehicle routing problems via knowledge distillation." Advances in Neural Information Processing Systems 35 (2022): 31226-31238.
>
> [3] Zhou, Jianan, et al. "Towards omni-generalizable neural methods for vehicle routing problems." International Conference on Machine Learning. PMLR, 2023.
>
> [4] Luo, Fu, et al. "Neural combinatorial optimization with heavy decoder: Toward large scale generalization." Advances in Neural Information Processing Systems 36 (2023): 8845-8864.
>
> [5] Liu, Fei, et al. "Multi-task learning for routing problem with cross-problem zero-shot generalization." Proceedings of the 30th ACM SIGKDD Conference on Knowledge Discovery and Data Mining. 2024.
>
> [6] Zhou, Jianan, et al. "MVMoE: Multi-Task Vehicle Routing Solver with Mixture-of-Experts." arXiv preprint arXiv:2405.01029 (2024).

---

> > ### Author Response · Authors · 2024-12-02
> >
> > Dear Reviewer #Hw5y,
> >
> > Thank you for your time and effort in reviewing our manuscript and rebuttals. As the author-reviewer discussion is ending, we hope that our clarifications and revisions have addressed your concerns effectively. In light of that, we hope you can reconsider your evaluation. We are open and keen to discuss any remaining concerns and questions you might have. We sincerely thank you very much.

---

### Official Review · Reviewer_n7rT · 2024-11-03

**Soundness:** 2
**Presentation:** 2
**Contribution:** 3
**Rating:** 3
**Confidence:** 4

**Summary:**

The paper introduces a new setting for the classic VRP problem called MTMDVRP and the proposes SHIELD, a novel architecture for solving the problem. SHIELD makes use of recent deep learning architecture to enforce sparsity in the network. Although the proposed method achieved good results on several dataset, the main claims are somewhat unjustified.

**Strengths:**

- The paper makes use of recent advances in deep learning architecture in VRP, an important problem.
- The paper proposes the multi-distribution setting for VRP. However, the lack of discussion leaves me uncertain about the importance of the new setting.

**Weaknesses:**

- The multi-distribution setting is not described and discussed in enough detail although it is the main contribution of this paper. Why MTMD is important? How the distribution set $\mathcal{Q}$ is created and sampled? Where is the MD part in the training of SHIELD?
- The benefits of MoD is hypothetical (line 285-288). The author did not provide evidence to their claims of the benefits of MoD. If they could show evidence of under- and over-processing of tokens lead to bad results, the paper would be stronger.
- The finding in this paper seems to contradict that of MoD ((Raposo et al., 2024), the paper on which this paper is based on but the author didn't discuss it. Line 188 states that MoD makes the performance slightly worse but this paper report possitive result with MoD.
- The reason for choosing the 9 countries is not stated in the paper. What is the performance of the proposed method on other countries in the dataset?
- It is unclear that the proposed method really improves OOD generalization. What I can see from the tables is that SHIELD performs well InD so it performs well OOD. Other methods seem to have the about same OOD - InD performance gap with SHIELD (table 1). From the data, it is hard to conclude that MoD improves OOD generalization.
- There is no comparison on model size and speed. The author showed that by increasing the size of MVMoE, they can get better results. A natural questions is that whether the performance improvement of SHIELD simply due to its higher capacity?
- The writing is not good. There are many hard to understand statements and incorrect uses of words. Instead of using complex words, I think the author should use simple language to clearly explain their ideas.

**Questions:**

Please address the questions and weaknesses in the previous section.

---

> ### Author Response · Authors · 2024-11-23
> **Response to Reviewer n7rT - Part 1**
>
> We thank the reviewer for the valuable feedback which helps a lot to further strengthen our work. We understand the main concern about the importance of the MTMDVRP setting and we hope our response below with new empirical results would clear your concerns.
>
> **[W1: More details about MTMDVRP]:** Great feedback! We have revised the manuscript to address this in Section 4.1. The MTMDVRP scenario is particularly significant since it **1) reflects a more practical application encountered by the industry and 2) represents a critical step toward developing foundation models for academic research**.
>
> From the application perspective, suppose a well-known logistic company X has established its presence in a handful of countries/cities. It is able to train a model based on collected data across these countries/cities. Now, if company X wishes to expand its market to newer ones, it definitely will face data from new distributions, and potentially faces new tasks. As such, it is highly beneficial that its trained model is able to be quickly applied to new incoming data - suggesting the need for the model to be robust to new unseen tasks and distributions. We note that while the original MTVRP setting has taken steps toward learning a unified model for various VRP variants, all customers within the instances are uniformly generated. This does not reflect practical distributions, where customer locations are following underlying patterns within a country/city.
>
> From the machine learning perspective, foundation models typically exhibit similar behavior, whereby they are singular models that have learnt general representations capable of diverse applications across tasks and distributions. The MTMDVRP scenario is a crucial scenario that exhibits such challenges, the models trained have to be sufficiently flexible to generalize across not only tasks but also distributions. As explained before, facing new tasks and distributions is a very practical scenario that industrial companies encounter.
>
> **This is the MTMDVRP scenario we are introducing** - *from 9 countries, we sub-select 3 countries and denote them as the ones that company X has established presence (in our work, we use USA13509, JA9847, and BM33708 during training). Then, we challenge the models to train only on a subset of tasks (i.e., CVRP, OVRP, VRPB, VRPL, VRPTW, OVRPTW) drawn from these 3 distributions, and observe their OOD performances on data drawn from the remaining 6 distributions and 10 tasks. As such, we can observe how the models generalize across either task, distribution, or both.*
>
> **[W2: Benefits of MoD]:** Thanks for your comment. We understand the reviewer's concerns regarding the overall benefits of MoD. We have opted to remove the lines 285-288, and added a more detailed analysis of our results in lines 420-428. Specifically, in our experiments in Table 1 and ablation study in Table 2 of the manuscript, we have presented the results of varying the processing level. We note that the combined MVMoE-Deeper and MVMoE in Table 1 actually represent two extreme cases in model design, illustrating the trade-off between over-processing and under-processing tokens. In specific, MVMoE-Deeper has a decoder with the same number of layers as SHIELD and SHIELD-MoD (i.e., without clustering), but all tokens are needed to be processed at every layer (the case of over-processing). In contrast, MVMoE has only a single decoder layer that processes tokens (under-processing). These models exemplify the scenarios mentioned by the reviewer: over-processing (MVMoE-Deeper) and under-processing (MVMoE). Our method, positioned between these extremes, processes only a subset of tokens in each layer, with this operation performed independently for every layer for flexibility.
>
> From Table 1, we observe that in general, MVMoE-Deeper is a superior model to MVMoE, suggesting that increasing the capacity of the decoder is beneficial to the model. Unfortuately, a consequence of doing so renders MVMoE-Deeper to be untrainable on larger instances. SHIELD presents itself as compromise between the two ends, we opt for a larger capacity decoder but force the network to *learn* which tokens should be processed more frequently. Consequently, we see that both SHIELD and SHIELD-MoD are capable of overall stronger in-task in-distribution performance as shown in Table 1. Apart from that, both models also shows stronger generalization properties across both tasks. Furthermore, the results in Table 2 of the manuscript further demonstrate how the number of tokens processed (i.e., processing level) at each layer affects model performance. Increasing processing power improves in-task, and in-distribution performance but comes at the cost of reduced generalization. There is also a turning point at SHIELD(30%) whereby the model starts to deterioriate as a whole both in and out of task and distributions. Please see our analyses in Section 5.2.

---

> > ### Author Response · Authors · 2024-11-23
> > **Response to Reviewer n7rT - Part 2**
> >
> > **[W3: Findings of MoD]:** Thank you for your valuable comment. We acknowledge that the wording in line 188 was inaccurate and have revised the manuscript accordingly. We now directly cite Raposo et al., *"Despite aggressive routing around the blocks, transformers are able to achieve performance improvements relative to baselines."* This highlights that MoD enhances the baseline model's overall performance in the original MoD paper. Our empirical findings are fully aligned with this observation, as demonstrated in Table 1 of the manuscript, where sparsity improves the model's performance in the in-task, in-distribution setting. Furthermore, our experiments reveal an additional benefit: the model also achieves improved OOD generalization.
> >
> > **[W4: Selection of dataset]:** Thanks for your comment. The dataset consists of 9 representative countries from the National TSP collection of 29 different countries. These instances were retrieved from the WorldTSP and were presented as part of a large-scale TSP challenge. Note that our model was only trained on 3 out of 9 distributions, and both the in-distribution performance on these 3, and zero-shot out-of-distribution generalization performance on the remaining 6 distributions, were tested and reported in the original paper. The reported performance in Table 1 of the manuscript represents aggregated averaged performance across these distributions. Detailed results for each individual distribution are provided in Tables 13 through Table 21 (9 tables for each) in Appendix A.13.
> >
> > In selecting the 3 datasets to be trained on, we chose them based on the density of the map (BM33708 is highly dense), spread of the map (USA13509 is spread out throughout the entire area), and unique structure of the map (JA9847 has a very specific structure and concentration). Apart from that, the remaining 6 exhibit similar features as well, as shown in Figure 5 in Appendix A.13. By having such a varied set of distributions to train on, we hope that the model learns to process and exploit them. The benefits of having such varied distributions in training is further supported by our response in W5 & W6 below, where we evaluate the trained models on CVRPLib instances spanning various sizes and distributions.

---

> > > ### Author Response · Authors · 2024-11-23
> > > **Response to Reviewer n7rT - Part 3**
> > >
> > > **[W5 & W6: OOD Generalization]:** Insightful comment! We agree that additional experiments are needed to determine whether our proposed method improves OOD generalization or the observed improvements are simply due to increased model capacity.
> > >
> > > To evaluate OOD generalization more rigorously, we identified earlier training phases whereby SHIELD matches the in-distribution performance of the baseline models and then test the OOD performance. SHIELD-Ep400 and SHIELD-Ep600 are checkpoints at epochs 400 and 600 respectively, where we find that SHIELD-Ep400 matches the performance of MVMoE, and SHIELD-Ep600 matches the performance of MVMoE-Deeper (*the model with the same capacity as SHIELD*), at both MTMDVRP50 and MTMDVRP100. We find that when their in-task in-distribution performances are similar, our model displays **superior** OOD generalization. Here, we provide two tables for your reference.
> > >
> > > |||MTMDVRP50|||||MTMDVRP100||||
> > > |:-:|:-:|:-:|:-:|:-:|:-:|:-:|:-:|:-:|:-:|:-:|
> > > ||Model|In-dist||Out-dist|||In-dist||Out-dist||
> > > |||Obj.|Gap|Obj.|Gap||Obj.|Gap|Obj.|Gap|
> > > |In-task|MVMoE|6.0557|3.1479%|6.3924|3.5071%||9.3722|3.5969%|10.0827|4.6855%|
> > > ||SHIELD-400Ep|6.0597|3.1495%|**6.3830**|**3.2730%**||9.3785|3.5993%|**10.0559**|**4.3562%**|
> > > |||||||||||
> > > |Out-task|MVMoE|5.8328|7.1553%|6.2196|7.5174%||9.3811|7.4092%|10.1665|8.5140%|
> > > ||SHIELD-400Ep|**5.8290**|**7.1064%**|**6.2085**|**7.2927%**||**9.3499**|**6.9578%**|**10.1202**|**7.8332%**|
> > >
> > >
> > > |||MTMDVRP50|||||MTMDVRP100||||
> > > |:-:|:-:|:-:|:-:|:-:|:-:|:-:|:-:|:-:|:-:|:-:|
> > > ||Model|In-dist||Out-dist|||In-dist||Out-dist||
> > > |||Obj.|Gap|Obj.|Gap||Obj.|Gap|Obj.|Gap|
> > > |In-task|MVMoE-Deeper|6.0337|2.7343%|6.3677|3.1333%||OOM|OOM|OOM|OOM|
> > > ||SHIELD-600Ep|6.0333|2.7089%|**6.3653**|**2.9993%**||**9.3194**|**2.9498%**|**10.0111**|**3.8262%**|
> > > |||||||||||
> > > |Out-task|MVMoE-Deeper|5.8206|6.7924%|6.2136|7.2962%||OOM|OOM|OOM|OOM|
> > > ||SHIELD-600Ep|**5.8039**|**6.6539%**|**6.1823**|**6.8736%**||**9.3105**|**6.4308%**|**10.0765**|**7.2594%**|
> > >
> > > Clearly, SHIELD presents itself as a superior architecture compared to their equivalent counterparts.
> > >
> > > Additionally, we further evaluated our trained MTMDVRP100 models on CVRPLib Set-X instances to look at its performance in terms of OOD generalization. Set-X-1 contains 28 instances of various distributions and sizes, ranging from 101 nodes to 251 nodes. Set-X-2 contains 32 instances of various distributions and sizes, ranging from 502 nodes to 1001 nodes. Below are the average performance. Note that since we are using MTMDVRP100 models, MVMoE-Deeper could not be run as it is too large to be trained. We also report the result of SHIELD-Ep400 as it is comparable to MVMoE in terms of in-task in-distribution performance.
> > >
> > > |Set-X-1||POMO-MTVRP||MVMoE||MVMoE-Light||SHIELD-MoD||SHIELD||SHIELD-Ep400||
> > > |:-:|:-:|:-:|:-:|:-:|:-:|:-:|:-:|:-:|:-:|:-:|:-:|:-:|:-:|
> > > ||Opt.|Obj.|Gap|Obj.|Gap|Obj.|Gap|Obj.|Gap|Obj.|Gap|Obj.|Gap|
> > > |Avg.|31280|33601|7.4148%|33111|6.0845%|33174|6.1773%|32902|5.1979%|**32703**|**4.6437%**|32897|5.3961%
> > >
> > > |Set-X-2||POMO-MTVRP||MVMoE||MVMoE-Light||SHIELD-MoD||SHIELD||SHIELD-Ep400||
> > > |:-:|:-:|:-:|:-:|:-:|:-:|:-:|:-:|:-:|:-:|:-:|:-:|:-:|:-:|
> > > ||Opt.|Obj.|Gap|Obj.|Gap|Obj.|Gap|Obj.|Gap|Obj.|Gap|Obj.|Gap|
> > > |Avg.|101874|115725|14.0802%|116136|14.9631%|114225|12.7539%|111534|9.8527%|**111598**|**9.8164%**|111905|10.0618%|
> > >
> > > As shown, **SHIELD outperforms all baselines for OOD generalization on CVRPLib**.
> > >
> > > **[W6: Model capacity & runtime]:** We averaged the runtime of the models reported in Appendix A.8 (Tables 13-21) and updated Table 1 to reflect them. In specific, it refers to the average solving time on 1000 test instances over 9 countries. For convenience, we tabulate the number of parameters and runtimes here. From the table, we can see that utilizing SHIELD improves the inference time of a denser MVMoE-Deeper, and also renders MTMDVRP100 trainable.
> > >
> > > |Model|Num. Parameters|Runtime on MTMDVRP50|Runtime on MTMDVRP100|
> > > |:-:|:-:|:-:|:-:|
> > > |POMO-MTVRP|1.25M|2.74s|8.30s|
> > > |MVMoE|3.68M|3.72s|11.21s|
> > > |MVMoE-Light|3.70M|3.45s|10.38s|
> > > |MVMoE-Deeper|4.46M|9.23s|OOM|
> > > |SHIELD-MoD|4.37M|5.43s|17.70s|
> > > |SHIELD|4.59M|6.16s|20.07s|

---

> > > > ### Author Response · Authors · 2024-11-23
> > > > **Response to Reviewer n7rT - Part 4**
> > > >
> > > > **[W7: Writing]:** Thanks for your comment. We have followed your suggestions to polish our writing. Please see our updated manuscript. Here, we briefly summarize our major changes:
> > > > - We have revamped Figure 1 to showcase the overall process from sampling data to training the model and finally inference.
> > > > - We added a clearer description of the problem setup in lines 198-210 that accurately depicts the MTMDVRP scenario and its practicality.
> > > > - We describe the data and features faced by the model at each training epoch and decoding step for the MTMDVRP in lines 241-247.
> > > > - We remove confusing notations in section 4.3 and added a better description of the operation of a MoD layer in lines 279-285.
> > > > - We provide more details on the features used to represent various tasks in the soft clustering mechanism in lines 318-324.
> > > > - We describe the overall process of the soft clustering algorithm in lines 334-337, and add a full algorithm for it in Algorithm 1 in Appendix A.4.
> > > > - We update Table 1 with averaged runtimes from Table 13 to Table 21.
> > > > - We discuss the implications of various models and their meaning in greater detail in lines 420-428.

---

> ### Comment · Reviewer_n7rT · 2024-11-26
> **Reply**
>
> Thank you for your rebuttal. However, my decision remains the same. I believe this paper could be improved and resubmitted.

---

> ### Author Response · Authors · 2024-11-26
>
> Dear Reviewer #n7rT,
>
> Thank you for your previous constructive comments, which has indeed helped us to improve our paper. Following the comments from you and other reviewers, we have carefully revised the paper and uploaded the updated version in accordance with the ICLR policy. As outlined in the ICLR reviewer guidelines:
>
> > **The discussion phase at ICLR is different from most conferences in the AI/ML community.** During this phase, reviewers, authors and area chairs engage in asynchronous discussion and **authors are allowed to revise their submissions to address concerns** that arise. It is crucial that you are actively engaged during this phase. **Maintain a spirit of openness to changing your initial recommendation (either to a more positive or more negative) rating**.
>
> **We kindly request that you review our revised paper and let us know if there are any remaining concerns.** As the rebuttal phase is still active with a week to go, we would be grateful for the opportunity to address any further points you might raise. We deeply appreciate the time and effort you have dedicated to reviewing our work and look forward to your feedback.

---

> ### Author Response · Authors · 2024-11-26
>
> Dear Reviewer #n7rT,
>
> We have made significant efforts to address all your concerns. For your convenience, we have tabulated the key revisions and TL;DR points below based on your previous constructive comments.
>
> |Review Points|Location in the Paper|TL'DR|
> |:-|:-|:-|
> |W1: More details about MTMDVRP|Main Paper Figure 1, lines 198-210, lines 241-247|The MTMDVRP is significant as it is a major step towards the **foundation NCO model** across both tasks and distribution while **reflecting real-world challenges** faced by logistics companies.|
> |W2: Benefits of MoD| Main Paper lines 420-428|SHIELD compromises between a deeper decoder and learning token importance. This trade-off forces the model to construct meaningful representations capable of strong generalization across **both tasks and distributions**.|
> |W3: Findings of MoD| Main Paper lines 179, lines 420-428|Our work not only validates the training benefits identified by Raposo et al., **but also reveals a novel and significant generalization advantage within the NCO domain**.|
> |W4: Selection of dataset|Main Paper Figure 1|The SHIELD model **is actually trained on 3 distributions, rather than all 9**, and demonstrated strong in-distribution and zero-shot out-of-distribution performance on both tasks and distributions.|
> |W5 & W6: OOD Generalization| Appendix A.9 Table 7, Table 8, Appendix A.10 Table 9, Table 10|SHIELD and even its under-trained versions **outperform all models in all aspects of generalization**.|
> |W6: Model capacity & runtime| Main Paper Table 1, Appendix A.5 Table 5|We conducted **all required experiments** and confirmed that **SHIELD, when matching in-distribution performance, still maintains excellent OOD performance**.|
>
> We hope the table above provides clarity and assists in your review process.

---

> > ### Author Response · Authors · 2024-12-02
> >
> > Dear Reviewer #n7rT,
> >
> > Thank you for your time and effort in reviewing our manuscript and rebuttals. As the author-reviewer discussion is ending, we hope that our clarifications and revisions have addressed your concerns effectively. In light of that, we hope you can reconsider your evaluation. We are open and keen to discuss any remaining concerns and questions you might have. We sincerely thank you very much.

---

### Official Review · Reviewer_6GgF · 2024-11-04

**Soundness:** 4
**Presentation:** 3
**Contribution:** 3
**Rating:** 6
**Confidence:** 4

**Summary:**

This paper introduces the Multi-Task Multi-Distribution Vehicle Routing Problem (MTMDVRP), an extension of the MTVRP. The MTMDVRP effectively captures the complexities inherent in real-world industrial applications by incorporating various realistic customer distributions. To address these challenges, the authors propose a neural solver, SHIELD, which integrates soft clustering, Mixture of Experts, and Mixture-of-Depths (MoD), demonstrating remarkable generalization across various variants of VRP.

**Strengths:**

1.This paper introduces the MTMDVRP,, a variant of vehicle routing problems that aligns more closely with practical scenarios.

2.By effectively combining MoE and MoD, and soft clustering, the proposed neural solver SHIELD demonstrates outstanding performance across a range of VRP variant tasks.

3.The authors conduct extensive ablation experiments that involve nearly all modules, including MoD, MoE, and soft clustering, with detailed descriptions of the experimental procedures and results.

**Weaknesses:**

1. The meanings of $\mathcal{D}_t, z_t, t_t, l_t$ and $o_t$ in Figure 1 are not clearly defined. All symbols used in the figure should be adequately introduced in the text.


2. Section 4, Methodology, lacks clarity in its description and does not comprehensively explain the forward process of the model. First, in Section 4.3, the $k$ and $N-k$ are introduced, followed by $\beta$ and other components. However, the transition to $\beta$ feels abrupt, necessitating multiple readings for clarity. Additionally, the calculation of $\alpha_d$ in lines 312-319 appears abrupt, and the term $W_{\theta}^{T}$ is not sufficiently explained either beforehand or subsequently. I recommend that the authors reorganize the logic in Section 4 to facilitate a clearer understanding of the paper's core concepts.


3. As mentioned in the paper, the introduction of MoD and soft clustering enhances generalization; however, the generalization capabilities for larger-scale problems have not yet been tested. Therefore, I think the results of larger-scale experiments should be also presented.


4. Although the effects of soft clustering are briefly discussed in lines 468-472, this section is concise and does not adequately explain the advantages of this method. I suggest that the authors provide a detailed experimental comparison between SHIELD-MoD and SHIELD to offer a more intuitive and in-depth understanding of the benefits introduced by this approach.

**Questions:**

1. Please refer to the weaknesses. I am particularly concerned about weaknesses 3 and 4.

2. I think the authors should provide a unified introduction to all symbols and mathematical notations used throughout the paper. For example, including a dedicated section for the Preliminary may significantly facilitate reader understanding.

---

> ### Author Response · Authors · 2024-11-23
> **Response to Reviewer 6GgF - Part 1**
>
> We thank the reviewer for their positive feedback on the practicality of MTMDVRP, as well as the recognition of its outstanding performance and the detailed ablation studies. We would like to address your concerns as follows:
>
> **[W1: Meaning of symbols]:** Sorry for the confusion. We have revised the manuscript to better reflect the notations used and have added a notation table in Appendix A.6 for all symbols. For convenience, we provide the description of $\mathcal{D}_t = \{z_t, l_t, t_t, o_t \}$ here as well, copied verbatim from the manuscript (lines 241-247).
>
> *For our setup, we adopt the following feature set. At each epoch, we are faced with a problem instance $i$ such that $\mathcal{S}_i=\{x_i, y_i, \delta_i, w^o_i, w^c_i \}$, where $x_i$ and $y_i$ are the respective coordinates, $\delta_i$ the demand, $w^o_i$ and $w^c_i$ the respective opening and closing times of the time window. This is passed through the encoder resulting in a set $\mathbf{H}$ of $d$-dimensional embeddings. At the $t$-th decoding step, the decoder receives this set of embeddings $\mathbf{H}$, the clustering embeddings $\mathbf{C}$, and a set of dynamic features $\mathcal{D}_t = \{z_t, l_t, t_t, o_t \}$, where $z_t$ denotes the remaining capacity of the vehicle, $l_t$ the length of the current partial route, $t_t$ the current time step, and $o_t$ indicates if the route is an open route or not.*
>
> **[W2: Clarity of Section 4]:** We value your feedback! We have carefully revised the manuscript to enhance clarity. The changes can be found in the highlighted section of Section 4. As a whole, we provided more detailed description of MTMDVRP in lines 198-210, changed Figure 1 to reflect the overall process of our approach, and added clear details regarding the variables used in lines 241-247. Additionally, we have improved the explanation and description of the MoD layer in lines 279-285, as well as explained in greater detail the contextual clustering approach in lines 318-325, and lines 334-337. We have also added a full algorithm description of the soft clustering in Appendix A.4, with a full set of mathematical notations in Appendix A.6.
>
> With regards to section 4.3, we have opted to remove the confusing notation of $k$ and $N-k$, and only maintain $\beta$. $\beta$ refers to the percentage of tokens allowed through a MoD layer. By producing a set of scores via the router, we take the top $\beta$-th percentile of tokens.
>
> With regards to section 4.4, we have revised the section accordingly and changed some notations to synchronize the overall use of them. $\gamma_k$ is a one-hot encoded vector for task $k$. For a task $k$, $\gamma_k = [\gamma^1_k,\gamma^2_k,\gamma^3_k,\gamma^4_k]$, where $\gamma^1_k$ denotes *open*, $\gamma^2_k$ denotes *time-window*, $\gamma^3_k$ denotes *route length*, and $\gamma^4_k$ denotes *backhaul* constraints. $W_\theta^\top$ refers to a set of learnable parameters used to convert the one-hot encoded vector $\gamma_k$ into a latent representation $\alpha_k$.
>
> We hope this provides greater clarity and welcome any additional suggestions you may have!
>
> **[W3: Large-scale Generalization]:** Thanks for your suggestion. The focus of this paper is regarding the generalization properties across tasks and distributions. Nevertheless, we recognize the importance of size generalization and have further conducted multiple experiments on larger-scale CVRP. We apply our MTMDVRP100 models to two sets of data from Set-X in the CVRPLib, used in the DIMACS competition. Set-X-1 contains problems of various distributions and sizes, ranging from **101 nodes to 251 nodes (28 instances)**. Set-X-2 contains problems of various distributions and sizes, ranging from **502 nodes to 1001 nodes (32 instances)**. Below are the average performance. Note that since we are using MTMDVRP100 models, MVMoE-Deeper could not be run as it is too large to be trained on a A100-80GB GPU.
>
> |Set-X-1||POMO-MTVRP||MVMoE||MVMoE-Light||SHIELD-MoD||SHIELD||SHIELD-Ep400||
> |:-:|:-:|:-:|:-:|:-:|:-:|:-:|:-:|:-:|:-:|:-:|:-:|:-:|:-:|
> ||Opt.|Obj.|Gap|Obj.|Gap|Obj.|Gap|Obj.|Gap|Obj.|Gap|Obj.|Gap|
> |Avg.|31280|33601|7.4148%|33111|6.0845%|33174|6.1773%|32902|5.1979%|**32703**|**4.6437%**|32897|5.3961%
>
> |Set-X-2||POMO-MTVRP||MVMoE||MVMoE-Light||SHIELD-MoD||SHIELD||SHIELD-Ep400||
> |:-:|:-:|:-:|:-:|:-:|:-:|:-:|:-:|:-:|:-:|:-:|:-:|:-:|:-:|
> ||Opt.|Obj.|Gap|Obj.|Gap|Obj.|Gap|Obj.|Gap|Obj.|Gap|Obj.|Gap|
> |Avg.|101874|115725|14.0802%|116136|14.9631%|114225|12.7539%|111534|9.8527%|**111598**|**9.8164%**|111905|10.0618%|
>
>
> As shown by the above 2 tables, our model provides **significant benefits in terms of size generalization** as well. In addition to this, we have added another model, SHIELD-Ep400. This model is a checkpoint of SHIELD at the 400th epoch, where we find that its in-task in-distribution performance is similar to that of MVMoE.

---

> > ### Author Response · Authors · 2024-11-23
> > **Response to Reviewer 6Ggf - Part 2**
> >
> > **[W4: Impact of clustering]:** We would like to first clarify that SHIELD-MoD is a variant of SHIELD **without** the soft-clustering mechanism. This baseline was included in the comparison of the original Table 1 so as to show the impacts of clustering and the benefits it brings out in terms of generalization. We note that having the soft-clustering mechanism, as seen in Table 1, especially in the MTMDVRP100 case, **improves the model's generalization to the out-task out-distribution** setting. To further discuss the impact of the number of clusters to the model, we include an ablation study in Table 3 of the manuscript - results show that having too many is detrimental to the solution. This further supports our claim of **sparsity** in the model. Finally, the large-scale generalization experiments in the previous response W3  further shows the benefits of having the clustering module where we further benchmark the performance of SHIELD versus SHIELD-w/o clustering (i.e., SHIELD-MoD).
> >
> > We provide additional discussion on the benefits of clustering in SHIELD, particularly in improving generalization. Clustering addresses the challenges of the MTMDVRP setup by enabling the model to identify shared patterns across tasks and distributions. In the latent space, the soft clustering mechanism facilitates information exchange among dynamic clusters, enabling the model to capture high-level, generalizable features from neighboring hidden representations. This improves the model’s understanding of the node selection process and enhances decision-making. A limited number of clusters also promotes abstraction, encouraging the model to focus on broadly applicable patterns rather than overfitting to task-specific details. However, too many clusters dilute this effect, leading to over-segmentation and reduced generalization as the model prioritizes more complex patterns over shared structures. This aligns well with the empirical findings presented above.
> >
> > **[Q2: A unified introduction to all symbols and notations]:** Thanks for your comments. Following your suggestions, we add an introduction to all symbols and mathematical notations in Appendix A.6.

---

> ### Comment · Reviewer_6GgF · 2024-11-27
> **Respond to Author**
>
> Thank you for your response. The revised version has indeed improved the paper's readability, enhancing its overall clarity. The score I provided in the initial review already reflects my assessment of the methodology and experimental sections. I will temporarily retain this score, though I believe the rating could reasonably be raised to 7 if such an option were available.

---

> > ### Author Response · Authors · 2024-11-27
> >
> > Dear Reviewer #6GgF,
> >
> > We sincerely thank you for your support of our work and for considering raising the score beyond 6. If there are any concerns or comments we could further clarify, we will gladly address them promptly.
> >
> > Thank you once again.

---

### Author Response · Authors · 2024-12-04

Dear Reviewers, AC, SAC,

We thank you all for your feedback on our work. We summarize all our responses to your concerns here. We sincerely hope that the set of additional experimental results shows that our work is impactful.

We have presented a new architecture, SHIELD, capable of solving the Multi-Task Multi-Distribution VRP (MTMDVRP). This is a highly significant problem that represents a practical scenario that the industry faces.

**[What is the MTMDVRP]:** The MTMDVRP can be described as follows. *Suppose a well-known logistic company X has established its presence in a handful of countries. It is able to train a model based on collected data across these countries. Now, if company X wishes to expand to newer ones, it definitely will face data from new distributions, and potentially faces new tasks. As such, it is highly beneficial that its trained model is able to be quickly applied to new incoming data - suggesting the need for the model to be **robust** to new unseen tasks and distributions.* From the machine learning perspective, foundation models typically exhibit similar behavior, as they are unified models that have learnt general representations capable of diverse applications across tasks and distributions. The MTMDVRP scenario is a crucial scenario that exhibits such challenges.

**[Why is SHIELD better]:**  We believe that SHIELD presents an architecture that aims to learn a minimal representation set across both tasks and distributions. Having a set of shared parameters and enforcing sparsity on the network forces it to learn strong representations that are to be **highly useful for both task and distribution generalization**. The combination of MoD and a context-based soft-clustering results in SHIELD, a **flexible foundation model** that displays **superior predictive and generalization properties**.

**[Similar predictive power models]:** We identified earlier training epochs of SHIELD, SHIELD-Ep400 and SHIELD-Ep600 that correspond to similar performances in fully trained MVMoE and MVMoE-Deeper models. We find that these models exemplify **stronger generalization** across both tasks and distributions **compared to their equivalents**.

**[Size generalization]:** A concern was regarding size generalization. We applied our trained MTMDVRP100 models to Set-X in the CVRPLib instance, consisting of 60 instances ranging from 101 to 10001 nodes. Overall, we find that **SHIELD is the significantly superior model** while **not having been** trained specifically for any form of size generalization.

**[Are distributions important]:** A concern raised was whether the non-uniform distributions were important and whether or not the models trained on uniform data only were able to perform well in a cross-distribution test. Our experiments conclude that when the models are not trained on any structured distributions and only on the uniform one, there is a degradation in performance all around. This necessitates the training of a sufficiently **flexible model on varied and structured distributions**.

**[MDVRP]:** Another concern regarding distribution generalization was whether SHIELD is specifically catered to the MTMDVRP case. Experiments show, SHIELD displays **superior performance in and out of distribution** for the MDVRP case as well.

**[Significance of MoD]:** One concern noted that the benefits of having MoD instead of MoE were not apparent. In contrast to MoE, the MoD employed here presents 3 layers in the decoder, each of which **independently** decides a fraction of the tokens to be processed. Since all parameters are shared, the sparsity of the network forces it to learn the best set of parameters that balances between the shared and task(and/or distribution)-specific information. Experiments show that reducing sparsity improves in-task in-distribution prediction at the **expense of generalization**.

**[Model size and inference speeds]:** We averaged the runtimes of the models from the Appendix A.8 and updated Table 1 to reflect them, and tabulate the number of parameters and runtimes. SHIELD **improves the inference time of a denser MVMoE-Deeper**, and also renders MTMDVRP100 trainable.

**[MTMDVRP vs MDVRP]:** One reviewer raised concerns regarding previous works that have addressed either cross-task or cross-distribution generalization. These works have focused on training regimes that enable the original transformer network to be applied to various distributions or tasks. They require some form of pre-training, data from the task to transfer to, or sufficiently varied data in the training distribution. Instead, we attack the generalization aspect with an architecture that embodies sparsity which displays superior **zero-shot out-of-distribution generalization across both tasks and distributions concurrently**.

Overall, there were some concerns regarding writing clarity and mathematical notations. We have revised the manuscript accordingly and highlighted the changes in blue.

---

### Meta-Review · Area_Chair_bqxw · 2024-12-17

**Metareview:**

This paper proposes SHIELD to address the Multi-Task Multi-Distribution Vehicle Routing Problem (MTMDVRP). The authors claim that SHIELD, with its sparsity-inducing Mixture-of-Depths (MoD) and context-based clustering for hierarchy, outperforms existing methods on multiple VRP variants and shows strong generalization capabilities. However, the paper has several flaws. Strengths include its attempt to handle a more practical and complex MTMDVRP scenario, and the comprehensive ablation studies. But the weaknesses are prominent. There are issues with the clarity of the paper, such as unclear notations and a lack of coherence in the methodology section. The novelty of the proposed MoD and clustering techniques is not well-established, as they seem to be incremental improvements rather than significant breakthroughs. The generalization claims, especially for larger-scale problems, are not thoroughly validated. Overall, the paper fails to convince in terms of its scientific contribution and the rigor of its presentation, leading to the decision of rejection.

**Additional Comments On Reviewer Discussion:**

During the rebuttal period, reviewers pointed out various concerns, including the lack of clarity in explanations and notations, the need for more detailed analysis of the model's performance, and questions about the novelty and effectiveness of certain components. The authors responded by revising the manuscript to clarify notations, adding more experimental results, and attempting to justify the design choices. However, these efforts did not fully address the fundamental issues. The additional experiments, while somewhat helpful, did not sufficiently demonstrate the superiority and novelty of the proposed approach. In the final decision, the remaining weaknesses outweighed the improvements made, solidifying the rejection.

---

### Decision · Program_Chairs · 2025-01-22

Reject